# THE COVERAGE PRINCIPLE: HOW PRE-TRAINING ENABLES POST-TRAINING

**Fan Chen**
MIT
fanchen@mit.edu

**Audrey Huang**
UIUC
audreyh5@illinois.edu

**Noah Golowich**
Microsoft Research NYC
nzg@mit.edu

**Sadhika Malladi**
Microsoft Research NYC
sadhika.malladi98@gmail.com

**Adam Block**
Columbia University
adam.block@columbia.edu

**Jordan T. Ash**
Microsoft Research NYC
ash.jordan@microsoft.com

**Akshay Krishnamurthy**
Microsoft Research NYC
akshaykr@microsoft.com

**Dylan J. Foster**
Microsoft Research NYC
dylanfoster@microsoft.com

## ABSTRACT

Language models demonstrate remarkable abilities when pre-trained on large text corpora and fine-tuned for specific tasks, but how and why pre-training shapes the success of the final model remains poorly understood. Notably, although pre-training success is often quantified by cross-entropy loss, cross-entropy can be a poor predictor of downstream performance. Instead, we provide a theoretical perspective on this relationship through the lens of *coverage*, which quantifies the probability mass the pre-trained model places on high-quality responses and which is necessary and sufficient for post-training and test-time scaling methods such as Best-of-N to succeed. Our main results develop an understanding of *the coverage principle*, a phenomenon whereby next-token prediction (more generally, maximum likelihood) implicitly optimizes toward a model with good coverage. In particular, we uncover a mechanism that explains the power of coverage in predicting downstream performance: *coverage generalizes faster than cross-entropy*, avoiding spurious dependence on problem-dependent parameters such as the sequence length. We also study practical algorithmic interventions with provable benefits for improving coverage, including (i) model/checkpoint selection procedures, (ii) gradient normalization schemes, and (iii) test-time decoding strategies.

## 1 INTRODUCTION

The remarkable capabilities of language models stem from a two-stage training process: (1) large-scale pre-training via next-token prediction with the cross-entropy loss (predicting what token should follow a prefix) and (2) targeted post-training—typically via reinforcement learning—to adapt the model to specific domains and tasks. Investing more compute and data into pre-training often enables post-training to produce a stronger model, but theoretical understanding of how these stages interact is limited. Indeed, despite substantial investment into scaling pre-training (Gadre et al., 2025; Sardana et al., 2024; Hoffmann et al., 2022), several works have demonstrated that starting post-training from a better next-token predictor does not ensure stronger performance on downstream tasks (Liu et al., 2022; Zeng et al., 2025; Chen et al., 2025; Lourie et al., 2025). Motivated by this disconnect, we theoretically investigate the connection between pre-training objectives and downstream success, asking:

> *Can we precisely characterize the relationship between the next-token prediction loss and downstream performance? What metrics are most predictive of downstream success?*

Motivated by the recent interest in test-time scaling, we focus our attention on post-training via Best-of-$N$ (BoN) sampling or reinforcement learning with verifiable rewards. For a prompt $x$, Best-of-N draws $N$ responses $y$ from the model and returns the best response according to a task-specific

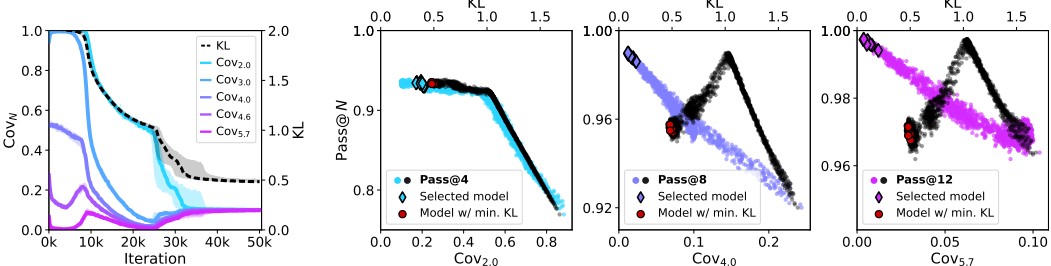

Figure 1: **The coverage profile predicts** Pass@$N$ **better than KL divergence.** *We train models in a graph reasoning task and record KL divergence, coverage profile (both measured w.r.t. $\pi_D$), and* Pass@$N$ *performance; see Appendix D for details. Left: Convergence of coverage and KL divergence over training, showing that KL improves monotonically but coverage can* degrade *with training. Right: Scatter plots of KL (top axis),* $\text{Cov}_{N/2}$ *(lower axis) and* Pass@$N$ *of checkpoints. Although KL and* $\text{Cov}_N$ *exhibit comparable predictive power for small $N$,* $\text{Cov}_N$ *is a better predictor for large $N$. Also visualized are checkpoints selected via the tournament procedure of Eq. (14) (marked ◇) and by minimizing KL (marked red), demonstrating that the former selects better models for* Pass@$N$.

reward. Several prior works have demonstrated that the performance of BoN is strongly indicative of how well the model will perform after post-training via reinforcement learning (Yue et al., 2025; Wu et al., 2025).

Our starting point is the observation that cross-entropy alone cannot provide meaningful answers to the questions above; see Figure 1, which illustrates that cross-entropy can be *anti-correlated* with BoN performance, echoing Chen et al. (2025). Instead, we show that the missing link is the *coverage profile*, a refinement of cross-entropy that explicitly quantifies the model's ability to assign sufficient probability to rare but high-quality responses.

**Definition 1.1** (Coverage profile). *The coverage profile of a model $\widehat{\pi}$ for a distribution $\pi$ is*

$$\text{Cov}_N(\pi \parallel \widehat{\pi}) := \mathbb{P}_{x \sim \mu, y \sim \pi(\cdot|x)}\left[\frac{\pi(y \mid x)}{\widehat{\pi}(y \mid x)} \geq N\right], \tag{1}$$

*where $N \geq 1$ is the number of Best-of-N sampling attempts.*

Here, $y$ is the full response when prompted with $x$, $\pi$ represents the pre-training data distribution, which we presuppose covers downstream tasks of interest, and $\widehat{\pi}$ is the pre-trained model. We prove that a **good coverage profile is necessary and sufficient for Best-of-N to succeed** (see Section 2, as well as Propositions F.6 and F.7). This is highlighted in Figure 1, where we find that the coverage profile is correlated with downstream performance for Best-of-N (which is exactly Pass@$N$), even when cross-entropy is not.[1] Motivated by this characterization of BoN performance, we ask: *When, and through what mechanism, does next-token prediction produce a model $\widehat{\pi}$ with good coverage?*

## 1.1 CONTRIBUTIONS

We develop a theoretical understanding of **the coverage principle**, whereby next-token prediction implicitly optimizes toward a model with good coverage, inheriting the training corpus' coverage over tasks of interest.

**Cross-entropy: Scaling laws and limitations (Section 3).** We begin by deriving provable scaling laws that link cross-entropy—specifically, a certain sequence-level notion—to coverage and hence downstream performance, but show that cross-entropy can be sensitive to sequence length and other problem parameters, leading to vacuous predictions; this motivates our main results.

**Next-token prediction implicitly optimizes coverage (Section 4).** The first of our main theoretical results (Theorem 4.1) is a new generalization analysis for next-token prediction (more generally, maximum likelihood) that exploits the unique structure of the logarithmic loss to show that **coverage can generalize faster than cross-entropy**; we refer to this as the coverage principle. Concretely,

---

[1]Formally, the coverage profile refines cross-entropy/KL divergence; the former is the cumulative distribution function (CDF) of the log density ratio $\log \frac{\pi(y|x)}{\widehat{\pi}(y|x)}$, while KL divergence is the mean; see Remark E.1.

our analysis shows that the coverage profile for models learned with next-token prediction (i) avoids spurious dependence on problem-dependent parameters such as sequence length (in contrast to cross-entropy), and (ii) converges *faster* still as the tail parameter $N$ is increased. Our analysis—which is similar in spirit to Mendelson's *small ball method* (Mendelson, 2014; 2017)—can be viewed as giving a new, fine-grained understanding of maximum likelihood.

**Stochastic gradient descent through the lens of coverage (Section 5).** The preceding results apply to general model classes $\Pi$, but consider the empirical maximizer of the next-token prediction (maximum likelihood) objective, in the vein of classical techniques in learning theory. For the second of our main results, we focus on a specific model class—overparameterized autoregressive linear models (3)—but take a more realistic approach and analyze stochastic gradient descent (SGD) on the next-token prediction objective, in the one-pass ("compute-optimal") regime. We show that while SGD provably optimizes the coverage profile, it experiences suboptimal dependence on the sequence length $H$. We then show that *gradient normalization* (which is loosely connected to Adam-like updates (Bernstein & Newhouse, 2024)) provably improves coverage, removing dependence on the sequence length.

**Interventions for better coverage (Section 6).** Finally, we look beyond standard next-token prediction and explore families of new interventions aimed at improving coverage in theory.
**(i) Test-time (Section 6.1).** We show that for standard token-level SGD, a decoding strategy inspired based on *test-time training* (Krause et al., 2019; Sun et al., 2024; Akyürek et al., 2025) provably improves coverage.
**(ii) Model/checkpoint selection (Section 6.2).** For selecting the best model (or checkpoint) from a small number of candidates, we give *tournament* procedures that enjoy significantly better coverage profile (particularly with respect to the tail parameter $N$) than naïve validation with cross-entropy.

**Additional results (Appendix G).** Beyond the results above, we show that: (1) MLE can find models with low coverage even in the presence of severe misspecification; (2) coverage can generalize better under additional structural properties of the model class such as convexity (Appendix G.2).

In summary, we believe that coverage offers a new perspective on the connection between pre-training objectives and downstream post-training success. Our results demonstrate that this perspective is mathematically rich and fundamental, opening the door to a deeper understanding; cf. Appendix A.

## 2 PROBLEM SETUP

We now introduce the formal problem setup for the remainder of the paper.

**Next-token prediction and maximum likelihood.** We work in the following setting, which subsumes next-token prediction: $\mathcal{X}$ is the prompt space, $\mathcal{Y}$ is the response space, and $\pi_\mathsf{D} : \mathcal{X} \to \Delta(\mathcal{Y})$ is the data distribution. We are given a dataset $\mathcal{D} = \{(x^i, y^i)\}_{i=1}^n$ where $x^i \sim \mu$ and $y^i \sim \pi_\mathsf{D}(\cdot \mid x^i)$. We consider the maximum likelihood objective

$$\widehat{L}_n(\pi) := \sum_{i=1}^n \log \pi(y^i \mid x^i). \tag{2}$$

and refer to $\widehat{\pi} := \arg\max_{\pi \in \Pi} \widehat{L}_n(\pi)$ as the *maximum likelihood estimator* for a user-specified model class $\Pi$. This is a generalization of the next-token prediction, where $\mathcal{Y} = \mathcal{V}^H$ is a token sequence and $\pi(y \mid x) = \prod_{h=1}^H \pi(y_h \mid x, y_{1:h-1})$ is explicitly autoregressive, so that $\widehat{L}_n(\pi) = \sum_{i=1}^n \sum_{h=1}^H \log \pi(y_h^i \mid x^i, y_{1:h-1}^i)$. We specialize to next-token prediction at certain points but otherwise focus on the general setting. We make the following realizability assumption throughout.

**Assumption 2.1** (Realizability). *The data distribution $\pi_\mathsf{D}$ is realizable by some model $\pi \in \Pi$.*

This formulation captures pre-training and SFT, with some caveats; see Appendix A.1.

**Post-training and the coverage profile.** Given a reward function $r_\mathsf{T}(x, y) \in \{0, 1\}$ representing success at a downstream task $\mathsf{T}$, the goal is to fine-tune $\widehat{\pi}$—through reinforcement learning or test-time scaling—to obtain near-optimal reward. We show (Propositions F.6 and F.7) that for any task-specific comparator policy $\pi_\mathsf{T} : \mathcal{X} \to \Delta(\mathcal{Y})$, Best-of-N sampling with $\widetilde{\Theta}(N)$ samples

satisfies $\mathbb{E}_{x \sim \mu}[r_{\mathsf{T}}(x, \pi_{\mathsf{T}}(x)) - r_{\mathsf{T}}(x, \widehat{\pi}^{\mathsf{BoN}}(x))] \asymp \mathsf{Cov}_N(\pi_{\mathsf{T}} \parallel \widehat{\pi})$, so a good coverage profile for $\pi_{\mathsf{T}}$ is sufficient for high reward. Further, while less well understood, some form of coverage is thought to be necessary for the success of post-training methods like GRPO (Yue et al., 2025).

Returning to pre-training, it is clear that there is little hope that next-token prediction will produce a model $\widehat{\pi}$ with good coverage with respect to a downstream task unless the data distribution $\pi_{\mathsf{D}}$ itself has reasonable coverage with respect to this task. We therefore posit that the data distribution covers such a downstream task, in the sense that it includes high-reward responses with some bounded-below probability. Since coverage satisfies a transitivity property, it follows that coverage with respect to $\pi_{\mathsf{D}}$ implies coverage with respect to the optimal policy for the downstream task. For example, if $\pi_{\mathsf{D}}$ has a 10% chance of generating a correct response, and $\mathsf{Cov}_{N/10}(\pi_{\mathsf{D}} \parallel \widehat{\pi}) = \varepsilon$, then we get $10\varepsilon$ error.[2] Thus, **going forward, we focus on understanding when next-token prediction achieves good coverage $\mathsf{Cov}_N(\pi_{\mathsf{D}} \parallel \widehat{\pi})$ relative to the data distribution $\pi_{\mathsf{D}}$ itself,** and avoid concerning ourselves with specific details of the task policy $\pi_{\mathsf{T}}$ or the specific relationship between $\pi_{\mathsf{T}}$ and $\pi_{\mathsf{D}}$.

**Autoregressive linear models.** We analyze next-token prediction and maximum likelihood for general model classes $\Pi$, but our running example throughout the paper will be the class $\Pi$ of *autoregressive linear models*, defined by a known feature map $\phi : \mathcal{X} \times \mathcal{V}^\star \to \mathbb{R}^d$. For each parameter $\theta \in \Theta \subset \mathbb{R}^d$, the model $\pi_\theta = (\pi_\theta)_{h=1}^H$ is defined by

$$\pi_\theta(y_h \mid x, y_{1:h-1}) \propto \exp(\langle \theta, \phi(x, y_{1:h}) \rangle). \tag{3}$$

In practice, autoregressive sequence models—such as those based on transformers—generate each token by sampling from a softmax distribution whose logits are given by a linear combination of learned features (Radford et al., 2019). Eq. (3) simplifies this by freezing the feature map, yet remains expressive enough to model complex non-Markovian dependencies, depending on the choice of features.

**Assumption 2.2.** *We assume* $\Theta \subseteq \{\theta : \|\theta\| \leq 1\}$ *is convex, and* $\sup_{h,x,y_{1:h}} \|\phi(x, y_{1:h})\| \leq B$.

## 3 CROSS-ENTROPY AND COVERAGE: SCALING LAWS AND LIMITATIONS

A natural approach to understanding when next-token prediction achieves good coverage is to appeal to cross-entropy—perhaps first showing that next-token prediction achieves low cross-entropy (which is true asymptotically), and then relating cross-entropy to coverage. In this section we motivate our main results by showing that while this is possible in a weak sense, it does not yield predictive guarantees for downstream performance in the finite-sample regime.

Define the *sequence-level* cross-entropy for $\widehat{\pi}$ as $D_{\mathsf{CE}}(\pi_{\mathsf{D}} \parallel \widehat{\pi}) := \mathbb{E}_{\pi_{\mathsf{D}}}\left[\sum_{h=1}^H \log \frac{1}{\widehat{\pi}(y_h \mid x, y_{1:h-1})}\right]$. Since $\mathbb{E}_{\mathcal{D} \overset{\text{i.i.d.}}{\sim} \pi_{\mathsf{D}}}[\widehat{L}_n(\pi)] = -n \cdot D_{\mathsf{CE}}(\pi_{\mathsf{D}} \parallel \pi)$, one expects that as we scale up compute, number of samples $n$, and model capacity $\Pi$, $D_{\mathsf{CE}}(\pi_{\mathsf{D}} \parallel \widehat{\pi}) \to D_{\mathsf{CE}}(\pi_{\mathsf{D}} \parallel \pi_{\mathsf{D}})$, or equivalently $D_{\mathsf{KL}}(\pi_{\mathsf{D}} \parallel \widehat{\pi}) \to 0$, where $D_{\mathsf{KL}}(\pi_{\mathsf{D}} \parallel \widehat{\pi}) := \mathbb{E}_{\pi_{\mathsf{D}}}\left[\sum_{h=1}^H \log \frac{\pi_{\mathsf{D}}(y_h \mid x, y_{1:h-1})}{\widehat{\pi}(y_h \mid x, y_{1:h-1})}\right]$ is the sequence-level KL divergence.

**A simple scaling law for cross-entropy.** We show below that if the model $\widehat{\pi}$ has reasonable KL divergence to the data distribution, the coverage profile can be bounded:

**Proposition 3.1** (KL-to-coverage; see Proposition F.1)**.** *For all* $N \geq e$, $\mathsf{Cov}_N(\pi_{\mathsf{D}} \parallel \widehat{\pi}) \leq \frac{D_{\mathsf{KL}}(\pi_{\mathsf{D}} \parallel \widehat{\pi})}{\log(N/e)}$.

Combining Proposition 3.1 with Proposition F.6 and our assumption that $\pi_{\mathsf{D}}$ has good coverage with respect to the downstream task yields a simple "scaling law" for test-time compute with BoN:

> Consider a task of interest with reward $r_{\mathsf{T}}(x, y)$, and suppose the data distribution $\pi_{\mathsf{D}}$ itself has constant probability of success (i.e., sampling $y \sim \pi_{\mathsf{D}}(\cdot \mid x)$ with $r_{\mathsf{T}}(x, y) = 1$). To achieve sub-optimality $\varepsilon$ with Best-of-N, it suffices to choose the compute budget $N$ as
>
> $$N \approx \exp\left(\frac{D_{\mathsf{KL}}(\pi_{\mathsf{D}} \parallel \widehat{\pi})}{\varepsilon}\right). \tag{4}$$

---

[2]See Proposition F.5 for formal results.

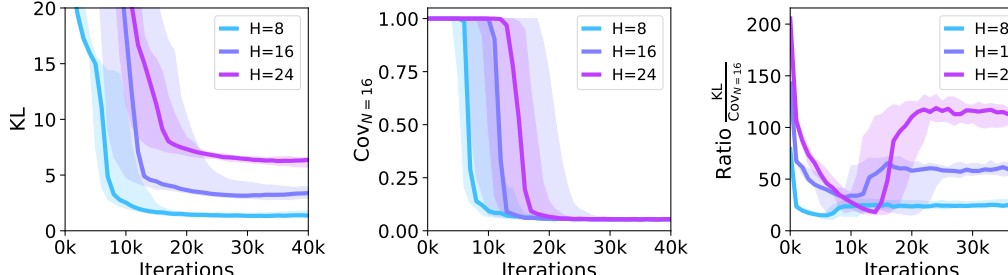

Figure 2: **The coverage profile avoids spurious dependence on sequence length.** *We train models in a graph reasoning task and record their KL divergence and coverage profile, measured w.r.t. $\pi_D$ as we vary the problem horizon (sequence length); see Appendix D for details. Left: Convergence of KL over training for three horizons $H$, demonstrating that KL at convergence scales linearly in the horizon $H$. Center: Convergence of $\mathrm{Cov}_N$ over training, manifesting no dependence on $H$ at convergence. Right: Ratio of KL over $\mathrm{Cov}_N$, showing that Proposition 3.1 can be overly conservative.*

That is, for a fixed model $\widehat{\pi}$ and KL-divergence level $D_{\mathsf{KL}}(\pi_D \,\|\, \widehat{\pi}) \leq D_{\mathsf{CE}}(\pi_D \,\|\, \widehat{\pi})$, Eq. (4) predicts that test-time compute should increase exponentially with the desired accuracy $\varepsilon$.[3]

**Insufficiency of cross-entropy.** At first glance, this seems to be in line with empirical test-time scaling laws (OpenAI, 2024), but there is an issue: While *token-level* cross-entropy has been observed to be modest in contemporary language models (Kaplan et al., 2020; Hoffmann et al., 2022; Xia et al., 2022), the *sequence-level* cross-entropy (and KL-divergence) generally grows with the length $H$ of the sequence, so that Eq. (4) predicts exponential test-time scaling in the sequence length. Moreover, such a law cannot hold if we only assume token-level cross-entropy is bounded; see Proposition F.7.

Is this the end of the story? On the one hand, it is simple to show (Proposition F.2) that Proposition 3.1 is tight for a worst-case pair of models. Moreover, even for the autoregressive linear model in Eq. (3), sequence-level KL divergence scales linearly with the sequence length $H$, as shown in the next result.

**Proposition 3.2.** *Fix $H \in \mathbb{N}$ and $d = 1$. There exists $\phi : \mathcal{X} \times \mathcal{V}^\star \to [-1, 1]$ and induced autoregressive linear class $\Pi$ with parameter space $\Theta = [-1, 1]$, distribution $\mu$ over $\mathcal{X}$, such that for any proper estimator $\widehat{\pi} = \widehat{\pi}(\mathcal{D}) \in \Pi$, there exists data distribution $\pi_D \in \Pi$ such that w.p. at least $0.25$, $D_{\mathsf{KL}}(\pi_D \,\|\, \widehat{\pi}) \geq \frac{H}{4n}$.*

This behavior is reflected empirically in Figure 2 for a graph reasoning task. Yet, for this task, we find (Figure 2) that in spite of large cross-entropy/KL, next-token prediction learns a model $\widehat{\pi}$ with a good coverage profile across a range of sequence lengths and that downstream Best-of-N succeeds. Why is this happening? In light of the discussion above, it must be related to specific inductive bias of the next-token prediction objective itself.

**A glimmer of hope: Case study in Bernoulli models.** To see why large cross-entropy may not be a barrier to coverage, consider perhaps the simplest setting, *Bernoulli models*, where $\mathcal{X} = \{\bot\}$, $\mathcal{Y} = \{0, 1\}$, $\Pi = \{\mathrm{Ber}(p)\}_{p \in (0, 1/2)}$, and $\pi_D = \mathrm{Ber}(p^\star)$ for some small $p^\star \in (0, 1/2)$.

The maximum likelihood model is $\widehat{\pi} = \mathrm{Ber}(\widehat{p})$, where $\widehat{p}$ is the empirical frequency of $y = 1$ in the dataset. We observe that with positive probability (and constant probability if $n \leq 1/p^\star$), the dataset $\mathcal{D}$ will only contain examples where $y = 0$, so that the maximum likelihood model is $\widehat{\pi} = \mathrm{Ber}(0)$. This implies that expected KL divergence is infinite: $\mathbb{E}[D_{\mathsf{KL}}(\pi_D \,\|\, \widehat{\pi})] = +\infty$. However, the coverage profile turns out to be well-behaved; a direct calculation shows that $\mathrm{Cov}_N(\pi_D \,\|\, \widehat{\pi}) \lesssim \frac{\log(\delta^{-1})}{n}$ with probability at least $1 - \delta$ for all $N \geq 2$; this gives hope that even though cross-entropy itself is infinite, maximum likelihood may actually learn a model with good coverage in the background. In what follows, we will show that this is not a fluke, but a general phenomenon.

**Remark 3.1** (Missing mass). *The underlying issue in both of the preceding examples is* missing mass*: there are responses that even a well-generalizing learner will fail to cover, and for these*

---

[3]Neither KL divergence nor the coverage profile are observable quantities (though cross-entropy is an estimable upper bound on KL), so this is a theoretical prediction rather than a practical one as-is; see Remark E.2.

*we may incur a large contribution to the KL-divergence. More generally, KL-divergence and cross-entropy are susceptible to contributions of the scale $\log W_{\max}$ where $W_{\max} = \max_{\pi \in \Pi} \left\| \frac{\pi_D}{\pi} \right\|_\infty$ (which could be as large as $H$, as in Proposition 3.2) when the model does not have enough information to generalize/extrapolate. This phenomenon is particularly pronounced when the prompt distribution is heterogeneous.*

## 4 NEXT-TOKEN PREDICTION IMPLICITLY OPTIMIZES COVERAGE

We now present our main result (Theorem 4.1): due to the unique structure of the logarithmic loss, maximum likelihood can learn models with a good coverage profile even when the cross-entropy is vacuously large. Henceforth, we abbreviate $\mathrm{Cov}_N(\pi) := \mathrm{Cov}_N(\pi_D \,\|\, \pi)$. We make use of the following covering number.

**Definition 4.1.** *For a class $\Pi$ and $\alpha \geq 0$, we let $\mathcal{N}_\infty(\Pi, \alpha)$ denote the size of the smallest cover $\Pi'$ such that for all $\pi \in \Pi$, there exists $\pi' \in \Pi'$ such that $\sup_{x \in \mathcal{X}, y \in \mathcal{Y}} |\log \pi(y \mid x) - \log \pi'(y \mid x)| \leq \alpha$.*

**Theorem 4.1** (Fast generalization for coverage). *Fix $N \geq 8$ and let $c > 0$ be an absolute constant. Suppose Assumption 2.1 holds. With probability at least $1 - \delta$, the maximum likelihood estimator has*

$$\mathrm{Cov}_N(\widehat{\pi}) \lesssim \frac{1}{\log N} \cdot \underbrace{\inf_{\varepsilon > 0} \left\{ \frac{\log \mathcal{N}_\infty(\Pi, \varepsilon)}{n} + \varepsilon \right\}}_{=: \, \mathcal{C}_{\mathrm{fine}}(\Pi, n)} + \underbrace{\frac{\log \mathcal{N}_\infty(\Pi, c \log N) + \log(\delta^{-1})}{n}}_{=: \, \mathcal{C}_{\mathrm{coarse}}(\Pi, N, n)}. \tag{5}$$

Eq. (5) has a *fine-grained term* $\mathcal{C}_{\mathrm{fine}}(\Pi, n)$ and *coarse-grained term* $\mathcal{C}_{\mathrm{coarse}}(\Pi, N, n)$; we interpret each below.

**Fine-grained term.** $\mathcal{C}_{\mathrm{fine}}(\Pi, n)$ evaluates the covering number $\mathcal{N}_\infty(\Pi, \varepsilon)$ at a small scale $\varepsilon$ (typically $\varepsilon \approx \mathrm{poly}(1/n)$), which matches typical bounds for conditional density estimation (e.g., Bilodeau et al. (2023)) in KL divergence; however, unlike KL-based bounds this term has *no explicit dependence on sequence length $H$ or density ratios $\log W_{\max}$*. The term is further scaled by $1/\log N$, which implies that *coverage enjoys faster convergence as we move further into the tail* by increasing $N$; this reflects the unique structure of the logarithmic loss, and may be viewed as a new form of implicit bias.

Summarizing, the fine-grained term in Eq. (5) witnesses the phenomenon we term the *coverage principle*: the coverage profile enjoys faster generalization than cross-entropy; roughly, the rate is what we would expect (via Proposition 3.1) if we could somehow control KL without paying for the sequence length $H$ or density ratio $\log W_{\max}$. See Appendix C for a detailed comparison to standard (asymptotic and non-asymptotic) generalization bounds for maximum likelihood based on Hellinger distance and KL-divergence.

**Coarse-grained term.** The coarse-grained term $\mathcal{C}_{\mathrm{coarse}}(\Pi, N, n)$ captures the *missing mass* phenomenon exemplified by the Bernoulli example in the prequel. This term is not explicitly normalized by $1/\log N$ (compared to the fine-grained term), but depends on the covering number $\mathcal{N}_\infty(\Pi, \alpha)$ only at a very large scale $\alpha \approx \log N$. As such, the dependence on the complexity/richness of $\Pi$ in this term vanishes as we increase $N$.

Overall, while the guarantee in Eq. (5) might look surprising at first glance (particularly the coarse term, as we are not aware of any existing generalization bounds with dependence on covering numbers at such a large scale), we show in Proposition G.1 (Appendix J) that both terms are tight in general.

**Overview of analysis.** The proof of Theorem 4.1 is given in Appendix J (with a high-level sketch in Appendix J.1). The basic idea is to interpret the condition $\mathrm{Cov}_N(\pi) \geq \varepsilon$ as a small ball-like *anti-concentration* condition in the vein of Mendelson (2014; 2017). That is, for models $\pi$ where coverage is large, the condition $\mathrm{Cov}_N(\pi) \geq \varepsilon$ witnesses a *one-sided* bound which implies that the empirical likelihood of $\pi$ is *not too large* with high probability, and thus $\pi$ cannot be a maximum-likelihood solution.

The coarse-grained term $\mathcal{C}_{\mathrm{coarse}}(\Pi, N, n)$ enters because we only need to show that the coverage profile concentrates, not the log loss itself. The fine-grained term $\mathcal{C}_{\mathrm{fine}}(\Pi, n)$ enters from one-sided concentration of the empirical likelihood, with the $1/\log N$ scaling arising from the following form of implicit bias: If an example $(x^i, y^i)$ has $\pi_D(y^i|x^i)/\pi(y^i|x^i) \geq N$, this witnesses a negative contribution of order $\log N$ to the difference $\widehat{L}_n(\pi) - \widehat{L}_n(\pi_D)$.

**Discussion.** We emphasize that while covering numbers are a fundamental and widely used noton of capacity in statistical learning and estimation (van de Geer, 2000; Zhang, 2002; Rakhlin & Sridharan, 2012; Bilodeau et al., 2023), they are conservative from a modern generalization perspective. Nonetheless, Theorem 4.1 shows that they are sufficient to capture rich aspects of generalization for coverage, and we expect that our core analysis techniques can be combined with contemporary advances in generalization theory for overparameterized models (Belkin et al., 2019; Bartlett et al., 2020).

## 4.1 EXAMPLES

To build intuition, we analyze the behavior of Theorem 4.1 under a growth assumption on the covering number, then specialize to autoregressive linear models, showing how they exemplify the coverage principle.

**Corollary 4.1.** *(i) Parametric regime: Suppose that there are parameters $d \geq 2$ and $C \geq 2$ such that $\log \mathcal{N}_\infty(\Pi, \alpha) \leq d \log(C/\alpha)$ for $\alpha \in (0, C/2]$. Then for any $N \geq 8$, with probability at least $1 - \delta$, $\mathsf{Cov}_N(\widehat{\pi}) \lesssim \frac{d\left[[\log(C/\log N)]_+ + \frac{\log(Cn)}{\log N}\right] + \log(1/\delta)}{n}$.*

*(ii) Nonparametric regime: Suppose that there are parameters $C \geq 2$ and $p > 0$ such that $\log \mathcal{N}_\infty(\Pi, \alpha) \leq (C/\alpha)^p$ for $\alpha \in (0, C/2]$. Then for any $N \geq 8$ and $n \geq \log^{1/p} N \cdot (C/\log N)^p$, with probability at least $1 - \delta$, $\mathsf{Cov}_N(\widehat{\pi}) \lesssim \frac{1}{\log N}\left(\frac{C^p}{n}\right)^{\frac{1}{p+1}} + \frac{\log(1/\delta)}{n}$.*

This result shows that for sufficiently rich classes (e.g., when $p > 0$), the fine-grained term dominates the coarse-grained term for $n$ sufficently large. On the other hand, for simple classes (e.g., when $p = 0$), the coarse-grained term can dominate the fine-grained term.

**Autoregressive linear models: Low dimension.** We now consider the autoregressive linear model in Eq. (3). When the dimension $d$ is small, this class satisfies $\log \mathcal{N}_\infty(\Pi, \alpha) \asymp d \log(BH/\alpha)$ (corresponding to the parametric regime in Corollary 4.1), and so, coverage generalizes in a (nearly) horizon-independent fashion, in stark contrast to the cross-entropy lower bound in Proposition 3.2. The only drawback (which is fundamental) is that since the class has low capacity, the coarse-grained term dominates for most parameter regimes, and the improvement as $N$ scales is quite modest.

**Autoregressive linear models: High dimension.** As a more interesting example, we next look at the behavior of next-token prediction for autoregressive linear models in an "overparameterized" regime where the dimension $d$ is arbitrarily large (Zhang, 2002; Neyshabur et al., 2015; Bartlett et al., 2017). Here, we control the richness of the class $\Pi$ by the norm parameter $B$. In this regime, it turns out that in the worst-case, the capacity $\log \mathcal{N}_\infty(\Pi, \alpha)$ scales polynomially in $H$. To address, this we prove a refined version of Theorem 4.1 that adapts to the variance in the data distribution $\pi_\mathsf{D}$, avoiding explicit dependence on sequence length.

Define the *inherent variance* for the data distribution as

$$\sigma_\star^2 := \mathbb{E}_{\pi_\mathsf{D}}\left[\sum_{h=1}^H \left\| \phi(x, y_{1:h}) - \overline{\phi}_{\pi_\mathsf{D}}(x, y_{1:h-1}) \right\|^2\right], \tag{6}$$

where $\overline{\phi}_{\pi_\mathsf{D}}(x, y_{1:h-1}) := \mathbb{E}_{y_h \sim \pi_\mathsf{D}(\cdot | x, y_{1:h-1})}[\phi(x, y_{1:h})]$ is the average feature vector given the prefix $(x, y_{1:h-1})$. We can interpret the inherent variance $\sigma_\star^2$ as a notion of effective sequence length; it captures the number tokens that are "pivotal" in the sense that they have high variation conditioned on the prefix; the name reflects a noted phenomenon in language modeling that most tokens are near-deterministic and easy to predict given their prefix, with only a few having high entropy (Abdin et al., 2024). Thus, while $\sigma_\star^2$ can be as large as $B^2 H$ in the worst case, we expect it to be smaller in general.

**Theorem 4.2** (Overparameterized autoregressive linear models). *Consider the autoregressive linear model (3), and suppose Assumptions 2.1 and 2.2 hold. For any $N \geq 2$, next-token prediction achieves*

$$\mathbb{E}[\mathsf{Cov}_N(\widehat{\pi})] \lesssim \sqrt{\frac{\sigma_\star^2}{n \cdot \log N}} + \frac{B^2}{n}. \tag{7}$$

Similar to Theorem 4.1, the first term in Eq. (7) can be viewed as "fine-grained" and the second term as "coarse-grained"; the former is typically larger, but decreases with the tail parameter $N$, while the latter does not decrease with $N$ but is typically smaller to begin with. We prove (details

in Proposition K.1) that this result is tight in the sense that if $\sigma_\star^2 \asymp H$, $n \geq H$ is indeed necessary to achieve good non-trivial coverage in the overparameterized regime.

We view the introduction of the inherent variance $\sigma_\star^2$ as an instance-dependent notion of complexity for autoregressive models to be a non-trivial conceptual contribution, which may find broader use.

## 5 STOCHASTIC GRADIENT DESCENT THROUGH THE LENS OF COVERAGE

The coverage-based generalization guarantees for next-token prediction in the prequel apply to general model classes $\Pi$, but consider the empirical maximizer $\widehat{\pi} = \arg\max_{\pi \in \Pi} \widehat{L}_n(\pi)$ of the next-token prediction (maximum likelihood) objective, in the vein of classical techniques in learning theory. For our second set of main results, we focus on autoregressive linear models (3) but take a more realistic approach and analyze stochastic gradient descent (SGD) in the single-pass regime. This setup is motivated by contemporary ("compute-optimal") language model training, which typically uses one or fewer passes over the training corpus (Kaplan et al., 2020; Hoffmann et al., 2022).

### 5.1 STOCHASTIC GRADIENT DESCENT HAS SUBOPTIMAL COVERAGE

For the next-token prediction objective, single-pass stochastic gradient descent (SGD) takes the form[4]

$$\theta^{t+1} \leftarrow \mathrm{Proj}_\Theta(\theta^t + \eta \nabla \log \pi_{\theta^t}(y^t \mid x^t)), \tag{8}$$

for $x^t \sim \mu$ and $y^t \sim \pi_{\mathsf{D}}(\cdot \mid x^t)$, where $\eta > 0$ is the learning rate. As the next-token prediction loss $L(\theta) := \mathbb{E}_{\pi_{\mathsf{D}}}[-\log \pi_\theta(y \mid x)]$ is convex under the parameterization (3), we can show that SGD converges to $\pi_{\mathsf{D}}$ in KL divergence. This implies a coverage bound, albeit a suboptimal one.

**Proposition 5.1** (SGD for autoregressive linear models). ***Upper bound:** Suppose Assumptions 2.1 and 2.2 hold. As long as $\eta \leq \frac{1}{2HB^2}$, it holds that $\mathbb{E}\big[\frac{1}{T}\sum_{t=1}^T D_{\mathsf{KL}}(\pi_{\mathsf{D}} \,\|\, \pi_{\theta^t})\big] \leq \frac{4}{\eta T} + 2\eta\sigma_\star^2$. Choosing $\eta$ to minimize this bound gives*

$$\mathbb{E}\big[\tfrac{1}{T}\textstyle\sum_{t=1}^T \mathrm{Cov}_N(\pi_{\theta^t})\big] \lesssim \tfrac{1}{\log N} \cdot \big(\sqrt{\tfrac{\sigma_\star^2}{T}} + \tfrac{B^2 H}{T}\big). \tag{9}$$

***Lower bound:** Suppose that $B \geq c \cdot \log^2(TH)$. Then there exists an autoregressive linear class $\Pi$ such that for any constant step size $\eta > 0$, there exists an instance $\pi_{\mathsf{D}} \in \Pi$ with $\sigma_\star \leq 1$ such that with probability at least $0.5$, the SGD iterates satisfy $\mathrm{Cov}_N(\pi_{\mathsf{D}} \,\|\, \pi_{\theta^t}) \geq c \cdot \min\big\{\frac{H}{T \log N}, 1\big\}$ for any $t \in [T]$.*

The coverage bound in Eq. (9) (which follows by passing from KL to coverage through Proposition 3.1) is similar to Theorem 4.2, except that the second term $\frac{B^2 H}{T}$ has an unfortunate dependence on the sequence length $H$. The lower bound shows that this dependence is tight, and SGD can indeed experience poor coverage. This failure of SGD is related to *heterogeneity* across prompts: there are some prompts for which the effective scale of the gradient in Eq. (8) grows with $H$, leading to divergence unless we use a small learning rate $\eta \lesssim \frac{1}{HB}$. Yet for other prompts, the effective gradient range is small, leading to slow convergence (on the order of $\Omega(H)$ steps) unless $\eta \gg \frac{1}{HB}$.

**Remark 5.1** (Sequence-level SGD). *The update in Eq. (8) can be interpreted as a "sequence-level" form of SGD, since we perform a single gradient step for each full sequence $y^t$ (note that $\nabla \log \pi_{\theta^t}(y^t \mid x^t) = \sum_{h=1}^H \nabla \log \pi_{\theta^t}(y_h^t \mid x^t, y_{1:h-1}^t)$). We view this as a model for what is done in practice, whereby one performs SGD on sequences of tokens spanning some fixed context window. While this context window may be shorter than the full training example (e.g., a long article), understanding the implications of a limited context window is beyond the scope of this work.*

### 5.2 GRADIENT NORMALIZATION IMPROVES COVERAGE

To address the suboptimality of SGD, we consider *gradient normalization* as a simple intervention. For a mini-batch $\mathcal{D} = \{(x^i, y^i)\}_{i=1}^K$ of $K$ samples from $\pi_{\mathsf{D}}$, define the batch stochastic gradient as $\widehat{g}(\theta; \mathcal{D}) = \frac{1}{|\mathcal{D}|} \sum_{(x,y) \in \mathcal{D}} \nabla \log \pi_\theta(y \mid x)$. We consider the following normalized SGD update:

$$\theta^{t+1} \leftarrow \mathrm{Proj}_\Theta\big(\theta^t + \eta \cdot \tfrac{\widehat{g}(\theta^t; \mathcal{D}^t)}{\lambda + \|\widehat{g}(\theta^t; \mathcal{D}^t)\|}\big); \tag{10}$$

---

[4]$\mathrm{Proj}_\Theta(\cdot)$ denotes Euclidean projection onto $\Theta$, so this is SGD on the loss $L(\theta) := \mathbb{E}[-\log \pi_\theta(y \mid x)]$.

here $\mathcal{D}^t$ is a mini-batch with $K$ fresh samples drawn i.i.d. from $\pi_{\mathsf{D}}$, and $\lambda > 0$ is a regularization parameter for numerical stability. We show that this update achieves a horizon-independent coverage bound.

**Theorem 5.1.** *Suppose Assumption 2.1 and Assumption 2.2 hold. Let $T, K \geq 1, N \geq 3$ be given. For an appropriate choice of $\eta, \lambda > 0$, the normalized SGD update (10) achieves the following bound:*

$$\mathbb{E}\left[\frac{1}{T}\sum_{t=1}^{T}\mathsf{Cov}_N(\pi_{\theta^t})\right] \lesssim \sqrt{\frac{\sigma_\star^2}{T\cdot\log N}} + \frac{B^2}{T} + \frac{B}{K\cdot\log N}. \tag{11}$$

*To achieve $\mathbb{E}[\mathsf{Cov}_N(\widehat{\pi})] \leq \varepsilon$ for a target level $\varepsilon > 0$, it suffices to choose $T = O\left(\frac{\sigma_\star^2}{\varepsilon^2\log N} + \frac{B^2}{\varepsilon}\right)$, $K = O\left(\frac{B}{\varepsilon\log N} + 1\right)$, giving total sample complexity $n = TK = O\left(\frac{\sigma_\star^2 B}{\varepsilon^3\log^2 N} + \frac{B^3+\sigma_\star^2}{\varepsilon^2\log N} + \frac{B^2}{\varepsilon}\right)$.*

Theorem 5.1 shows that gradient normalization achieves horizon-independent coverage with a qualitatively similar rate to the guarantee for next-token prediction in Theorem 4.2: To achieve coverage $\varepsilon$, both rates scale as $\mathrm{poly}\left(\frac{\sigma_\star^2}{\log N}, B, \varepsilon^{-1}\right)$, though the dependence on $\varepsilon$ for Theorem 5.1 is worse. We view this as another instance of the coverage principle, as the rate achieved by gradient normalization goes beyond what can be achieved by passing through KL divergence. We emphasize that minibatching alone is not enough to achieve this result; rather, minibatching is necessary to avoid excessive bias once we introduce gradient normalization.

As a remark, the normalized SG update in (10) is closely related to SignSGD (Balles & Hennig, 2018) and *Adam* (Kingma & Ba, 2015) as shown by Bernstein & Newhouse (2024). We believe that similar coverage guarantees could potentially be shown for these methods using our techniques.

**Distillation.** As an additional result, we show (Theorem G.2 in Appendix G.4) that for a *distillation* setting, where $\pi_{\mathsf{D}}$ corresponds to a teacher model and we have access to its per-token logits, we can derive an improved gradient normalization scheme that fully closes the gap with Theorem 4.2.

## 6 Interventions for Better Coverage

In this section, we develop new interventions that improve coverage (and downstream performance) beyond the conventional algorithms analyzed in Sections 4 and 5. We view these results as promising proofs of concept for further research into interventions driven by coverage.

### 6.1 Improving Coverage at Test Time

In this section, we show that a modified decoding strategy based on *test-time training* (or, *dynamic evaluation*) (Mikolov et al., 2010; Krause et al., 2018; 2019; Sun et al., 2024; Akyürek et al., 2025) leads to improved coverage when combined with token-level SGD.

We focus on autoregressive linear models, but depart from Eq. (8) by learning models with a *token-level* SGD update, defined as

$$\theta^{t,h+1} = \mathrm{Proj}_\Theta\left(\theta^{t,h} + \eta\nabla\log\pi_{\theta^{t,h}}(y_h^t \mid x^t, y_{1:h-1}^t)\right), \text{ for } h = 0, \cdots, H-1, \tag{12}$$

and $\theta^{t+1} \equiv \theta^{t+1,0} := \theta^{t,H}$ for $t \in [T]$, and where $(x^t, y_{1:H}^t) \sim \pi_{\mathsf{D}}$. We will show that—when combined with a test-time training-like update that performs token-level gradient updates *during test time*—the updates in Eq. (12) can circumvent the $H$-dependence in the lower bound of Proposition 5.1.

Concretely, we consider a distribution $\pi_\theta^{\mathsf{TTT}} : \mathcal{X} \to \Delta(\mathcal{Y}^H)$ formally introduced in Appendix K.6, which can be interpreted as an augmented version of the autoregressive linear model $\pi_\theta$ that uses test-time training to sample. Given a prompt $x$, we first sample $y_1 \sim \pi_\theta(\cdot \mid x)$, then perform a gradient step $\theta' \leftarrow \mathrm{Proj}_\Theta(\theta + \eta\nabla\log\pi_\theta(y_1 \mid x))$ to increase the probability of the token we just sampled. We then sample $y_2 \sim \pi_{\theta'}(\cdot \mid x, y_1)$, update $\theta'' \leftarrow \mathrm{Proj}_\Theta(\theta' + \eta\nabla\log\pi_{\theta'}(y_2 \mid x, y_1))$, and so on. Once the full sequence $y_{1:H}$ is sampled, we reset back to $\theta$ (so that we can process the next test-time example). This bears similarity to many test-time training methods in the literature, and specifically coincides with the method used in Krause et al. (2019); Rannen-Triki et al. (2024). We show that when augmented with this test-time sampling scheme, token-level SGD achieves a horizon-independent coverage bound that matches and even slightly improves upon the bound for next-token prediction in Theorem 4.2.

**Theorem 6.1** (Token-level SGD with test-time training). *Suppose Assumption 2.1 and Assumption 2.2 hold. For a suitably chosen parameter $\eta > 0$, token-level SGD (12) achieves*
$$\mathbb{E}\big[\tfrac{1}{T}\sum_{t=1}^T D_{\mathsf{KL}}(\pi_{\mathsf{D}} \,\|\, \pi_{\theta^t}^{\mathsf{TTT}})\big] \lesssim \sqrt{\tfrac{\sigma_\star^2}{T} + \tfrac{B^2}{T}}, \text{ and thus } \mathbb{E}\big[\tfrac{1}{T}\sum_{t=1}^T \mathsf{Cov}_N(\pi_{\theta^t}^{\mathsf{TTT}})\big] \lesssim \tfrac{1}{\log N}\big(\sqrt{\tfrac{\sigma_\star^2}{T} + \tfrac{B^2}{T}}\big).$$

This improves Theorem 4.2 by a factor of $1/\sqrt{\log N}$ on the leading term and a factor of $1/\log N$ on the second term. Furthermore, the algorithm bypasses the lower bound on KL divergence for *proper* methods in Proposition 3.2, demonstrating a provable benefit of being *improper*.

## 6.2 SELECTING FOR COVERAGE

We last consider the problem of selecting a model (e.g., checkpoint) from a small number of candidates to achieve the best coverage. We introduce a tournament-like procedure that improves upon maximum likelihood in that it removes the requirement that $\pi_{\mathsf{D}} \in \Pi$; it is guaranteed to find a model in the class with good coverage if one exists, even if $\pi_{\mathsf{D}}$ itself is not in the class. As an algorithmic intervention, we envision using this procedure to select a single training checkpoint or hyperparameter configuration to use for RL fine-tuning or test-time scaling. Indeed, as demonstrated in Figure 1, using cross-entropy as a selection criterion—as is standard—may result in poor coverage, while these procedures can select better checkpoints. Our results here concern the general setting in Section 2, and are not restricted to autoregressive linear models.

While their main motivation is model/checkpoint selection with a finite class $\Pi$, both estimators can also be applied to general, infinite classes $\Pi$. In this case, they improve upon the coverage achieved by the maximum likelihood estimator in Theorem 4.1, even in the well-specified case where $\pi_{\mathsf{D}} \in \Pi$; informally, the tournament estimators allow us to remove the fine-grained term in Theorem 4.1, leaving only a coarse-grained term.

**A simple tournament for maximizing coverage.** Given a dataset $\mathcal{D} = \{(x^i, y^i)\}_{i\in[n]}$, define

$$\widehat{\mathsf{Cov}}_N(\pi' \,\|\, \pi) \coloneqq \tfrac{1}{n}\Big|\Big\{i \in [n] : \tfrac{\pi'(y^i|x^i)}{\pi(y^i|x^i)} \geq N\Big\}\Big|, \tag{13}$$

which can be interpreted as an empirical version of the coverage profile $\mathsf{Cov}_N(\pi' \,\|\, \pi)$ in Eq. (1) when $\pi' = \pi_{\mathsf{D}}$ (see Lemma J.2). For $N \geq 1$, we consider the estimator

$$\widehat{\pi} \coloneqq \arg\min_{\pi \in \Pi} \max_{\pi' \in \Pi} \widehat{\mathsf{Cov}}_N(\pi' \,\|\, \pi). \tag{14}$$

Informally, this estimator chooses the model $\pi$ that minimizes the maximum coverage against any other model $\pi'$ in the class $\Pi$. When $\Pi$ is small, we can implement this tournament by simply evaluating the empirical coverage in Eq. (13) for each pair. The main guarantee for this estimator is as follows.

**Theorem 6.2.** *Let $N \geq 1$ be given. Then, for any $a \in [0, 1]$, with probability at least $1 - \delta$, the tournament estimator (14) achieves*

$$\mathsf{Cov}_{N^{1+a}}(\widehat{\pi}) \lesssim \min_{\pi \in \Pi} \mathsf{Cov}_{N^a}(\pi) + \tfrac{1}{N^{1-a}} + \tfrac{\log(|\Pi|/\delta)}{n}. \tag{15}$$

This shows that the tournament achieves a coverage profile nearly as good as the best-in-class, except for a small polynomial blow up, in that we bound the coverage at level $N^{1+a}$ in terms of the coverage for the best-in-class at level $N^a$.

**Infinite class and improving the tournament.** Eq. (14) can also be applied to general, infinite classes $\Pi$. In this case, it turns out that it improves upon the coverage achieved by the maximum likelihood estimator in Theorem 4.1 (see Theorem 6.2′). Furthermore, in Appendix G.5, we describe an improved tournament estimator that is able to remove the $1/N^{1-a}$ term from Theorem 6.2, thereby achieving nontrivial guarantees even when the coverage parameter $N$ is constant.

## DISCUSSION AND FUTURE WORK

See Appendix A for discussion and open problems, and Appendix G for additional results.

## ACKNOWLEDGEMENTS

We thank Clayton Sanford, Matus Telgarsky, and Nati Srebro for helpful discussions. FC acknowledges support from ARO through award W911NF-21-1-0328, Simons Foundation and the NSF through awards DMS-2031883 and PHY-2019786, and DARPA AIQ award.

## Reproducibility Statement

We provide full proofs for all theoretical results in the appendix. Appendix D includes extensive experiment setup and implementation details for all empirical results. The source code is included in the supplementary material, along with the plotting scripts and data to reproduce Figure 1 and Figure 2.

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

CONTENTS OF APPENDIX

# Part I

# Additional Discussion and Results

## A  DISCUSSION AND FUTURE WORK

Our work, through the lens of coverage, takes a first step toward clarifying the mechanisms through which pre-training with next-token prediction leads to models for which post-training is effective.

### A.1  SIMPLIFICATIONS IN THE PROBLEM FORMULATION

In the course of the paper we have made various simplifying assumptions. Some of these can be relaxed in a straightforward fashion, while others are more fundamental.

- In language model pre-training, the pre-training corpus consists of sequences $y$ with varying lengths $H$, and does not typically split examples into prompts and responses. Our formulation in Section 2 is a simplification (one that is closer in spirit to supervised fine-tuning), but we expect that the insights derived here can extend to the general setting.

- Much of our analysis focuses on the realizable/well-specified setting where $\pi_{\mathrm{D}} \in \Pi$. We give evidence in Appendix G that the coverage profile is more tolerant to misspecification than KL-divergence, but we leave a deeper investigation for future work.

- Our treatment assumes the distribution over prompts $\mu$ is the same for pre-training and post-training. This is straightforward to relax at the cost of introducing an additional coverage or distribution shift coefficient to handle the mismatch between the two distributions.

- We show that a good coverage profile is necessary for BoN to succeed on downstream tasks. While there is ample evidence current RL techniques can fail in the absence of coverage (Yue et al., 2025; Gandhi et al., 2025; Wu et al., 2025), it is not clear what the *minimal* conditions required for RL are.

- Our results focus on coverage at the *sequence level*. For reasoning tasks, it is natural to explicitly factorize the response $y = (y_{\mathsf{cot}}, y_{\mathsf{ans}})$ into a chain-of-thought (reasoning trajectory) component $y_{\mathsf{cot}}$ and an answer component $y_{\mathsf{ans}}$. For this setting, a weaker notion coverage is the following *answer-level coverage profile*: $\mathsf{Cov}_N^{\mathsf{ans}}(\pi_{\mathrm{D}} \parallel \widehat{\pi}) := \mathbb{P}_{\pi_{\mathrm{D}}}\left[\frac{\pi_{\mathrm{D}}(y_{\mathsf{ans}}|x)}{\widehat{\pi}(y_{\mathsf{ans}}|x)} \geq N\right]$. The answer-level coverage profile is sufficient for downstream BoN success for tasks where it is only important to produce the right answer, not a correct reasoning trace. We have $\mathsf{Cov}_N^{\mathsf{ans}}(\pi_{\mathrm{D}} \parallel \widehat{\pi}) \leq \mathsf{Cov}_N(\pi_{\mathrm{D}} \parallel \widehat{\pi})$, but the former can be strictly smaller in general.

### A.2  FUTURE WORK

Our results open several new directions for future research.

**Interventions for coverage.**  There is much to be done in understanding and improving existing algorithms such as optimizers through the lens of coverage. Our results in Section 6 show initial promise for using coverage to guide design of optimizers and model selection schemes, but the algorithm design space remains opaque, and there may be significant room for futher improvement. More ambitiously, one could imagine re-structuring the entire language modeling pipeline itself around coverage.

**Semantic coverage.**  The notion of coverage we focus on, the *coverage profile*, is mathematically convenient but may be conservative in regard to downstream performance, since it only depends on the model through its predicted probabilities. An important direction for future work is to understand pre-training and post-training through fine-grained "semantic" notions of coverage that more explicitly account for the representations learned by next-token prediction.

## B  RELATED WORK

**Related empirical observations.**  On the empirical side, our results are connected to a line of work that studies scaling laws for zero-shot downstream performance based on pre-training metrics such as cross-entropy (Gadre et al., 2024; Huang et al., 2024; Chen et al., 2024b; Sardana et al.,

2024). Several empirical works have also investigated how specific capabilities scale with additional pre-training, including machine translation (Ghorbani et al., 2022), knowledge capacity and memorization (Allen-Zhu & Li, 2025; Lu et al., 2024), and multi-hop reasoning (Wang et al., 2025). Our findings are consistent with Liu et al. (2022); Zeng et al. (2025); Lourie et al. (2025); Springer et al. (2025), who observe that cross-entropy is not always sufficient for predicting downstream performance, and in some cases can be anti-correlated.

Perhaps most closely related, Chen et al. (2025) show empirically that decreasing cross-entropy in pre-training does not necessarily lead to better Pass@N performance, and that Pass@N can even degrade as pre-training proceeds—a finding similar to Figure 1.[5] Our results can be viewed as placing their findings on stronger theoretical footing; conversely, their empirical results provide strong motivation for our theoretical treatment. Chen et al. (2025) also study a modification to the maximum likelihood objective aimed at improving coverage (in the spirit of Section 6); their approach targets the structure of outcome-based reward, whereas our notion of coverage profile and results are agnostic to the downstream task/reward structure.

We mention in passing some additional works. Chu et al. (2025) explored the different (synergistic) roles that supervised fine-tuning (SFT) and RL play in language model development, and subsequent work observed that the best checkpoint to start RL from can sometimes be in the middle of SFT training (Jin et al., 2025). Bansal et al. (2025) empirically identified the coverage of teacher-generated synthetic data as an important indicator for how effective distillation can be for reasoning tasks. Several papers have also investigated empirical tradeoffs between model size and reasoning performance under best-of-N sampling (Snell et al., 2025; Brown et al., 2025).

**Coverage in post-training.** Coverage metrics similar to coverage profile play a central role in theoretical literature on post-training and test-time algorithms (Huang et al., 2025a;b;c; Foster et al., 2025; Liu et al., 2024; Song et al., 2024; Gao et al., 2024; Liu et al., 2024; Ji et al., 2024), which analyze algorithms under the assumption that the base model has good coverage; our work can be viewed as providing theoretical motivation for this assumption. Formally, one can use Markov's inequality to bound the coverage profile by the $L_p$-like coverage quantities considered in these works.

Various notions of coverage similar to coverage profile have also appeared in the more classical literature on offline reinforcement learning (Farahmand et al., 2010; Chen & Jiang, 2019; Xie & Jiang, 2020; Jin et al., 2021; Foster et al., 2022; Jiang & Xie, 2024); here coverage is typically used to quantify the quality of an offline dataset rather than a model/policy itself.

**Generalization in deep learning.** Understanding the generalization behavior of deep learning models has been a central focus of the theory community for the last decade (Neyshabur et al., 2015; Zhang et al., 2017; Bartlett et al., 2017; Jacot et al., 2018; Belkin et al., 2019; Nagarajan & Kolter, 2019; Bartlett et al., 2020; Bartlett & Montanari, 2021). Our approach is somewhat complementary, in the sense that it focuses on the specific objective of next-token prediction with the logarithmic loss, and aims to understand when minimizing this loss leads to generalization for an *alternative* objective, coverage profile. We expect that our techniques can be combined with these contemporary generalization results to provide a more refined understanding of generalization for the coverage profile with deep models.

From this line of work, perhaps most closely related are Lotfi et al. (2023; 2024); Finzi et al. (2025), which aim to provide non-vacuous generalization bounds for the cross-entropy loss itself for autoregressive models.

**Analysis of maximum likelihood.** Our theoretical results are closely related to a classical line of work in statistics (Wong & Shen, 1995; van de Geer, 2000; Zhang, 2006), which shows that maximum likelihood can converge to the true model in Hellinger distance (or other Renyi divergences) under minimal assumptions, even when KL divergence is poorly behaved (large or infinite); see Appendix C below for a detailed comparison. Our results in Section 4 are similar in spirit, but provide a more fine-grained perspective, showing that the coverage profile can converge even faster than these results might suggest, particularly as one ventures further into the tail. Our analysis has some conceptual similarity to the small ball method of Mendelson (2014; 2017), which we elaborate on in Appendix J.1.

---

[5]Note that Chen et al. (2025) also uses the term "coverage", but as a synonym for Pass@N; this is not specifically related to the notion of the coverage profile we consider here.

Our techniques are also related to recent work of Foster et al. (2024); Rohatgi et al. (2025), which specializes the general techniques above to autoregressive models (e.g., under Hellinger distance).

## C  COMPARISON TO CLASSICAL GENERALIZATION BOUNDS FOR MLE

In this section we briefly compare our main coverage-based generalization bound for maximum likelihood to classical generalization bounds for maximum likelihood based on Hellinger distance and KL-divergence.

**Comparison to KL concentration.**  For general model classes $\Pi$, the best non-asymptotic KL-based generalization bound we are aware of is Proposition F.9 (Appendix F), which scales as roughly

$$D_{\mathsf{KL}}(\pi_{\mathsf{D}} \,\|\, \widehat{\pi}) \lesssim \log W_{\mathsf{max}} \cdot \mathcal{C}_{\mathtt{fine}}(\Pi, n)$$

under the assumption that all $\pi \in \Pi$ obey a sequence-level density ratio bound $\left\| \frac{\pi_0}{\pi} \right\|_\infty \leq W_{\mathsf{max}}$. Note that for the autoregressive linear class, we have $\log W_{\mathsf{max}} = BH$, matching Proposition 3.2. Combining such a guarantee with Proposition 3.1 gives a coverage bound of roughly

$$\mathsf{Cov}_N(\widehat{\pi}) \lesssim \frac{\log W_{\mathsf{max}}}{\log N} \cdot \mathcal{C}_{\mathtt{fine}}(\Pi, n);$$

this is rather uninteresting since $\mathsf{Cov}_N(\widehat{\pi}) = 0$ for $N \geq W_{\mathsf{max}}$; in other words, we do not get a meaningful improvement as we scale $N$.

**Comparison to Hellinger concentration.**  The Hellinger distance is a standard metric of distribution estimation, defined via $D_{\mathsf{H}}^2(\mathbb{P}, \mathbb{Q}) = \frac{1}{2} \int (\sqrt{\mathbb{P}} - \sqrt{\mathbb{Q}})^2$. The guarantees of maximum likelihood estimation (Wong & Shen, 1995; Van der Vaart, 2000; Zhang, 2006) also imply convergence in Hellinger distance. For general model classes $\Pi$, the best non-asymptotic Hellinger-based generalization bound we are aware of is Proposition F.8 (Appendix F), which scales as roughly

$$D_{\mathsf{H}}^2(\pi_{\mathsf{D}}, \widehat{\pi}) \lesssim \mathcal{C}_{\mathtt{fine}}(\Pi, n)$$

Combining such a guarantee with Proposition 3.1 gives a coverage bound of

$$\mathsf{Cov}_N(\widehat{\pi}) \lesssim \mathcal{C}_{\mathtt{fine}}(\Pi, n)$$

for all $N \geq 2$. Compare to the KL-based result above, this result gives a non-trivial bound on coverage when $N$ is constant (comparable to Theorem 4.1), but the issue is that it gives no further improvement as we scale $N$.

**Asymptotic bounds for maximum likelihood.**  We also note that the classical theory of maximum likelihood (e.g., Van der Vaart (2000)) provides *asymptotic* convergence rates for $d$-dimensional parametric classes $\Pi$ which have the following form:

$$D_{\mathsf{KL}}(\pi_{\mathsf{D}} \,\|\, \widehat{\pi}) \lesssim \frac{d}{n} \lesssim \mathcal{C}_{\mathtt{fine}}(\Pi, n), \qquad \text{as } n \to +\infty.$$

While this upper bound does *not* scale with $\log W_{\mathsf{max}}$, it can only be attained with $n \geq n_0$ for a sufficiently large burn-in cost $n_0$, which itself will typically scale with $\log W_{\mathsf{max}}$ or similar problem-dependent parameters; see, e.g., Spokoiny (2012) for non-asymptotic bounds of this type. Our lower bounds (e.g., Proposition 3.2) imply that there is no hope of removing such a burn-in cost in general.

## D    EXPERIMENTS

This section presents details for the experiments in Figure 1 and Figure 2. We describe the general graph search task used throughout our experiments in Appendix D.1, then detail the specific setups used for Figure 1 in Appendix D.2, and for Figure 2 in Appendix D.3.

### D.1    GRAPH REASONING TASK

We evaluate our theoretical predictions using experiments in graph reasoning tasks, in which transformer models are trained to find paths between source and target nodes in graphs. Both graph reasoning benchmarks and synthetic datasets have seen increasing use as abstractions for reasoning problems and for probing language modeling phenomena (Sanford et al., 2024; Nagarajan et al., 2025; Saparov et al., 2025; Bachmann & Nagarajan, 2024; Yehudai et al., 2025; Taylor et al., 2024; Wang et al., 2023; Fatemi et al., 2024; Tang et al., 2025). These tasks provide minimal abstractions of core reasoning problems, yet are expressive enough to capture pre-training and fine-tuning phenomena. They also offer flexibility in problem structure and difficulty: by specifying different graph topologies and path depths, we can modulate difficulty and expose sources of hardness.

#### D.1.1    GRAPH SEARCH TASK DESCRIPTION

The graph search tasks for all of our experiments in Appendix D.2 and Appendix D.3 share the same high-level components, and are comprised of

- **Problem instances.** A set of graph search problems $\mathcal{G}$ that map bijectively to a set of prompts $\mathcal{X}$.

- **Data distribution.** A distribution over the prompts $\mu \in \Delta(\mathcal{X})$. and a data collection policy $\pi_{\mathsf{D}} : \mathcal{X} \to \Delta(\mathcal{Y})$

- **Dataset.** The training dataset $\mathsf{D} = \{(x, y)\}$ is comprised of prompts $x \sim \mu$ and $y \sim \pi_{\mathsf{D}}(x)$.

Next, we describe the general details of the graph search task common to all experiments, as well as how the graph search task is converted to a sequence modeling problem for language models.

**Graph problem instances.**    Each graph search problem in $\mathbf{G} \in \mathcal{G}$ is specified by a tuple $\mathbf{G} = (G, s, t)$. Here, $G = (V, E)$ is a graph structure with nodes (or vertices) $V$ and edges $E = \{(u, v) : u, v \in V, u \neq v\}$, $s \in V$ is the source node, and $t$ is the target node. The nodes $V$ are represented as integers, so that $V \subset [m]$ for some fixed $m \in \mathbb{Z}$.

For all experiments, we utilize a *layered directed acyclic graph (layered DAG)* for each graph structure $(G, \_, \_) \in \mathcal{G}$, in which nodes are organized into sequential layers with edges flowing only from one layer to the next. The graph $G = (V, E)$ has $L + 2$ layers with disjoint sets of nodes, so that $V = \sqcup_{i \in \{1, \ldots, L+2\}} V^i$ where $V^i$ denotes the set of nodes in layer $i$. The first and last layers contain only the source and target nodes, respectively, so that $V^1 = \{s\}$ and $V^{L+2} = \{t\}$.

The edge structure $E$ connects only a subset of nodes in each layer to the next. We refer to this subset in each layer $i \in \{1, \ldots, L+2\}$ as its *passable nodes* $V_*^i \subseteq V^i$, or the set of nodes with non-zero out-degree,
$$V_*^i = \left\{ v \in V^i : \deg^+(v) > 0 \right\}.$$
The passable nodes in layer $i$ are fully connected to all nodes in the next layer, that is,
$$E = \left\{ (u, v) : u \in V_*^i, v \in V^{i+1}, i \in \{1, \ldots, L+1\} \right\}.$$
The remaining nodes in $V^i \setminus V_*^i$ have no outgoing edges, and are thus nodes the model must learn to avoid in order to output valid paths.

**Data distribution.**    The model's task is to imitate the data collection policy $\pi_{\mathsf{D}}$, which samples only a subset of the (potentially many) valid paths from source to target based on global features of the graph. A valid path from $s$ to $t$ is a list of nodes of the form $(s, v_2, \ldots, v_{L+1}, t)$ where $v_i \in V_*^i$ for each $i \in \{2, \ldots, L+1\}$; that is, the path must start with the source node $s$ and end with the target node $t$, and each intermediate node in the path must be a passable node from its respective layer. A graph may have many valid paths, specifically, $\prod_{i \in [L+2]} |V_*^i|$ many. In order for a model to learn valid paths, learning a simple local rule suffices: it can output any node in the next layer with $> 0$ out-degree, which is representable by a fairly shallow transformer.

However, imitating $\pi_{\mathsf{D}}$ is a much harder problem. The data collection policy $\pi_{\mathsf{D}}$ samples a subset of these valid paths determined via *global rules*, or complex functions computed over features of

the entire graph that go beyond those required for path validity alone. By varying the complexity of these rules, we can modulate both the difficulty and the nature of the learning problem. This structure naturally maps onto reasoning tasks: following passable nodes corresponds to taking "reasoning steps" that make progress towards the solution, while selecting non-passable nodes corresponds to reasoning errors that lead to invalid solutions. Moreover, when $\pi_D$ selects among valid paths via such global rules, this corresponds to learning high-quality solutions that accurately reflect desired properties for the problem.

**Dataset.** Recall that the model learns to imitate $\pi_D$ from a dataset $D = \{(x, y)\}$, where each prompt $x$ corresponds to a graph search problem $\mathbf{G} = (G, s, t) \in \mathcal{G}$, and each response $y \sim \pi_D(\cdot \mid x)$ is an expert response, formatted as follows.

We convert a given graph search problem $\mathbf{G} = (G, s, t) \in \mathcal{G}$ with graph structure $G = (V, E)$ to a prompt $x$ by concatenating the edge list $E$, the source node $s$, and the target node $t$, formatted as

$$x: \ \texttt{u\_1 v\_1 | u\_2 v\_2 | . . . | u\_k v\_k / s t =}$$

where $(u_i, v_i) \in [m]^2$ are the vertices of the $i$-th edge in the edge set $E$. For formatting, the special character | separates two edges, the character / separates the adjacency list from the source and target nodes, while the character = marks the end of the prompt.

As an example, for edge set $E = \{(10, 23), (86, 47), \ldots, (45, 32)\}$, the prompt is

$$x: \ \texttt{10 23 | 86 47 | . . . | 45 32 / 10 45 =} \qquad .$$

Next, each response $y$ encodes the path from the source to the target node in $G$ as a sequence of nodes. That is, the response takes the form of a string

$$y: \ \texttt{v\_1 v\_2 v\_2 v\_3 . . . v\_H-1 v\_H}$$

where $v_i \in [m]$ is the $i$'th nodes in the path for each $i \in [H]$, and $v_1 = s$ while $v_H = t$. Here, the horizon $H$ corresponds to the path length in $\mathcal{G}$, and in the layered DAG we have $H = L + 2$.

**Summary: Graph search to sequence modeling problem.** In summary, a graph search task with set of problem instances $\mathcal{G}$ induces an autoregressive sequence modeling problem with a vocabulary space $\mathcal{V} = [m] \cup \{\texttt{|}, \texttt{/}, \texttt{=}\}$, prompts $\mathcal{X} \subseteq \mathcal{V}^*$ corresponding to search problems in a *layered DAG graph structure* with $L + 2$ layers, and responses $\mathcal{Y} \subseteq \mathcal{V}^H$ corresponding to paths with length $H = L + 2$. In addition, the task is equipped with $\mu \in \Delta(\mathcal{X})$ and $\pi_D : \mathcal{X} \to \Delta(\mathcal{Y})$ that is used to collect the training dataset $D = \{(x, y)\}$, where $x \sim \mu$ and $y \sim \pi_D(x)$.

### D.1.2 MODEL DETAILS

Next, we describe the common implementation details for the models we train to solve the graph search task.

**Tokenizer.** We use a numeral tokenizer, which is standard for graph reasoning tasks (Sanford et al., 2024; Bachmann & Nagarajan, 2024). Each node $v \in [m]$ is tokenized as its integer node value, and the special characters |, /, and = are tokenized as $m + 1, m + 2, m + 3$, respectively.

**Transformer model.** We train causally-masked GPT2-like transformer models to minimize the cross-entropy loss using the Adam optimizer with fixed learning rate, and perform a grid search over the parameters displayed in Table 1. Parameters with fixed values were chosen based on related papers such as Bachmann & Nagarajan (2024). In both experiments, the model architecture with 4 heads, 6 hidden layers, and 384 hidden dimensions worked best. We use absolute positional encodings. Training iterations and grid search values for the learning rate are different for each experiment, and discussed further below.

### D.2 EXPERIMENT DETAILS FOR FIGURE 1

The graph search task for Figure 1 exposes natural properties of pre-training data under which cross-entropy reduction comes at the cost of a worse coverage profile. The key idea is that because the pre-training data is diverse (with multiple distinct modes or graph classes), the model is unable to perfectly fit the distribution. As a result, when one mode of behavior is better-represented than another, cross-entropy minimization, which is an average-case distribution-matching metric, can sacrifice coverage across the different modes in order to increase performance on a single mode.

| Hyperparameter | Values |
|---|---|
| Number of heads | {4, 6, 8} |
| Number of layers | {3, 4, 6, 8} |
| Hidden dimensions | 384 |
| Activation function | GeLU |
| Batch size | 128 |
| Weight decay | 0.01 |

Table 1: Hyperparameter grid search values for transformer models in graph search.

Concretely, the graph search task for Figure 1 is a mixture of two classes of graph structures. Due to representational and finite-sample constraints, the model is unable to fit both perfectly during training, and, in particular, fitting one class well (in the sense of cross-entropy loss) comes at the cost of worse performance on the other. The checkpoint with the best coverage arises at some middle point in training when the model learns both classes of graphs equally well, and has good coverage over both classes (the dip $\mathrm{Cov}_N$ in the leftmost subplot of Figure 1). Further reduction of cross-entropy loss over the latter half of training requires the model to lose coverage over $\pi_D$ in the less-represented graph class (observed as the increase in $\mathrm{Cov}_N$ in the latter half of training iterations).

Even though the task cannot be learned perfectly from the supervised learning feedback, the model can still learn a policy that always samples a correct path matching $\pi_D$'s with $N = O(1)$ Best-of-N sampling attempts, which means that it leads to efficient downstream post-training (e.g., on one of the modes or with reward-based feedback), and also achieves optimal performance with test-time scaling methods.

For the experiments in Figure 1, we first pre-train a model on a larger set of graph structure classes so that it learns a diverse set of behaviors, then finetune its behavior on two. The performance on the fine-tuning task is displayed in Figure 1, and we first describe the fine-tuning dataset, followed by the pre-training dataset.

### D.2.1 TASK DESCRIPTION

All graphs in $\mathcal{G}$ follow the *layered DAG* structure described in Appendix D.1 with $L = 8$ intermediate layers that each have 4 nodes, i.e., $|V^i| = 4$ for layers $i \in \{2, \ldots, 9\}$ (recall the first and last layers contain only $s$ and $t$, respectively).

Recall that in a layer $i$, $V^i_* = \{v \in V^i : \deg^+(v) > 0\}$ denotes the set of *passable nodes*. For each graph problem $\mathbf{G} = (G, s, t) \in \mathcal{G}$ with graph structure $G = (V, E)$, a subset of the layers indexed by $I_2 \subset \{2, \ldots, 9\}$ with $|I_2| = 2$ is randomly selected. Then, the edges $E$ are defined so that the layers in $I_2$ have two passable nodes each (i.e., $|V^i_*| = 2$ for $i \in I_2$), while the remaining layers have only one passable node each (i.e., $|V^i_*| = 1$ for $i \in \{2, \ldots, 9\} \setminus I_2$). The passable nodes in each layer are chosen at random, but for the layers in $I_2$ are guaranteed to have one even and one odd node. For each graph in $\mathcal{G}$, there are $2^2 = 4$ total valid paths since $|I_2| = 2$ layers have two passable nodes each while the other layers have one.

**Data distribution.** The set of problem instances $\mathcal{G} = \mathcal{G}_1 \sqcup \mathcal{G}_2$ is comprised of two disjoint classes of problems, $\mathcal{G}_1$ and $\mathcal{G}_2$. The prompt distribution in the fine-tuning task is a skewed mixture over the two classes with $\widetilde{\mu} \in \Delta(\{1, 2\})$ denoting the probability of each class in the data; within each class, the graphs are drawn uniformly at random (described at the end of this section). Although there are 4 valid paths from source to target, in each class $\mathcal{G}_1$ or $\mathcal{G}_2$ the policy $\pi_D$ chooses one path based on a different global rule, described below.

**Class $\mathcal{G}_1$ (probability $\widetilde{\mu}(1) = 0.9$).** For an integer $j \in \mathbb{Z}$, let the function $p(j) = (j \mod 2)$ denote its parity. For layers $i$ with $|V^i_*| = 1$, $\pi_D$ deterministically selects the unique passable node. For layers $i \in I_2$ (where $|V^i_*| = 2$), the set $V^i_*$ contains one even and one odd node, and $\pi_D$ deterministically chooses the node $v \in V^i_*$ such that $p(v) = p(i)$; that is, the node whose parity matches the parity of the layer index.

**Class $\mathcal{G}_2$ (probability $\widetilde{\mu}(2) = 0.1$).** For layers $i$ with $|V_*^i| = 1$, $\pi_D$ deterministically selects the unique passable node. For layers $i \in I_2$ (where $|V_*^i| = 2$), $\pi_D$ chooses the node $v \in V_*^i$ such that $p(v) = 1 \oplus p(i)$; that is, the node whose parity is opposite to the parity of the layer index.

The class of a graph is technically identifiable from the prompt by computing a parity-based feature over a randomly selected subset of the nodes, but this problem is too difficult for the model to learn in the fine-tuning stage. Let $V' \subseteq V$ be a fixed subset of nodes whose cardinality is half the total number of nodes in the graph (i.e., $|V'| = |V|/2$). Then all graphs in $\mathcal{G}_1$ satisfy $1 = \bigoplus_{u \in V'} p(u)$, while all graphs in $\mathcal{G}_2$ satisfy $0 = \bigoplus_{u \in V'} p(u)$. However, determining which nodes belong to $V'$ requires complex reasoning over the graph structure.

**Dataset.** Each sample in the dataset $D = \{(x, y)\}$ is then generated via the following procedure.

1. First sample an index $i \sim \widetilde{\mu}$.

2. Sample $G \in \mathcal{G}_i$ by randomly drawing $V \subset [m]$ without replacement, and instantiate the edges according to the description for each class above.

3. Format the prompt $x$ per Appendix D.1.

4. Draw $y \sim \pi_D(\cdot \mid x)$ according to description for each class above.

### D.2.2 PRE-TRAINING DESCRIPTION

The graph problem instances in the pre-training task, $\mathcal{G}_{\text{pre}}$, are a superset of the graphs in the fine-tuning task, that is, $\cup_{i \in [K]} \mathcal{G}_i = \mathcal{G}_{\text{pre}}$ with $K = 3$, and $\mathcal{G}_1$ and $\mathcal{G}_2$ defined as in the previous section for the finetuning dataset. The data distribution is a uniform mixture of these 3 classes, $\widetilde{\mu}(i) = \frac{1}{K}$ for each $i \in [K]$, and the third class $\mathcal{G}_3$ shares the same layered DAG structure as $\mathcal{G}_1$ and $\mathcal{G}_2$ (with $L = 8$ intermediate layers, where two layers are randomly chosen to have multiple passable nodes). However, in $\mathcal{G}_3$, $\pi_D$ is a stochastic policy and samples one of the $2^2 = 4$ valid paths at random. The dataset is then drawn using the same data generation procedure described for the fine-tuning task above.

### D.2.3 TASK-SPECIFIC IMPLEMENTATION DETAILS

The transformer model is first pre-trained on a fixed dataset drawn from the pre-training distribution, with $8 \times 64,000$ prompts in total, using a learning rate of $1e{-}4$ for 200k iterations, which was chosen based on a grid search over learning rates $\{5e{-}5, 1e{-}4, 5e{-}4\}$.

The final checkpoint is then finetuned for 50k iterations in an online fashion, where fresh samples are drawn for each batch (this is equivalent to offline training with a dataset that has an equivalent number of samples). The learning rate is $5e{-}6$, which was chosen based on a grid search over learning rates $\{5e{-}6, 1e{-}5\}$.

### D.3 EXPERIMENT DETAILS FOR FIGURE 2

For Figure 2, we consider a family of tasks that is parameterized by the horizon $H$, in order to expose the fact that cross-entropy is sensitive to horizon, but the coverage profile is not. This construction leverages the intuition from Remark 3.1. The training data is heterogeneous, with a fraction consisting of difficult graph problems that the model cannot learn to cover with the given number of training samples. This un-learnable subset of the data contributes to the large KL-divergence, but does not affect the coverage profile.

### D.3.1 TASK DESCRIPTION

For Figure 2, we devise a family of tasks parameterized by the number of intermediate layers $H \in \{8, 16, 24\}$. For a fixed $H$, each task $\mathcal{G}_H$ utilizes the *layered DAG* graph structure described in Appendix D.1 with $L = H$ intermediate layers, each containing 4 nodes, so that each graph has $H + 2$ total layers (including source and target). The response space is $\mathcal{Y} = \mathcal{V}^{H+2}$, corresponding to paths of length $H + 2$ (including the source and target nodes).

**Data distribution.** The task is a heterogeneous mixture over 3 classes of graphs described below that we refer to as $\mathcal{G}_{H,1} \cup \mathcal{G}_{H,2} \cup \mathcal{G}_{H,3} = \mathcal{G}_H$. The classes $\mathcal{G}_{H,2}$ and $\mathcal{G}_{H,3}$ are significantly harder to learn and the model will fail to do so with the given number of training samples, even though $\mathcal{G}_{H,1}$ is learned quickly (and also provides useful features for learning the other two tasks). The distribution over these 3 classes is fixed for all $H$ and specified by $\widetilde{\mu} \in \Delta(\{1, 2, 3\})$.

**Class $\mathcal{G}_{H,1}$ (probability $\widetilde{\mu}(1) = 0.94$).** All $H$ intermediate layers have only 1 passable node each (i.e., $|V_*^i| = 1$ for all $i \in \{2, \ldots, H+1\}$), so each $G \in \mathcal{G}_{H,1}$ has only one valid path from source to target. For prompts corresponding to graphs in this class, $\pi_\mathsf{D}$ deterministically selects the unique valid path.

**Class $\mathcal{G}_{H,2}$ (probability $\widetilde{\mu}(2) = 0.05$).** For each graph, half of the intermediate layers (or $H/2$) are randomly selected to have two passable nodes, while the rest have one. More formally, a subset $I_{H/2} \subset \{2, \ldots, H+1\}$ with $|I_{H/2}| = H/2$ is randomly selected, such that $|V_*^i| = 2$ for $i \in I_{H/2}$ and $|V_*^i| = 1$ for $i \in \{2, \ldots, H+1\} \setminus I_{H/2}$.

There are $2^{H/2}$ valid paths from source to target, and $\pi_\mathsf{D}$ deterministically selects one of them. For layers $i$ with $|V_*^i| = 1$, $\pi_\mathsf{D}$ selects the unique passable node. For layers $i \in I_{H/2}$ (where $|V_*^i| = 2$), $\pi_\mathsf{D}$ selects the node $v \in V_*^i$ by following a difficult, deterministic rule. This rule requires $\pi_\mathsf{D}$ to select the node $v$ whose parity matches the parity of the layer index, XOR'ed with the parity of each passable node in the entire graph. More specifically, recall that $p(j)$ denotes the parity of an integer $j \in [m]$, and let $V_* := \bigcup_{i=2}^{H+1} V_*^i$ denote the set of all passable nodes across all intermediate layers (including those with just one passable node). Then in layer $i \in I_{H/2}$, $\pi_\mathsf{D}$ selects the node $v \in V_*^i$ such that $p(v) = p(i) \oplus \left( \bigoplus_{u \in V_*} p(u) \right)$.

**Class $\mathcal{G}_{H,3}$ (probability $\widetilde{\mu}(3) = 0.01$).** Regardless of $H$, for each graph a subset $I_4 \subset \{2, \ldots, H+1\}$ with $|I_4| = 4$ is randomly selected, such that $|V_*^i| = 2$ for $i \in I_4$ and $|V_*^i| = 1$ for $i \in \{2, \ldots, H+1\} \setminus I_4$. There are $2^4 = 16$ valid paths from source to target. The policy $\pi_\mathsf{D}$ samples uniformly at random from these valid paths.

Note that prompts/graphs from each class are distinguishable from each other (or, identifiable) based on prompt features alone, so a powerful-enough model can achieve perfect performance across all of them simultaneously. $\mathcal{G}_{H,2}$, for example, has more edges and thus a longer prompt than $\mathcal{G}_{H,1}$; similar statements apply to $\mathcal{G}_{H,3}$. Dataset generation occurs in the same manner as described in Appendix D.2.

### D.3.2 TASK-SPECIFIC IMPLEMENTATION DETAILS

Lastly, we describe experiment-specific implementation details on top of those previously described in Appendix D.1, which are common to all experiments. In addition to a grid search over the parameters in Table 1, we perform a search over learning rates $\{5e{-}5, 1e{-}4, 5e{-}4\}$, for which the learning rate of $1e{-}4$ exhibited the best validation performance. The model is trained for 40k iterations over a fixed dataset of $8 \times 64{,}000$ samples.

The results in Figure 2 are computed from evaluations of training checkpoints on per-class validation datasets of 1024 prompts from each $\mathcal{G}_{H,i}$ for $i \in [3]$; these metrics are then averaged according to the probabilities in $\widetilde{\mu}$ to obtain the final result. In total we ran 16 seeds, and plot their median. The shaded region in Figure 2 displays the region between the $\frac{1}{16}$ quantile and $\frac{15}{16}$ quantile.

# E  PROPERTIES OF THE COVERAGE PROFILE

Before proceeding, we briefly discuss some properties of the coverage profile that will be helpful to keep in mind.

**Remark E.1** (Coverage profile as a refinement of cross-entropy). *The coverage profile can be viewed as a fine-grained, inference budget-sensitive* refinement *of cross-entropy. Concretely, if we write*

$$\mathsf{Cov}_N(\pi_{\mathsf{D}} \,\|\, \widehat{\pi}) = \mathbb{P}_{\pi_{\mathsf{D}}}\left[\log \frac{\pi_{\mathsf{D}}(y \mid x)}{\widehat{\pi}(y \mid x)} \geq \log N\right], \tag{16}$$

*it becomes clear that the coverage profile is simply the cumulative distribution function (CDF) of the log density ratio $X := \log \frac{\pi_{\mathsf{D}}(y|x)}{\widehat{\pi}(y|x)}$, while KL-divergence corresponds to the mean: $\mathbb{E}_{\pi_{\mathsf{D}}}[X]$. It is well known that the CDF of a random variable is a more informative statistic than its mean (Durrett, 2019); the former can be much more sensitive to the model's behavior at the tail than the latter. Indeed, the coverage profile can behave very differently across scales, as shown by Figure 1.*[6]

**Remark E.2** (KL divergence and coverage profile are not estimable). *We emphasize that KL-divergence and the coverage profile are not estimable quantities in general, due to the fact both depend on the unknown density $\pi_{\mathsf{D}}(y \mid x)$ for the data distribution. This motivates the use of cross-entropy in practice, as the former is an estimable upper bound on $D_{\mathsf{KL}}(\pi_{\mathsf{D}} \,\|\, \widehat{\pi})$. Analogously, we show in Section 6.2 that various estimable proxies for the coverage profile can be used to select models with good coverage. One exception is the* expert distillation *setting (see Appendix G.4), where $\pi_{\mathsf{D}}$ is a teacher network for which the log-probabilities $\log \pi_{\mathsf{D}}(y \mid x)$ are available.*

---

[6]Interestingly, we show (Proposition F.1) that if the coverage profile satisfies a certain growth condition uniformly for all scales $M$, then it implies a bound on KL-divergence—a weak converse to Proposition 3.1.

## F  SUPPORTING RESULTS

This section presents technical results used throughout the paper. Appendix F.1 presents basic properties of the coverage profile. Appendix F.2 analyzes the performance of the Best-of-N algorithm under coverage. Appendix F.3 presents properties of the maximum likelihood estimator, and Appendix F.4 presents structural results relating the coverage profile to a "stopped" KL-divergence, which are useful for analyzing autoregressive models.

### F.1  PROPERTIES OF THE COVERAGE PROFILE

This section presents elementary properties of the coverage profile.

**Proposition F.1** (KL-to-coverage conversion). *For all models $\pi_{\mathsf{D}}$ and $\pi$ and $M \geq 2$, we have*

$$\mathsf{Cov}_N(\pi) \leq \frac{D_{\mathsf{KL}}(\pi_{\mathsf{D}} \parallel \pi)}{\log N - 1 + \frac{1}{N}}.$$

**Proof of Proposition F.1.** Lemma 27 of Block & Polyanskiy (2023) states that for any $N > 1$ and any convex $f : [0, \infty] \to [0, \infty]$ with $f(1) = f'(1) = 0$,

$$\mathsf{Cov}_N(\pi) = \mathbb{P}_{\pi_{\mathsf{D}}} \left[ \frac{\pi_{\mathsf{D}}(y \mid x)}{\pi(y \mid x)} > N \right] \leq \frac{N D_f(\pi_{\mathsf{D}} \parallel \pi)}{f(N)}, \tag{17}$$

where $D_f(\pi_{\mathsf{D}} \parallel \pi) := \mathbb{E}_\pi \left[ f\left( \frac{d\pi_{\mathsf{D}}}{d\pi} \right) \right]$. Applying this with KL-divergence, which corresponds to $f(x) = x \log x - x + 1$ with $f'(x) = \log x$, we have that

$$\frac{N}{f(N)} = \frac{1}{\log N - 1 + 1/N}, \tag{18}$$

which gives the result.

$\square$

**Proposition F.2** (Tightness of KL-to-coverage conversion). *For any $N \geq 2$, there exist models $\pi_{\mathsf{D}}$ and $\widehat{\pi}$ such that*

$$\mathsf{Cov}_N(\widehat{\pi}) \geq \frac{D_{\mathsf{KL}}(\pi_{\mathsf{D}} \parallel \widehat{\pi})}{\log N - \frac{1}{2} + \frac{1}{2N}}.$$

**Proof of Proposition F.2.** Consider $\pi_{\mathsf{D}} = \mathrm{Ber}(p)$ and $\widehat{\pi} = \mathrm{Ber}(p/N)$ with $p \leq \frac{1}{2}$. Then $\mathsf{Cov}_N(\widehat{\pi}) = p$ and

$$
\begin{aligned}
D_{\mathsf{KL}}(\pi_{\mathsf{D}} \parallel \widehat{\pi}) &= p \log N + (1-p) \log \frac{1-p}{1 - \frac{p}{N}} \leq p \log N + (1-p) \left( \frac{1-p}{1 - \frac{p}{N}} - 1 \right) \\
&= p \left( \log N - (1-p) \frac{1 - \frac{1}{N}}{1 - \frac{p}{N}} \right) \\
&\leq p \cdot \left( \log N - \frac{1}{2} + \frac{1}{2N} \right).
\end{aligned}
$$

This is the desired result.

$\square$

**Proposition F.3** (Uniform coverage decay implies bounded KL). *Given $\pi, \pi_{\mathsf{D}} : \mathcal{X} \to \Delta(\mathcal{Y})$, define $W_{\mathsf{max}} := \sup_{x,y} \frac{\pi_{\mathsf{D}}(y|x)}{\pi(y|x)}$ and*

$$C := \sup_{N \geq 1} \{ \mathsf{Cov}_N(\pi) \cdot \log N \},$$

*where we note that $C \leq \log W_{\mathsf{max}}$. It holds that*

$$D_{\mathsf{KL}}(\pi_{\mathsf{D}} \parallel \pi) \leq C \cdot (1 + \log(\log(W_{\mathsf{max}})/C)). \tag{19}$$

**Proof of Proposition F.3.** Let $\delta > 0$ a fixed parameter, and define $X := \pi_\mathsf{D}/\pi$. Then we have

$$D_{\mathsf{KL}}(\pi_\mathsf{D} \,\|\, \pi) = \mathbb{E}_{\pi_\mathsf{D}}[\log(X)] \leq \mathbb{E}_{\pi_\mathsf{D}}[\log(X)\mathbb{I}\{\log(X) > \delta\}] + \delta. \tag{20}$$

Since $X \leq W_{\mathsf{max}}$ almost surely, we can write

$$\mathbb{E}_{\pi_\mathsf{D}}[\log(X)\mathbb{I}\{\log(X) > \delta\}] = \int_\delta^{\log(W_{\mathsf{max}})} \mathbb{P}_{\pi_\mathsf{D}}[\log(X) > t]\mathrm{d}t \tag{21}$$

$$= \int_\delta^{\log(W_{\mathsf{max}})} \mathbb{P}_{\pi_\mathsf{D}}\big[X > e^t\big]\mathrm{d}t \tag{22}$$

$$\leq C \int_\delta^{\log(W_{\mathsf{max}})} \frac{1}{t}\mathrm{d}t \tag{23}$$

$$= C \log\!\left(\frac{\log(W_{\mathsf{max}})}{\delta}\right). \tag{24}$$

The result now follows by setting $\delta = C$. $\qquad\square$

**Proposition F.4** (Hellinger-to-coverage conversion). *For all models $\pi_\mathsf{D}$ and $\pi$ and $N > 1$, we have*

$$\mathsf{Cov}_N(\pi_\mathsf{D} \,\|\, \pi) \leq \frac{2N}{(\sqrt{N}-1)^2} \cdot D_\mathsf{H}^2(\pi_\mathsf{D}, \pi).$$

**Proof of Proposition F.4.** Without loss of generality, we assume $\mathcal{Y}$ is discrete in the following proof. By definition,

$$D_\mathsf{H}^2(\pi_\mathsf{D}, \pi) = \frac{1}{2} \mathbb{E}_{x\sim\pi_\mathsf{D}}\left[\sum_y \left(\sqrt{\pi_\mathsf{D}(y \mid x)} - \sqrt{\pi(y \mid x)}\right)^2\right]$$

$$\geq \frac{1}{2} \mathbb{E}_{x\sim\pi_\mathsf{D}}\left[\sum_y \pi_\mathsf{D}(y \mid x)\left(1 - \frac{1}{\sqrt{N}}\right)^2 \mathbb{I}\left\{\pi(y \mid x) \leq \frac{1}{N}\pi_\mathsf{D}(y \mid x)\right\}\right]$$

$$= \frac{1}{2}\left(1 - \frac{1}{\sqrt{N}}\right)^2 \mathbb{P}_{\pi_\mathsf{D}}\left[\frac{\pi_\mathsf{D}(y \mid x)}{\pi(y \mid x)} > N\right],$$

where the inequality follows from the fact that $\sqrt{\pi_\mathsf{D}(y \mid x)} - \sqrt{\pi(y \mid x)} \geq \left(1 - \frac{1}{\sqrt{N}}\right)\sqrt{\pi_\mathsf{D}(y \mid x)}$ is implied by $\pi(y \mid x) \leq \frac{1}{N}\pi_\mathsf{D}(y \mid x)$. Re-organizing completes the proof. $\qquad\square$

**Proposition F.5** (Chain rule for coverage profile). *For any models $\pi_\mathsf{D}$, $\pi_\mathsf{T}$, and $\widehat{\pi}$, and any $M_1, M_2 \geq 2$, we have*

$$\mathsf{Cov}_{M_1}(\pi_\mathsf{T} \,\|\, \widehat{\pi}) \leq M_2 \cdot \mathsf{Cov}_{M_1/M_2}(\pi_\mathsf{D} \,\|\, \widehat{\pi}) + \mathsf{Cov}_{M_2}(\pi_\mathsf{T} \,\|\, \pi_\mathsf{D}). \tag{25}$$

**Proof of Proposition F.5.** We can write

$$\mathsf{Cov}_{M_1}(\pi_\mathsf{T} \,\|\, \widehat{\pi}) = \mathbb{P}_{\pi_\mathsf{T}}\left[\frac{\pi_\mathsf{T}(y \mid x)}{\widehat{\pi}(y \mid x)} > M_1\right]$$

$$= \mathbb{P}_{\pi_\mathsf{T}}\left[\frac{\pi_\mathsf{T}(y \mid x)}{\widehat{\pi}(y \mid x)} > M_1, \frac{\pi_\mathsf{T}(y \mid x)}{\pi_\mathsf{D}(y \mid x)} \leq M_2\right] + \mathbb{P}_{\pi_\mathsf{T}}\left[\frac{\pi_\mathsf{T}(y \mid x)}{\widehat{\pi}(y \mid x)} > M_1, \frac{\pi_\mathsf{T}(y \mid x)}{\pi_\mathsf{D}(y \mid x)} > M_2\right]$$

$$\leq M_2 \mathbb{P}_{\pi_\mathsf{D}}\left[\frac{\pi_\mathsf{D}(y \mid x)}{\widehat{\pi}(y \mid x)} > M_1/M_2\right] + \mathbb{P}_{\pi_\mathsf{T}}\left[\frac{\pi_\mathsf{T}(y \mid x)}{\pi_\mathsf{D}(y \mid x)} > M_2\right]$$

$$= M_2 \mathsf{Cov}_{M_1/M_2}(\pi_\mathsf{D} \,\|\, \widehat{\pi}) + \mathsf{Cov}_{M_2}(\pi_\mathsf{T} \,\|\, \pi_\mathsf{D}).$$

$$\square$$

## F.2 ANALYSIS OF BEST-OF-N SAMPLING UNDER A GOOD COVERAGE PROFILE

In this section we analyze the performance of the Best-of-N algorithm under a good coverage profile. Let a base model $\widehat{\pi}$ be given, and let a reward function $r_{\mathsf{T}}(x, y) \in [0, 1]$ be given. Let $\pi_{\mathsf{T}} : \mathcal{X} \to \Delta(\mathcal{Y})$ denote an arbitrary task-specific comparator policy.

We let $\widehat{\pi}_N^{\mathsf{BoN}}(x)$ denote the distribution of the Best-of-N algorithm with parameter $N$, which draws $N$ responses $y^1, \ldots, y^N \overset{\text{i.i.d.}}{\sim} \widehat{\pi}(\cdot \mid x)$ and returns $y = \arg\max_{y_i} r_{\mathsf{T}}(x, y_i)$.

**Proposition F.6** (Coverage implies success for BoN). *Let $M \geq 1$ be given. For any $\varepsilon > 0$, if $N \geq 2M \log(\varepsilon^{-1})$ and $\mathsf{Cov}_M(\pi_{\mathsf{T}} \parallel \widehat{\pi}) \leq \frac{1}{2}$, then we are guaranteed that*

$$\mathbb{E}_{x \sim \mu}\big[r_{\mathsf{T}}(x, \pi_{\mathsf{T}}(x)) - r_{\mathsf{T}}(x, \widehat{\pi}_N^{\mathsf{BoN}}(x))\big] \leq \mathsf{Cov}_M(\pi_{\mathsf{T}} \parallel \widehat{\pi}) + \varepsilon. \tag{26}$$

**Proof of Proposition F.6.** This is an immediate consequence of Lemma F.1 in Huang et al. (2025b), noting that we can bound $\mathcal{E}_M(\pi_{\mathsf{T}} \parallel \widehat{\pi}) \leq \mathsf{Cov}_M(\pi_{\mathsf{T}} \parallel \widehat{\pi})$. $\qquad\square$

**Proposition F.7** (Coverage is necessary for BoN). *For any model $\widehat{\pi}$ and reference $\pi_{\mathsf{T}}$, and for any $N \geq 2$, there exists a reward function $r_{\mathsf{T}}(x, y) \in \{0, 1\}$ such that*

$$\mathbb{E}_{x \sim \mu}\big[r_{\mathsf{T}}(x, \pi_{\mathsf{T}}(x)) - r_{\mathsf{T}}(x, \widehat{\pi}_N^{\mathsf{BoN}}(x))\big] \geq \frac{1}{2}\mathsf{Cov}_{2N}(\pi_{\mathsf{T}} \parallel \widehat{\pi}). \tag{27}$$

**Proof of Proposition F.7.** For any $x \in \mathcal{X}$, we define $S_x := \big\{y \in \mathcal{Y} : \frac{\pi_{\mathsf{T}}(y|x)}{\widehat{\pi}(y|x)} \geq 2N\big\}$ and let $r_{\mathsf{T}}(x, y) = \mathbb{I}\{y \in S_x\}$.

By definition, for any fixed $x \in \mathcal{X}$, it holds that

$$
\begin{aligned}
r_{\mathsf{T}}(x, \widehat{\pi}_N^{\mathsf{BoN}}(x)) = \mathbb{P}_{y \sim \widehat{\pi}_N^{\mathsf{BoN}}(x)}(y \in S_x) &= \mathbb{P}_{y^1, \ldots, y^N \overset{\text{i.i.d.}}{\sim} \widehat{\pi}(\cdot|x)}(\exists i \in [N], y^i \in S_x) \\
&= 1 - \big(1 - \mathbb{P}_{y \sim \widehat{\pi}(\cdot|x)}(y \in S_x)\big)^N \leq N \cdot \mathbb{P}_{y \sim \widehat{\pi}(\cdot|x)}(y \in S_x) \\
&= N \cdot \sum_{y \in S_x} \widehat{\pi}(y \mid x) \leq N \cdot \sum_{y \in S_x} \frac{1}{2N}\pi_{\mathsf{T}}(y \mid x) = \frac{1}{2}\mathbb{P}_{y \sim \pi_{\mathsf{T}}(\cdot|x)}(S_x),
\end{aligned}
$$

where we use the fact that $\widehat{\pi}(y \mid x) \leq \frac{1}{2N}\pi_{\mathsf{T}}(y \mid x)$ for any $y \in S_x$. We also note that $\mathbb{P}_{x \sim \mu, y \sim \pi_{\mathsf{T}}(\cdot|x)}(y \in S_x) = \mathsf{Cov}_{2N}(\pi_{\mathsf{T}} \parallel \widehat{\pi})$. Therefore,

$$\mathbb{E}_{x \sim \mu}\big[r_{\mathsf{T}}(x, \pi_{\mathsf{T}}(x)) - r_{\mathsf{T}}(x, \widehat{\pi}_N^{\mathsf{BoN}}(x))\big] \geq \frac{1}{2}\mathsf{Cov}_{2N}(\pi_{\mathsf{T}} \parallel \widehat{\pi}).$$

$\qquad\square$

## F.3 PROPERTIES OF MAXIMUM LIKELIHOOD

In this section, we specialize standard guarantees for maximum likelihood (Wong & Shen, 1995; van de Geer, 2000; Zhang, 2006) to derive bounds on the coverage profile; as discussed in Appendix C, these results are not tight compared to Theorem 4.1.

**Proposition F.8** (Convergence of maximum likelihood in Hellinger distance). *Assume that $\pi_{\mathsf{D}} \in \Pi$. With probability at least $1 - \delta$, the maximum likelihood estimator $\widehat{\pi} := \arg\max_{\pi \in \Pi} \widehat{L}_n(\pi)$ satisfies,*

$$D_{\mathsf{H}}^2(\pi_{\mathsf{D}}, \widehat{\pi}) \lesssim \inf_{\varepsilon > 0}\bigg\{\frac{\log \mathcal{N}_\infty(\Pi, \varepsilon)}{n} + \varepsilon\bigg\}, \tag{28}$$

*and consequently*

$$\mathsf{Cov}_M(\widehat{\pi}) \lesssim \inf_{\varepsilon > 0}\bigg\{\frac{\log \mathcal{N}_\infty(\Pi, \varepsilon)}{n} + \varepsilon\bigg\}. \tag{29}$$

*for all $M \geq 2$.*

**Proof of Proposition F.8.** The first bound follows from Proposition B.2 of Foster et al. (2024). The second bound follows from applying Proposition F.4. $\qquad\square$

**Proposition F.9** (Convergence of maximum likelihood in KL). *Assume that $\pi_{\mathsf{D}} \in \Pi$, and that all $\pi \in \Pi$ satisfy $\left\| \frac{\pi_{\mathsf{D}}}{\pi} \right\|_\infty \leq W_{\mathsf{max}}$. With probability at least $1 - \delta$, the maximum likelihood estimator $\widehat{\pi} := \arg\max_{\pi \in \Pi} \widehat{L}_n(\pi)$ satisfies,*

$$D_{\mathsf{KL}}(\pi_{\mathsf{D}} \,\|\, \widehat{\pi}) \lesssim \log W_{\mathsf{max}} \cdot \inf_{\varepsilon > 0} \left\{ \frac{\log \mathcal{N}_\infty(\Pi, \varepsilon)}{n} + \varepsilon \right\}, \tag{30}$$

*and consequently*

$$\mathsf{Cov}_M(\widehat{\pi}) \lesssim \frac{\log W_{\mathsf{max}}}{\log M} \cdot \inf_{\varepsilon > 0} \left\{ \frac{\log \mathcal{N}_\infty(\Pi, \varepsilon)}{n} + \varepsilon \right\}, \tag{31}$$

*for all $M \geq 2$.*

We remark that the $\log(W_{\mathsf{max}})$-factor in Eq. (30) can be tight in general. For example, for the class $\Pi$ considered in Proposition 3.2, it holds that $\log \mathcal{N}_\infty(\Pi, \varepsilon) \lesssim \log(1/\varepsilon) \vee 1$ and $\left\| \frac{\pi_{\mathsf{D}}}{\pi} \right\|_\infty \leq e^{2H}$.

**Proof of Proposition F.9.** By Lemma 4 of Yang & Barron (1998), it holds that

$$D_{\mathsf{KL}}(\pi_{\mathsf{D}} \,\|\, \widehat{\pi}) \leq (2 + \log(W_{\mathsf{max}})) D_{\mathsf{H}}^2(\pi_{\mathsf{D}}, \widehat{\pi}).$$

Therefore, the first bound then follows from Eq. (28). The second bound follows from applying Proposition F.1. $\qquad\square$

### F.4 Autoregressive Models: Coverage and Stopped KL-Divergence

This section shows that we can relate the coverage profile to a "stopped" KL-divergence defined in Eq. (32). This is a useful result in the context of autoregressive models because the stopped KL-divergence is always bounded, even when KL-divergence itself may not be.

**Proposition F.10.** *Define the stopped KL-divergence for parameter $N$ as*

$$D_{\mathsf{seq},N}(\pi_{\mathsf{D}} \,\|\, \pi) = \mathbb{E}_{(x, y_{1:H}) \sim \pi_{\mathsf{D}}} \left[ \min\left\{ \log N, \sum_{h=1}^H D_{\mathsf{KL}}(\pi_{\mathsf{D}}(\cdot \mid x, y_{1:h-1}) \,\|\, \pi(\cdot \mid x, y_{1:h-1})) \right\} \right]. \tag{32}$$

*Then as long as $N > e$, it holds that*

$$\mathsf{Cov}_N(\pi_{\mathsf{D}} \,\|\, \pi) \leq \frac{2}{\log N - 1} D_{\mathsf{seq},N}(\pi_{\mathsf{D}} \,\|\, \pi). \tag{33}$$

**Proof of Proposition F.10.** Consider the stopping time

$$\tau := \min\left\{ h : h = H \text{ or } \sum_{j \leq h} D_{\mathsf{KL}}(\pi_{\mathsf{D}}(y_{j+1} = \cdot \mid x, y_{1:j}) \,\|\, \pi(y_{j+1} = \cdot \mid x, y_{1:j})) > \log N \right\}.$$

Then, for the process $Y^\tau = (x, y_{1:\tau})$, we have the chain rule:

$$D_{\mathsf{KL}}(\pi_{\mathsf{D}}(Y^\tau = \cdot) \,\|\, \pi(Y^\tau = \cdot))$$

$$= \mathbb{E}_{\pi_{\mathsf{D}}} \left[ \sum_{h=1}^\tau D_{\mathsf{KL}}(\pi_{\mathsf{D}}(y_h = \cdot \mid x, y_{1:h-1}) \,\|\, \pi(y_h = \cdot \mid x, y_{1:h-1})) \right]$$

$$\leq \mathbb{E}_{\pi_{\mathsf{D}}} \min\left\{ \log N, \sum_{h=1}^H D_{\mathsf{KL}}(\pi_{\mathsf{D}}(y_h = \cdot \mid x, y_{1:h-1}) \,\|\, \pi(y_h = \cdot \mid x, y_{1:h-1})) \right\},$$

where the inequality uses $\sum_{j < \tau} D_{\mathsf{KL}}(\pi_{\mathsf{D}}(y_{j+1} = \cdot \mid x, y_{1:j}) \,\|\, \pi(y_{j+1} = \cdot \mid x, y_{1:j})) \leq \log N$, which follows from the definition of $\tau$. Therefore, by Proposition F.1, we have

$$\mathbb{P}_{\pi_{\mathsf{D}}}\left( \frac{\pi_{\mathsf{D}}(Y^\tau)}{\pi(Y^\tau)} \geq \log N \right) \leq \frac{D_{\mathsf{KL}}(\pi_{\mathsf{D}}(Y^\tau = \cdot) \,\|\, \pi(Y^\tau = \cdot))}{\log N - 1 + 1/N}.$$

Finally, we bound

$$\mathbb{P}_{\pi_{\mathsf{D}}}\left(\frac{\pi_{\mathsf{D}}(y_{1:H}\mid x)}{\pi(y_{1:H}\mid x)}\geq N\right)\leq\mathbb{P}_{\pi_{\mathsf{D}}}(\tau<H)+\mathbb{P}_{\pi_{\mathsf{D}}}\left(\frac{\pi_{\mathsf{D}}(Y^{\tau})}{\pi(Y^{\tau})}\geq\log N\right).$$

By Markov's inequality,

$$\mathbb{P}_{\pi_{\mathsf{D}}}(\tau<H)\leq\mathbb{P}_{\pi_{\mathsf{D}}}\left(\sum_{h=1}^{H}D_{\mathsf{KL}}(\pi_{\mathsf{D}}(\cdot\mid x,y_{1:h-1})\,\|\,\pi(\cdot\mid x,y_{1:h-1}))>\log N\right)$$

$$\leq\frac{1}{\log N}\,\mathbb{E}_{\pi_{\mathsf{D}}}\min\left\{\log N,\sum_{h=1}^{H}D_{\mathsf{KL}}(\pi_{\mathsf{D}}(\cdot\mid x,y_{1:h-1})\,\|\,\pi(\cdot\mid x,y_{1:h-1}))\right\}.$$

Combining the inequalities above completes the proof. $\qquad\square$

The following result is a sort of partial converse to Proposition F.10, showing that the coverage profile can be lower bounded in terms of the tail behavior for a sum of step-wise Hellinger distances.

**Proposition F.11.** *For any $N\geq 1$ and $\delta\in(0,1)$, it holds that*

$$\mathsf{Cov}_N(\pi_{\mathsf{D}}\,\|\,\pi)\geq\mathbb{P}_{\pi_{\mathsf{D}}}\left(\sum_{h=1}^{H}D_{\mathsf{H}}^2(\pi_{\mathsf{D}}(\cdot\mid x,y_{1:h-1}),\pi(\cdot\mid x,y_{1:h-1}))\geq\log(N/\delta)\right)-\delta.$$

**Proof of Proposition F.11.** By definition,

$$\mathbb{E}_{y_h\sim\pi_{\mathsf{D}}(\cdot\mid x,y_{1:h-1})}\exp\left(-\frac{1}{2}\log\frac{\pi_{\mathsf{D}}(y_h\mid x,y_{1:h-1})}{\pi(y\mid x,y_{1:h-1})}\right)$$
$$=\sum_{y_h\in\mathcal{Y}}\sqrt{\pi_{\mathsf{D}}(y_h\mid x,y_{1:h-1})\cdot\pi(y\mid x,y_{1:h-1})}$$
$$=1-D_{\mathsf{H}}^2(\pi_{\mathsf{D}}(\cdot\mid x,y_{1:h-1}),\pi(\cdot\mid x,y_{1:h-1}))\leq\exp\left(-D_{\mathsf{H}}^2(\pi_{\mathsf{D}}(\cdot\mid x,y_{1:h-1}),\pi(\cdot\mid x,y_{1:h-1}))\right).$$

Therefore, it holds that

$$\mathbb{E}_{\pi_{\mathsf{D}}}\exp\left(\sum_{h=1}^{H}D_{\mathsf{H}}^2(\pi_{\mathsf{D}}(\cdot\mid x,y_{1:h-1}),\pi(\cdot\mid x,y_{1:h-1}))-\frac{1}{2}\log\frac{\pi_{\mathsf{D}}(y_h\mid x,y_{1:h-1})}{\pi(y\mid x,y_{1:h-1})}\right)\leq 1.$$

By Markov inequality, this implies

$$\mathbb{P}_{\pi_{\mathsf{D}}}\left(\frac{1}{2}\log\frac{\pi_{\mathsf{D}}(y_{1:H}\mid x)}{\pi(y_{1:H}\mid x)}\leq\sum_{h=1}^{H}D_{\mathsf{H}}^2(\pi_{\mathsf{D}}(\cdot\mid x,y_{1:h-1}),\pi(\cdot\mid x,y_{1:h-1}))-\log(1/\delta)\right)\leq\delta.$$

To conclude, we note that

$$\mathbb{P}_{\pi_{\mathsf{D}}}\left(\sum_{h=1}^{H}D_{\mathsf{H}}^2(\pi_{\mathsf{D}}(\cdot\mid x,y_{1:h-1}),\pi(\cdot\mid x,y_{1:h-1}))\geq\log(N/\delta)\right)$$
$$\leq\mathbb{P}_{\pi_{\mathsf{D}}}\left(\sum_{h=1}^{H}D_{\mathsf{H}}^2(\pi_{\mathsf{D}}(\cdot\mid x,y_{1:h-1}),\pi(\cdot\mid x,y_{1:h-1}))\geq\frac{1}{2}\log\frac{\pi_{\mathsf{D}}(y_{1:H}\mid x)}{\pi(y_{1:H}\mid x)}+\log(1/\delta)\right)$$
$$+\mathbb{P}_{\pi_{\mathsf{D}}}\left(\frac{1}{2}\log\frac{\pi_{\mathsf{D}}(y_{1:H}\mid x)}{\pi(y_{1:H}\mid x)}+\log(1/\delta)\geq\log(N/\delta)\right)$$
$$\leq\delta+\mathsf{Cov}_N(\pi_{\mathsf{D}}\,\|\,\pi).$$

Re-organizing gives the desired result. $\qquad\square$

# G  ADDITIONAL RESULTS

## G.1  TIGHTNESS OF THEOREM 4.1

To conclude, we show that the coarse and fine-grained terms in Theorem 4.1 are both tight in general.

**Proposition G.1.** *The following lower bounds on coverage hold for the maximum likelihood estimator.*

*(a) Coarse rate: For any $n, d \geq 1$ and $B \geq \log(5n)$, there exists a class $\Pi$ with $\log \mathcal{N}_\infty(\Pi, \alpha) \lesssim d \log(B/\alpha) \vee 1$ and $\pi_{\mathsf{D}} \in \Pi$ such that with probability at least $0.5$, it holds that for any $N \leq e^B$,*

$$\mathsf{Cov}_N(\widehat{\pi}) \geq c \cdot \frac{d}{n}.$$

*(b) Fine rate: For any $n \geq d \geq 1, N \geq 1$, there exists a class $\Pi$ and $\pi_{\mathsf{D}} \in \Pi$ such that $|\Pi| = 2^d + 1$ and $\mathcal{N}_\infty(\Pi, \alpha) \leq 2$ for any $\alpha \geq \sqrt{\frac{d}{n}}$, and with probability at least $0.5$, it holds that*

$$\mathsf{Cov}_N(\widehat{\pi}) \geq c \cdot \frac{d}{n \cdot \log N}.$$

Informally, case (a) shows that for the class $\Pi$ under consideration, the coverage does not decrease with $\log N$ until $N$ is trivially large such that $\log \mathcal{N}_\infty(\Pi, \log N) = 0$; this is precisely the behavior of the coarse term in Theorem 4.1, so this implies there is no hope of removing this term. Meanwhile, case (b) can be interpreted as showing that there is no hope of replacing the high-precision covering number found in the fine-grained term in Theorem 4.1 with a coarser notion (e.g, at the scale in the coarse-grained term), since the rate grows with $d \approx \log|\Pi|$ even though $\log \mathcal{N}(\Pi, \alpha)$ is constant for $\alpha \geq \sqrt{\frac{d}{n}}$. We note that Proposition G.1 is an algorithm-specific lower bound, not an information-theoretic lower bound; we show in Section 6.2 *is* that it is possible to improve over Theorem 4.1 with algorithms explicitly designed to optimize for coverage.

## G.2  MAXIMUM LIKELIHOOD: BETTER COVERAGE FOR CONVEX CLASSES

In this section, we give an extension to Theorem 4.1 which shows that maximum likelihood can achieve a faster convergence rate for coverage—as well as strong tolerance to misspecification—when the model class is convex.

**Assumption G.1** (Convex model class). *The class $\Pi$ satisfies $\Pi = \{\pi_\theta : \theta \in \Theta\}$ for a convex, compact parameter space $\Theta$, and the mapping $\theta \mapsto \pi_\theta(y \mid x)$ is concave for all $x \in \mathcal{X}, y \in \mathcal{Y}$.*

**Theorem G.1** (Fast convergence of coverage for convex classes). *Let $\alpha \geq 0, N' \geq 1, N \geq 2e^{2\alpha}N'$ be given, and suppose that Assumption G.1 holds. Let*

$$\theta^\star \in \operatorname*{arg\,min}_{\theta \in \Theta} D_{\mathsf{KL}}(\pi_{\mathsf{D}} \,\|\, \pi_\theta).$$

*With probability at least $1 - \delta$, the maximum likelihood estimator $\widehat{\pi} := \arg\max_{\pi \in \Pi} \widehat{L}_n(\pi)$ satisfies*

$$\mathsf{Cov}_N(\widehat{\pi}) \leq \mathsf{Cov}_{N'}(\pi_{\theta^\star}) + C\frac{\log \mathcal{N}_\infty(\Pi, \alpha) + \log(\delta^{-1})}{n} + \frac{Ce^{2\alpha}N'}{N} \cdot \inf_{\varepsilon > 0}\left\{\frac{\log \mathcal{N}_\infty(\Pi, \varepsilon)}{n} + \varepsilon\right\}, \tag{34}$$

*where $C > 0$ is an absolute constant.*

Note that we allow for misspecification here, as Eq. (34) shows that the coverage of $\widehat{\pi}$ can be upper bounded by the coverage of $\pi_{\theta^\star}$, the best-in-class approximator of $\pi_{\mathsf{D}}$ with respect to KL-divergence. In the well-specified case where $\pi_{\mathsf{D}} \in \Pi$, the bound simplifies to

$$\mathsf{Cov}_N(\widehat{\pi}) \lesssim \frac{1}{N^{1-2c}} \cdot \inf_{\varepsilon > 0}\left\{\frac{\log \mathcal{N}_\infty(\Pi, \varepsilon)}{n} + \varepsilon\right\} + \frac{\log \mathcal{N}_\infty(\Pi, c\log N) + \log(\delta^{-1})}{n}$$

$$= \frac{\mathcal{C}_{\mathsf{fine}}(\Pi, n)}{N^{1-2c}} + \mathcal{C}_{\mathsf{coarse}}(\Pi, N, n),$$

which improves upon the rate $\mathsf{Cov}_N(\widehat{\pi}) \lesssim \frac{\mathcal{C}_{\mathsf{fine}}(\Pi, n)}{\log N} + \mathcal{C}_{\mathsf{coarse}}(\Pi, N, n)$ in Theorem 4.1. The proof of Theorem G.1 is presented in Appendix J.3.

### G.3 LOWER BOUND FOR MAXIMUM LIKELIHOOD UNDER MISSPECIFICATION

In the following proposition, we show that without a well-specified model class (Assumption 2.1), maximum likelihood may have coverage profile scaling with $\frac{1}{\log M} \min_{\pi \in \Pi} D_{\mathsf{KL}}(\pi_{\mathsf{D}} \parallel \pi)$ (cf. Proposition F.3), even when there exists $\pi \in \Pi$ such that $\mathsf{Cov}_N(\pi) = 0$.

**Proposition G.2** (MLE under misspecification). *For any $\alpha \in [0, 1]$, $M > e^{\alpha}$, there exists a problem instance $\pi_{\mathsf{D}}$ and class $\Pi = \{\pi_1, \pi_2\}$ such that*

$$\sup_{x,y} |\log \pi_{\mathsf{D}}(y \mid x) - \log \pi_1(y \mid x)| \leq \alpha, \qquad \mathsf{Cov}_N(\pi_2) \geq \frac{c\alpha^2}{\log M},$$

*and for any $n \geq 1$, it holds that with probability at least $\frac{1}{4}$, the MLE $\widehat{\pi} = \pi_2$, i.e., $\mathsf{Cov}_N(\widehat{\pi}) = \Omega\left(\frac{\alpha^2}{\log M}\right)$.*

**Proof of Proposition G.2.** Let $p = \frac{\alpha}{32 \log M}$. Consider $\mathcal{X} = \{+, -\}$, $\mathcal{Y} = \{0, 1\}$, $\rho(-) = p, \rho(+) = 1 - p$, and $\pi_{\mathsf{D}}$ is given by

$$\pi_{\mathsf{D}}(\cdot \mid +) = \pi_{\mathsf{D}}(\cdot \mid +) = \mathrm{Ber}\left(\frac{1}{2}\right).$$

We construct the class $\Pi = \{\pi_1, \pi_2\}$ as

$$\pi_1(\cdot|+) = \mathrm{Ber}\left(\frac{1}{2e^{\alpha}}\right), \qquad \pi_2(\cdot|-) = \mathrm{Ber}\left(\frac{1}{2}\right),$$

$$\pi_2(\cdot|+) = \mathrm{Ber}\left(\frac{1}{2}\right), \qquad \pi_2(\cdot|-) = \mathrm{Ber}\left(\frac{1}{2M}\right).$$

Given the dataset $\mathcal{D} = \{(x^t, y^t)\}_{t \in [n]}$ sampled from $\pi_{\mathsf{D}}$, we define $N(x, y) = \#\{t \in [n] : (x^t, y^t) = (x, y)\}$ and $N(x) = N(x, 0) + N(x, 1)$. Then

$$\widehat{L}_n(\pi_2) - \widehat{L}_n(\pi_1) = N(+, 1) \cdot \alpha + N(+, 0) \cdot \log\left(\frac{e^{\alpha}}{2e^{\alpha} - 1}\right) - N(-, 1) \cdot \log M + N(-, 0) \cdot \log\left(2 - \frac{1}{M}\right).$$

By symmetric, it holds that $\mathbb{P}(N(+, 1) \geq N(+, 0)) \geq \frac{1}{2}$. Further, by Markov's inequality, it holds that $\mathbb{P}(N(-) \geq 4np) \leq \frac{1}{4}$. Therefore, for the event $E = \{N(+, 1) \geq N(+, 0), N(-) \leq 4np\}$, we have $\mathbb{P}(E) \geq \frac{1}{4}$. In the following, we show that $\widehat{L}_n(\pi_2) - \widehat{L}_n(\pi_1) > 0$ under $E$.

We condition on $E$. We first note that under this event, we have $N(+, 1) \geq \frac{1}{2}N(+), N(+, 0) \leq \frac{1}{2}N(+)$. Hence,

$$\widehat{L}_n(\pi_2) - \widehat{L}_n(\pi_1) \geq N(+)\left[\frac{1}{2} \cdot \alpha + \frac{1}{2} \cdot \log\left(\frac{e^{\alpha}}{2e^{\alpha} - 1}\right)\right] - N(-) \cdot \log M$$

$$= N(+) \cdot D_{\mathsf{KL}}\left(\mathrm{Ber}\left(\frac{1}{2}\right) \parallel \mathrm{Ber}\left(\frac{1}{2e^{\alpha}}\right)\right) - N(-) \cdot \log M$$

$$\geq N(+) \cdot (1 - e^{-\alpha})^2 - N(-) \cdot \log M.$$

Finally, using the fact that $1 - e^{-\alpha} > \frac{1}{2}\alpha$ and $N(+) \geq (1 - 4p)n \geq \frac{1}{2}n$ under $E$, we have

$$\widehat{L}_n(\pi_2) - \widehat{L}_n(\pi_1) > \left(\frac{\alpha^2}{8} - 4p \log M\right)n = 0.$$

Hence, under the event $E$, we have $\widehat{\pi} = \pi_2$. However, it is clear that

$$\mathsf{Cov}_N(\pi_2) = p, \qquad \mathsf{Cov}_N(\pi_1) = 0.$$

This completes the proof. $\qquad\qquad\square$

### G.4 STOCHASTIC GRADIENT DESCENT: IMPROVED GRADIENT NORMALIZATION FOR DISTILLATION

In this section, we focus on autoregressive linear models (3), and consider a variant of our setting inspired by distillation . We assume that for each example $(x^i, y^i_{1:H})$, for each $h = 1, \dots, H$, we have access to the true next-token probabilities $\pi_{\mathsf{D}}(y_h \mid x^i, y^i_{1:h-1})$ for all $y_h \in \mathcal{V}$. This is an unrealistic assumption for general pre-training, but it is natural for distillation, where $\pi_{\mathsf{D}}$ corresponds to a teacher model (in particular, the next-token probabilities are already computed as part of a standard forward pass through the teacher model).

For the distillation setting, we give an improved gradient normalization scheme that improves upon the rate achieved by Theorem 5.1, closing the gap between SGD and maximum likelihood by matching the guarantee for Theorem 4.2.

Define $\epsilon_\theta(x, y_{1:h-1}) := D_{\mathsf{KL}}(\pi_{\mathsf{D}}(\cdot \mid x, y_{1:h-1}) \,\|\, \pi_\theta(\cdot \mid x, y_{1:h-1}))$; note that for the distillation setting, we can compute this quantity in closed form for any prefix $x, y_{1:h-1}$ in the training corpus. We consider the following (single-sample) truncated/normalized stochastic gradient estimator:

$$\widehat{g}_\theta(y \mid x) = \sum_{h=1}^H \alpha_\theta(x, y_{1:h-1}) \nabla \log \pi_\theta(y_h \mid x, y_{1:h-1}), \tag{35}$$

where $A := \log N$, and where

$$\alpha_\theta(x, y_{1:h-1}) = \begin{cases} 1, & \sum_{j \le h-1} \epsilon_\theta(x, y_{1:j}) \le A, \\ 0, & \sum_{j < h-1} \epsilon_\theta(x, y_{1:j}) > A, \\ \frac{A - \sum_{j < h-1} \epsilon_\theta(x, y_{1:j})}{\epsilon_\theta(x, y_{1:h-1})}, & \text{otherwise.} \end{cases} \tag{36}$$

With this definition, we define the following normalized SGD update:

$$\theta^{t+1} = \text{Proj}_\Theta(\theta^t + \eta \widehat{g}_{\theta^t}(y^t \mid x^t)). \tag{37}$$

Intuitively, the idea behind the update in Eq. (35) is to truncate the gradient at the point where the KL divergence between the teacher and student model is too large, and then normalize the gradient by the KL divergence; this is inspired by the structural result Proposition F.10 in Appendix F.4, where we show a close connection between the coverage profile and a certain "stopped" variant of KL divergence.

**Theorem G.2.** *Let $T, N \ge 1$ be given. With a suitably chosen stepsize $\eta > 0$, the normalized SGD update (10) achieves the following coverage bound:*

$$\mathbb{E}\left[\frac{1}{T} \sum_{t=1}^T \mathsf{Cov}_N(\pi_{\theta^t})\right] \lesssim \sqrt{\frac{\sigma_\star^2}{T \log N}} + \frac{B^2}{T}. \tag{38}$$

This guarantee matches the rate of Theorem 4.2 for the maximum likelihood estimator. The proof is presented in Appendix K.7.

### G.5 AN IMPROVED TOURNAMENT VIA ON-POLICY GENERATION

We describe an improved tournament estimator that is able to remove that $1/N^{1-a}$ term from Theorem 6.2, meaning it achieves nontrivial guarantees even when the coverage parameter $N$ is a constant.

Note that the term $1/N^{1-a}$ of Eq. (15) comes from the fact that $\mathbb{P}_{\pi_{\mathsf{D}}}(\frac{\pi(y|x)}{\pi_{\mathsf{D}}(y|x)} \ge N)$ can be as large as $1/N$ in the worst case, implying that the $\widehat{\pi}$ produced by Eq. (14) may at best achieve a coverage of $1/N$. To overcome this, we introduce an *offset term*:

$$\widehat{\pi} := \arg\min_{\pi \in \Pi} \max_{\pi' \in \Pi} \left\{ \widehat{\mathsf{Cov}}_N(\pi' \,\|\, \pi) - 2N^a \cdot \widehat{\mathsf{Cov}}_N^\pi(\pi' \,\|\, \pi) \right\}, \tag{39}$$

where we define $\widehat{\mathsf{Cov}}_N^{\overline{\pi}}(\pi' \,\|\, \pi) := \frac{1}{n} \sum_{i=1}^n \mathbb{P}_{y \sim \overline{\pi}(\cdot|x^i)}\left(\frac{\pi'(y|x^i)}{\pi(y|x^i)} \ge N\right)$ for models $\pi, \pi', \overline{\pi}$. This estimator augments the simple tournament in Eq. (14) with an "offset" term that accounts for the fact that some of the models might be quite far from $\pi_{\mathsf{D}}$. The main guarantee is as follows.

**Theorem G.3.** *Fix $N \geq 1$, $a > 0$ such that $N^{1-2a} \geq 4$. Suppose that there exists $\bar{\pi} \in \Pi$ such that $|\log \pi_D(y \mid x) - \log \bar{\pi}(y \mid x)| \leq a \log N$ for any $x \in \mathcal{X}, y \in \mathcal{Y}$. Then with probability $1 - \delta$, the tournament estimator (39) achieves* $\mathrm{Cov}_{2N^{1+a}}(\widehat{\pi}) \lesssim \frac{\log(|\Pi|/\delta)}{n}$.

Compared to Theorem 6.2, this tournament eliminates the additive $1/N^{1-a}$ term. It does, however, require a stronger condition on the best-in-class model $\bar{\pi}$ that $|\log \pi_D(y \mid x) - \log \bar{\pi}(y \mid x)| \leq a \log N$, which implies in particular that $\mathrm{Cov}_{N^a}(\bar{\pi}) = 0$.

**Infinite classes: Beating maximum likelihood.** While we motivated the tournament estimators through model/checkpoint selection with a finite class $\Pi$, both estimators can also be applied to general, infinite classes $\Pi$. In this case, it turns out that they both improve upon the coverage achieved by the maximum likelihood estimator in Theorem 4.1, even in the well-specified case where $\pi_D \in \Pi$; informally, the tournament estimators allow us to remove the fine-grained term in Theorem 4.1, leaving only a coarse-grained term. See Theorem 6.2′ and Theorem G.3′ for the formal statements.

# Part II

# Proofs

## H    TECHNICAL TOOLS

**Notation.**    We denote by $\mathbb{B}_2^d(R) := \left\{ v \in \mathbb{R}^d : \|v\| \leq R \right\}$ the $d$-dimensional Euclidean ball of radius $R$. We drop the superscript when the dimension $d$ is clear from context.

### H.1    CONCENTRATION INEQUALITIES

**Lemma H.1** (Freedman's inequality). *Let $(Z^i)_{i \leq n}$ be a real-valued martingale difference sequence adapted to a filtration $(\mathscr{F}_i)_{i \leq n}$. If $|Z^i| \leq R$ almost surely, then for any $\eta \in (0, 1/R)$, with probability at least $1 - \delta$, for all $n' \leq n$,*

$$\sum_{i=1}^{n'} Z^i \leq \eta \sum_{i=1}^{n'} \mathbb{E}_{i-1}\big[(Z^i)^2\big] + \frac{\log(\delta^{-1})}{\eta}.$$

The next result is a standard consequence of Lemma H.1 (e.g., Foster et al. (2021)).

**Lemma H.2.** *Let $(Z^i)_{i \leq n}$ be a sequence of random variables adapted to a filtration $(\mathscr{F}_i)_{i \leq n}$. If $0 \leq Z^i \leq R$ almost surely, then with probability at least $1 - \delta$, for all $n' \leq n$,*

$$\sum_{i=1}^{n'} Z^i \leq \frac{3}{2} \sum_{i=1}^{n'} \mathbb{E}_{i-1}[Z^i] + 4R\log(2\delta^{-1}), \tag{40}$$

*and*

$$\sum_{i=1}^{n'} \mathbb{E}_{i-1}[Z^i] \leq 2 \sum_{i=1}^{n'} Z^i + 8R\log(2\delta^{-1}). \tag{41}$$

The following lemma is a uniform version of, e.g., Lemma 23 in Foster & Rakhlin (2023).

**Lemma H.3.** *Suppose that $\mu$ is a distribution over $\mathcal{Z}$, and let $\mathcal{F} \subseteq (\mathcal{Z} \to \mathbb{R})$ be a function class. We let $N(\mathcal{F}, \epsilon; \|\cdot\|_\infty)$ be the $\epsilon$-covering number of $\mathcal{F}$ under the norm $\rho(f, f') := \sup_{z \in \mathcal{Z}}|f(z) - f'(z)|$. Let $\mathcal{D} = \{Z^1, \cdots, Z^n\}$ be drawn i.i.d. from $\mu$. Then the following holds with probability at least $1 - \delta$:*

$$\sum_{i=1}^{n} f(Z^i) \leq n\log \mathbb{E}_\mu[\exp(f(Z))] + \log(1/\delta) + \inf_{\epsilon \geq 0}\{\log N(\mathcal{F}, \epsilon; \|\cdot\|_\infty) + 2n\epsilon\}, \quad \forall f \in \mathcal{F}.$$

**Proof of Lemma H.3.**    Fix $\epsilon \geq 0$ attaining the minimum of $\log N(\mathcal{F}, \epsilon; \|\cdot\|_\infty) + 2n\epsilon$, and let $f_1, \cdots, f_J$ be an $\epsilon$-covering of $\mathcal{F}$ of size $J = N(\mathcal{F}, \epsilon; \|\cdot\|_\infty)$. For each $j \in [J]$, we define $g_j(z) := f_j(z) - \log \mathbb{E}_\mu[\exp(f_j(Z))]$. Then, it is clear that $\mathbb{E}_\mu\big[e^{g_j(Z)}\big] = 1$, and hence

$$\mathbb{E}\left[\exp\left(\sum_{i=1}^{n} g_j(Z^i)\right)\right] = 1, \qquad \forall j \in [J].$$

By Markov's inequality and the union bound, it holds that with probability at least $1 - \delta$,

$$\sum_{i=1}^{n} g_j(Z^i) \leq \log(J/\delta), \qquad \forall j \in [J]. \tag{42}$$

Note that for any $f \in \mathcal{F}$, there exists $j \in [J]$ such that $\rho(f, f_j) \leq \epsilon$, and in particular

$$f(Z^i) - \log \mathbb{E}_\mu[\exp(f(Z))] \leq 2\epsilon + f_j(Z^i) - \log \mathbb{E}_\mu[\exp(f_j(Z))] = 2\epsilon + g_j(Z^i), \qquad \forall i \in [n],$$

and hence Eq. (42) implies that $\sum_{i=1}^{n} f(Z^i) \leq n\log \mathbb{E}_\mu[\exp(f(Z))] + \log(J/\delta) + 2n\epsilon$. By the arbitrariness of $f$, the proof is hence completed.    $\square$

## H.2 INFORMATION-THEORETIC INEQUALITIES

**Lemma H.4.** *For distribution $P, Q \in \Delta(\mathcal{X})$, function $f : \mathcal{X} \to [-B, B]$, it holds that*

$$|\mathbb{E}_P[f] - \mathbb{E}_Q[f]| \leq 4\sqrt{\mathrm{Var}_Q[f] \cdot D_{\mathsf{H}}^2(P, Q)} + 8BD_{\mathsf{H}}^2(P, Q).$$

*More generally, for any $g : \mathcal{X} \to \mathbb{B}_2(B)$, it holds that*

$$\|\mathbb{E}_P[g] - \mathbb{E}_Q[g]\| \leq 4\sqrt{\mathbb{E}_Q\|g - \mathbb{E}_Q[g]\|^2} \cdot D_{\mathsf{H}}(P, Q) + 8BD_{\mathsf{H}}^2(P, Q). \tag{43}$$

*and*

$$\mathbb{E}_P\|g - \mathbb{E}_P[g]\|^2 \leq 3\,\mathbb{E}_Q\|g - \mathbb{E}_Q[g]\|^2 + 16B^2 D_{\mathsf{H}}^2(P, Q). \tag{44}$$

**Proof of Lemma H.4.** We denote $P(x)$ (resp. $Q(x)$) to be the density function of $P$ (resp. $Q$). Then for any function $f : \mathcal{X} \to \mathbb{R}$,

$$\begin{aligned}
|\mathbb{E}_P[f] - \mathbb{E}_Q[f]|^2 &= \left(\int_{\mathcal{X}} (f(x) - \mathbb{E}_Q[f])(P(x) - Q(x))dx\right)^2 \\
&\leq \int_{\mathcal{X}} (f(x) - \mathbb{E}_Q[f])^2(\sqrt{P(x)} + \sqrt{Q(x)})^2 dx \cdot \int_{\mathcal{X}} (\sqrt{P(x)} - \sqrt{Q(x)})^2 dx \\
&\leq 4D_{\mathsf{H}}^2(P, Q) \cdot \left(\mathrm{Var}_Q[f] + \mathbb{E}_P(f - \mathbb{E}_Q[f])^2\right).
\end{aligned}$$

In particular, when $h : \mathcal{X} \to [0, M]$, the inequality above implies that

$$|\mathbb{E}_P[h] - \mathbb{E}_Q[h]| \leq 2D_{\mathsf{H}}(P, Q)\sqrt{M(\mathbb{E}_P[h] + \mathbb{E}_Q[h])} \leq \frac{1}{2}(\mathbb{E}_P[h] + \mathbb{E}_Q[h]) + 2MD_{\mathsf{H}}^2(P, Q),$$

and hence it holds that $\mathbb{E}_P[h] \leq 3\,\mathbb{E}_Q[h] + 4MD_{\mathsf{H}}^2(P, Q)$.

Now, suppose that $f : \mathcal{X} \to [-B, B]$. Applying the above inequality to $h(x) = (f - \mathbb{E}_Q[f])^2 \in [0, 4B^2]$ gives

$$\mathbb{E}_P(f - \mathbb{E}_Q[f])^2 \leq 3\,\mathbb{E}_Q(f - \mathbb{E}_Q[f])^2 + 16B^2 D_{\mathsf{H}}^2(P, Q). \tag{45}$$

Combining the above inequalities implies that

$$|\mathbb{E}_P[f] - \mathbb{E}_Q[f]| \leq 4\sqrt{\mathrm{Var}_Q[f] \cdot D_{\mathsf{H}}^2(P, Q)} + 8BD_{\mathsf{H}}^2(P, Q).$$

To prove the upper bound for a vector-valued function $g : \mathcal{X} \to \mathbb{B}_2(B)$, we can apply the above inequality with $f_v(x) := \langle v, g(x) \rangle$ and take the maximum over $v \in \mathbb{B}_2(1)$. The second upper bound follows similarly by applying Eq. (45). $\qquad \square$

**Lemma H.5.** *Suppose that $\phi : \mathcal{Y} \to \mathbb{B}_2(B)$ with $B \geq 1$, and for any $\theta \in \mathbb{B}_2(1)$, $\pi_\theta \in \Delta(\mathcal{Y})$ is defined as $\pi_\theta(y) \propto \exp(\langle\phi(y), \theta\rangle)$. Then for any $\theta^\star, \theta \in \mathbb{B}_2(1)$, it holds that*

$$\mathbb{E}_{y \sim \pi_{\theta^\star}}\langle\phi(y) - \mathbb{E}_{\pi_{\theta^\star}}[\phi], \theta - \theta^\star\rangle^2 \leq 15BD_{\mathsf{KL}}(\pi_{\theta^\star} \,\|\, \pi_\theta).$$

**Proof of Lemma H.5.** Denote $\bar{\phi}(y) := \phi(y) - \mathbb{E}_{\pi_{\theta^\star}}[\phi]$. By definition,

$$D_{\mathsf{KL}}(\pi_{\theta^\star} \,\|\, \pi_\theta) = \log \mathbb{E}_{y \sim \pi_{\theta^\star}}\left[\exp(\langle\bar{\phi}(y), \theta - \theta^\star\rangle)\right] \geq B \log \mathbb{E}_{y \sim \pi_{\theta^\star}}\left[\exp\left(\frac{1}{B}\langle\bar{\phi}(y), \theta - \theta^\star\rangle\right)\right].$$

Note that for $x \geq -4$, we have $e^x \geq 1 + x + \frac{1}{10}x^2$. Therefore, we have

$$\frac{1}{B}D_{\mathsf{KL}}(\pi_{\theta^\star} \,\|\, \pi_\theta) \geq \log\left(1 + \frac{1}{10B^2}\,\mathbb{E}_{y \sim \pi_{\theta^\star}}\langle\bar{\phi}(y), \theta - \theta^\star\rangle^2\right) \geq \frac{1}{15B^2}\,\mathbb{E}_{y \sim \pi_{\theta^\star}}\langle\bar{\phi}(y), \theta - \theta^\star\rangle^2,$$

where we use $\log(1 + x) \geq \frac{3}{4}x$ for all $x \in [0, \frac{8}{5}]$. $\qquad \square$

# I    PROOFS FROM SECTION 3

**Proof of Proposition 3.2.**    Consider the setting where $d = 1$, $\mathcal{X} = \{0, 1\}$, $\mathcal{V} = \{-1, 1\}$, the distribution $\mu$ is given by $\mu(1) = 1 - \mu(0) = \frac{1}{2n}$, and the feature map $\phi : \mathcal{X} \times \mathcal{V}^\star \to [-1, 1]$ is given by $\phi(0, \cdot) = 0$, and $\phi(1, y_{1:h}) = y_h$.

In the following, we fix any algorithm $\mathtt{Alg} : (\mathcal{X} \times \mathcal{Y})^n \to \Delta(\Pi)$. Let $\mathbb{P}^{\pi_\theta, \mathtt{Alg}}$ be the probability distribution of $(\mathcal{D} = \{(x^t, y^t)\}_{t \in [n]}, \widehat{\pi})$ where $x^t \sim \mu$, $y^t \sim \pi_\theta(\cdot \mid x^t)$ are sampled i.i.d. and $\widehat{\pi} \sim \mathtt{Alg}(\mathcal{D})$.

Note that under this construction, $\mathbb{P}^{\pi_\theta, \mathtt{Alg}}(x^t = 0 \ \forall t \in [T]) \geq 1 - n\mu(1) = \frac{1}{2}$. Consider the event $E = \{x^t = 0 \ \forall t \in [T]\}$. Then, for any $\theta^\star \in [-1, 1]$, event $A$, it holds that

$$\mathbb{P}^{\pi_{\theta^\star}, \mathtt{Alg}}(A \mid E) = \mathbb{E}^{\pi_0, \mathtt{Alg}}(A \mid E),$$

because for any $\theta \in \Theta$, the distribution $\pi_\theta(y_{1:H} = \cdot \mid 0) = \mathrm{Ber}\left(\frac{1}{2}\right)^{\otimes H}$ is a product of $H$ Bernoulli distributions and does not depend on $\theta$. Furthermore, for any $\theta \in [-1, 1]$,

$$D_{\mathsf{KL}}(\pi_{\theta^\star} \| \pi_\theta) = \mu(1) \cdot D_{\mathsf{KL}}(\pi_{\theta^\star}(y_{1:H} = \cdot \mid x = 1) \| \pi_\theta(y_{1:H} = \cdot \mid x = 1))$$

$$= H\mu(1) \cdot D_{\mathsf{KL}}\left(\mathrm{Ber}\left(\frac{e^{\theta^\star}}{e^{\theta^\star} + e^{-\theta^\star}}\right) \| \mathrm{Ber}\left(\frac{e^\theta}{e^\theta + e^{-\theta}}\right)\right),$$

and hence $\theta \mapsto D_{\mathsf{KL}}(\pi_1 \| \pi_\theta) + D_{\mathsf{KL}}(\pi_{-1} \| \pi)$ is minimized at $\theta = 0$, i.e., for any $\widehat{\pi} \in \Pi$,

$$D_{\mathsf{KL}}(\pi_1 \| \widehat{\pi}) + D_{\mathsf{KL}}(\pi_{-1} \| \widehat{\pi}) \geq \frac{H}{2n} \cdot 2D_{\mathsf{KL}}\left(\mathrm{Ber}\left(\frac{e}{e + e^{-1}}\right) \| \mathrm{Ber}\left(\frac{1}{2}\right)\right) \geq \frac{H}{2n}.$$

Therefore, consider the event $A_\theta := \left\{D_{\mathsf{KL}}(\pi_\theta \| \widehat{\pi}) \geq \frac{H}{4n}\right\}$, and we have shown that $A_1^c \subseteq A_{-1}$. Hence, we can lower bound

$$\mathbb{P}^{\pi_1, \mathtt{Alg}}(A_1) + \mathbb{P}^{\pi_{-1}, \mathtt{Alg}}(A_{-1}) \geq \mathbb{P}^{\pi_1, \mathtt{Alg}}(E)\mathbb{P}^{\pi_1, \mathtt{Alg}}(A_1 \mid E) + \mathbb{P}^{\pi_{-1}, \mathtt{Alg}}(E)\mathbb{P}^{\pi_{-1}, \mathtt{Alg}}(A_{-1} \mid E)$$

$$\geq \frac{1}{2} \mathbb{E}^{\pi_0, \mathtt{Alg}}[A_1 \mid E] + \frac{1}{2} \mathbb{E}^{\pi_0, \mathtt{Alg}}[A_{-1} \mid E] \geq \frac{1}{2}.$$

This gives $\max_{\theta^\star \in \{-1, 1\}} \mathbb{P}^{\pi_{\theta^\star}, \mathtt{Alg}}\left(D_{\mathsf{KL}}(\pi_{\theta^\star} \| \widehat{\pi}) \geq \frac{H}{4n}\right) \geq \frac{1}{4}$, and the desired result follows immediately. $\square$

As a remark, we note that the construction above can be modified so that the variance $\sigma_\star^2$ (defined in Section 4.1) can be bounded as $\sigma_\star^2 \lesssim \frac{He^{-2B}}{n}$. In particular, as long as $B \gtrsim \log H$, it holds that $\sigma_\star \leq 1$, implying that KL can converge slowly even when the "inherent variance" $\sigma_\star$ is small.

# J    PROOFS FROM SECTION 4

## J.1    PROOF SKETCH FOR THEOREM 4.1

The basic idea behind the proof of Theorem 4.1 is to interpret the condition $\mathsf{Cov}_N(\pi) \geq \varepsilon$ as an small-ball like *anti-concentration* condition in the vein of Mendelson (2014; 2017). That is, for models $\pi \in \Pi$ where coverage is large, the condition $\mathsf{Cov}_N(\pi) \geq \varepsilon$ witnesses a *one-sided* tail bound which implies that the empirical likelihood of $\pi$ is *not too large* with high probability, and hence $\pi$ cannot be a maximum-likelihood solution.

Let $c \in (0, 1/2)$ be the absolute constant in Theorem 4.1, and let $C \geq \log 4$ be another absolute constant. Fix $N$ such that $\log N \geq 4C$. For each model $\pi \in \Pi$, let $\mathcal{S}_N(\pi) := \frac{1}{n}|\{i \in [n] \mid \frac{\pi_\mathsf{D}(y^i | x^i)}{\pi(y^i | x^i)} \geq N^{1-2c}\}|$ denote the empirical probability that $\pi$ fails to cover $\pi_\mathsf{D}$. Our first step is to show via covering and concentration that with high-probability, all $\pi \in \Pi$ satisfy

$$\mathcal{S}_N(\pi) \geq \frac{1}{2}\mathsf{Cov}_N(\pi) - \mathcal{C}_{\mathsf{coarse}}(\Pi, N, n). \tag{46}$$

That is, a large coverage profile implies that the number of points in the data where $\pi$ fails to cover $\pi_\mathsf{D}$ is large. This argument only depends on the covering number at a coarse $\log N$ scale—leading to

the coarse-grained term in Theorem 4.1—because we only need to show that coverage concentrates, not the log-loss itself.[7]

We now argue that models with large coverage profile must have low log-likelihood compared to $\pi_D$. In particular, using Eq. (46), we have

$$
\widehat{L}_n(\pi) - \widehat{L}_n(\pi_D) = -\sum_{i=1}^n \left[ \log \frac{\pi_D(y^i \mid x^i)}{\pi(y^i \mid x^i)} - C \right]_+ + \sum_{i=1}^n \log \frac{\pi(y^i \mid x^i)}{\pi_D(y^i \mid x^i)} \vee (-C)
$$

$$
\overset{(\star)}{\leq} -|\mathcal{S}_N(\pi)|((1-2c)\log N - C) + \sum_{i=1}^n \log \frac{\pi(y^i \mid x^i)}{\pi_D(y^i \mid x^i)} \vee (-C)
$$

$$
\leq -\frac{n}{4}\log N \cdot \mathsf{Cov}_N(\pi) + \mathcal{C}_{\mathsf{coarse}}(\Pi, N, n) \cdot O(n \log N) + \sum_{i=1}^n \log \frac{\pi(y^i \mid x^i)}{\pi_D(y^i \mid x^i)} \vee (-C),
\tag{47}
$$

as long as $c \leq 1/8$ and $\log N \geq 4C$. We view step $(\star)$ as using a form of implicit bias in the logarithmic loss: If an example $(x^i, y^i)$ has $\pi_D(y^i|x^i)/\pi(y^i|x^i) \geq N$ (i.e., $\pi$ fails to cover $\pi_D$ on this example), this witnesses a negative contribution of order $\log N$ to the difference $\widehat{L}_n(\pi) - \widehat{L}_n(\pi_D)$.

Next, using a variation of a standard *one-sided* tail bound for the logarithmic loss (van de Geer, 2000; Zhang, 2006),[8] we show that with high probability, all $\pi \in \Pi$ satisfy

$$
\sum_{i=1}^n \log \frac{\pi(y^i \mid x^i)}{\pi_D(y^i \mid x^i)} \vee (-C) \lesssim \mathcal{C}_{\mathsf{fine}}(\Pi, n) \cdot n,
\tag{48}
$$

as long as $C \geq \log 4$. Combining Eq. (47) and Eq. (48), we conclude that all $\pi \in \Pi$ have

$$
\mathsf{Cov}_N(\pi) \lesssim \frac{\widehat{L}_n(\pi_D) - \widehat{L}_n(\pi) + \mathcal{C}_{\mathsf{fine}}(\Pi, n) \cdot n}{n \log N} + \mathcal{C}_{\mathsf{coarse}}(\Pi, N, n).
\tag{49}
$$

Since the maximum likelihood estimator $\widehat{\pi}$ has $\widehat{L}_n(\pi_D) - \widehat{L}_n(\widehat{\pi}) \leq 0$, the result follows.

To summarize the key ideas as they relate to the final guarantee in Theorem 4.1: The coarse-grained term $\mathcal{C}_{\mathsf{coarse}}(\Pi, N, n)$ enters because we only need to show that the coverage profile concentrates, not the log loss itself. The fine-grained term $\mathcal{C}_{\mathsf{fine}}(\Pi, n)$ enters concentration of the empirical likelihood, with the $1/\log N$ scaling arising from implicit bias. The reason this argument avoids dependence on the sequence length $H$ or other spurious parameters that would otherwise affect cross-entropy is that the argument is fundamentally *one-sided*: the conclusion Eq. (49) only shows that models with large coverage profile have low log-likelihood compared to $\pi_D$.

## J.2 PROOF OF THEOREM 4.1 (COVERAGE FOR MLE)

**Theorem 4.1′** (General version of Theorem 4.1). *Let $N \geq 8$ be given. With probability at least $1 - \delta$, any approximate maximum likelihood estimator $\widehat{\pi}$ with $\widehat{L}_n(\widehat{\pi}) \geq \max_{\pi \in \Pi} \widehat{L}_n(\pi) - n\varepsilon_{\mathsf{apx}}$ satisfies*

$$
\mathsf{Cov}_N(\widehat{\pi}) \lesssim \frac{\log \mathcal{N}_\infty(\Pi, c\log N) + \log(\delta^{-1})}{n} + \frac{1}{\log N} \left( \inf_{\varepsilon > 0} \left\{ \frac{\log \mathcal{N}_\infty(\Pi, \varepsilon)}{n} + \varepsilon \right\} + \varepsilon_{\mathsf{apx}} \right),
\tag{50}
$$

*where $c > 0$ is an absolute constant.*

In the following, for a fixed threshold $C \geq \log 4$, we define the clipped log loss as

$$
L_C^+(\pi) := \sum_{i=1}^n \max \left\{ \log \frac{\pi(y^i \mid x^i)}{\pi_D(y^i \mid x^i)}, -C \right\},
\tag{51}
$$

$$
L_C^-(\pi) := \sum_{i=1}^n \max \left\{ 0, \log \frac{\pi_D(y^i \mid x^i)}{\pi(y^i \mid x^i)} - C \right\}.
\tag{52}
$$

---

[7]The set $\mathcal{S}_N(\pi)$ is defined with the threshold as $N^{1-2c}$ rather than $N$ to account for approximation errors incurred by covering.

[8]That the bound is one-sided is critical, as this allows us to avoid paying for the range of the density ratios under consideration. For details, see Proposition J.1.

Note that $\widehat{L}_n(\pi) - \widehat{L}_n(\pi_{\mathsf{D}}) = L_C^+(\pi) - L_C^-(\pi)$. Furthermore, since $\pi_{\mathsf{D}} \in \Pi$, the approximate maximum likelihood estimator satisfies $\widehat{L}_n(\widehat{\pi}) \geq \widehat{L}_n(\pi_{\mathsf{D}}) - n\varepsilon_{\mathsf{apx}}$, and hence

$$L_C^-(\widehat{\pi}) \leq L_C^+(\widehat{\pi}) + n\varepsilon_{\mathsf{apx}}.$$

In the following, we show that $L_C^+(\pi)$ can be bounded by a one-sided uniform convergence argument, and show that $L_C^-(\pi)$ upper bounds the coverage profile $\mathsf{Cov}_N(\pi)$ for any $\pi \in \Pi$ and $\log N > C$.

**Proposition J.1.** *Suppose that $C \geq \log 4$. Then, with probability at least $1 - \delta$, it holds that for any $\pi \in \Pi$,*

$$L_C^+(\pi) \leq \log(1/\delta) + 2 \inf_{\epsilon \geq 0} \{\log \mathcal{N}_\infty(\Pi, \epsilon) + n\epsilon\}.$$

**Proposition J.2.** *Fix any $\alpha \in (0, \frac{\log N - C}{2})$. Then, with probability at least $1 - \delta$, it holds that*

$$\mathsf{Cov}_N(\pi) \leq \frac{2}{\log N - C - 2\alpha} \cdot L_C^-(\pi) + \frac{16 \log(2\mathcal{N}_\infty(\Pi, \alpha)/\delta)}{n}.$$

The proof of Theorem 4.1 and Theorem 4.1′ is completed by combining the propositions above and setting $\alpha = \frac{1}{4} \log N$. In what follows, we prove the propositions. □

**Proof of Proposition J.1.** This is a direct corollary of Lemma H.3. For each $\pi \in \Pi$, we let $f_\pi(x, y) := \frac{1}{2} \max\left\{\log \frac{\pi(y|x)}{\pi_{\mathsf{D}}(y|x)}, -C\right\}$ and consider the function class $\mathcal{F} = \{f_\pi : \pi \in \Pi\}$. Then, $N(\mathcal{F}, \epsilon; \|\cdot\|_\infty) \leq \mathcal{N}_\infty(\Pi, 2\epsilon)$ for any $\epsilon \geq 0$. Applying Lemma H.3 with Lemma J.1 (stated and proved below) gives the desired upper bound. □

**Lemma J.1.** *As long as $C \geq \log 4$, it holds that*

$$\mathbb{E}_{(x,y)\sim\pi_{\mathsf{D}}} \exp\left(\frac{1}{2} \max\left\{\log \frac{\pi(y \mid x)}{\pi_{\mathsf{D}}(y \mid x)}, -C\right\}\right) \leq 1. \tag{53}$$

**Proof of Lemma J.1.** We denote $u = e^{-C}$ and $E := \left\{(x, y) : \frac{\pi(y|x)}{\pi_{\mathsf{D}}(y|x)} \geq u\right\}$. Then it holds that

$$\mathbb{E}_{(x,y)\sim\pi_{\mathsf{D}}} \exp\left(\frac{1}{2} \max\left\{\log \frac{\pi(y \mid x)}{\pi_{\mathsf{D}}(y \mid x)}, -C\right\}\right)$$

$$= \mathbb{E}_{(x,y)\sim\pi_{\mathsf{D}}} \left[\sqrt{\frac{\pi(y \mid x)}{\pi_{\mathsf{D}}(y \mid x)}} \mathbb{I}\{(x,y) \in E\} + \sqrt{u}\, \mathbb{I}\{(x,y) \notin E\}\right]$$

$$= \mathbb{E}_{x\sim\pi_{\mathsf{D}}} \left[\sum_{y:(x,y)\in E} \sqrt{\pi(y \mid x)\pi_{\mathsf{D}}(y \mid x)}\right] + \sqrt{u}\, \mathbb{P}_{\pi_{\mathsf{D}}}(E^c).$$

For $x \in \mathcal{X}$, denote $E_x := \{y : (x, y) \in E\}$. By the Cauchy-Schwarz inequality, we have

$$\sum_{y:(x,y)\in E} \sqrt{\pi(y \mid x)\pi_{\mathsf{D}}(y \mid x)} \leq \sqrt{\sum_{y\in E_x} \pi(y \mid x) \cdot \sum_{y\in E_x} \pi_{\mathsf{D}}(y \mid x)} \leq \sqrt{\mathbb{P}_{y\sim\pi_{\mathsf{D}}(\cdot|x)}(E_x)}.$$

Therefore, as long as $u \leq \frac{1}{4}$ (or equivalently, $C \geq \log 4$), it holds that

$$\mathbb{E}_{(x,y)\sim\pi_{\mathsf{D}}} \exp\left(\frac{1}{2} \max\left\{\log \frac{\pi(y \mid x)}{\pi_{\mathsf{D}}(y \mid x)}, -C\right\}\right) \leq \sqrt{\mathbb{P}_{\pi_{\mathsf{D}}}(E)} + \frac{1}{2}\mathbb{P}_{\pi_{\mathsf{D}}}(E^c) \leq 1,$$

where we use $1 - p = (1 + \sqrt{p})(1 - \sqrt{p}) \leq 2(1 - \sqrt{p})$ for any $p \in [0, 1]$. □

**Proof of Proposition J.2.** Fix any $N \geq 1, \alpha \geq 0$. By definition, for any $\pi \in \Pi$,

$$
\begin{aligned}
L_C^-(\pi) &= \sum_{i=1}^n \max\left\{0, \log \frac{\pi_{\mathsf{D}}(y^i \mid x^i)}{\pi(y^i \mid x^i)} - C\right\} \\
&\geq (\log N - C)\left|\left\{i \in [n] : \log \frac{\pi_{\mathsf{D}}(y^i \mid x^i)}{\pi(y^i \mid x^i)} \geq \log N\right\}\right| \\
&= n(\log N - C) \cdot \widehat{\mathsf{Cov}}_N(\pi_{\mathsf{D}} \| \pi),
\end{aligned}
$$

where we recall that (see Eq. (13))

$$
\widehat{\mathsf{Cov}}_N(\pi_{\mathsf{D}} \| \pi) = \frac{1}{n}\left|\left\{t \in [n] : \frac{\pi_{\mathsf{D}}(y^t \mid x^t)}{\pi(y^t \mid x^t)} \geq N\right\}\right|.
$$

Then, by Lemma J.2 (stated and proved below), it holds that with probability at least $1 - \delta$, for any $\pi \in \Pi$,

$$
\widehat{\mathsf{Cov}}_N(\pi_{\mathsf{D}} \| \pi) \geq \frac{1}{2}\mathsf{Cov}_{e^{2\alpha}N}(\pi_{\mathsf{D}} \| \pi) - \frac{8\log(2\mathcal{N}_\infty(\Pi, \alpha)/\delta)}{n}.
$$

Rescaling $N \leftarrow e^{-2\alpha}N$ and reorganizing completes the proof. $\qquad\square$

**Lemma J.2.** *For any model $\pi, \pi'$, we consider the quantities*

$$
\widehat{\mathsf{Cov}}_N(\pi' \| \pi) = \frac{1}{n}\left|\left\{t \in [n] : \frac{\pi'(y^t \mid x^t)}{\pi(y^t \mid x^t)} \geq N\right\}\right|, \qquad \mathsf{Cov}_N^{\pi_{\mathsf{D}}}(\pi' \| \pi) = \mathbb{P}_{\pi_{\mathsf{D}}}\left(\frac{\pi'(y \mid x)}{\pi(y \mid x)} \geq M\right).
$$

*Fix $\alpha \geq 0$ and model $\overline{\pi}$. With probability at least $1 - \delta$, for any $\pi \in \Pi$, it holds that*

$$
\widehat{\mathsf{Cov}}_N(\overline{\pi} \| \pi) \geq \frac{1}{2}\mathsf{Cov}_{e^{2\alpha}N}^{\pi_{\mathsf{D}}}(\overline{\pi} \| \pi) - \frac{8\log(2\mathcal{N}_\infty(\Pi, \alpha)/\delta)}{n}.
$$

*Similarly, with probability at least $1 - \delta$, for any $\pi \in \Pi$, it holds that*

$$
\widehat{\mathsf{Cov}}_N(\pi \| \overline{\pi}) \leq 2\,\mathsf{Cov}_{e^{-2\alpha}N}^{\pi_{\mathsf{D}}}(\pi \| \overline{\pi}) + \frac{8\log(2\mathcal{N}_\infty(\Pi, \alpha)/\delta)}{n}.
$$

**Proof of Lemma J.2.** We only prove the first inequality. Let $\Pi' \subseteq \Pi$ be an $\alpha$-covering of $\Pi$ with $|\Pi'| = \mathcal{N}_\infty(\Pi, \alpha)$. Then, by Freedman's inequality (Lemma H.2) and union bound, it holds that with probability at least $1 - \delta$, for any $\pi' \in \Pi'$,

$$
\widehat{\mathsf{Cov}}_{e^\alpha N}(\overline{\pi} \| \pi') \geq \frac{1}{2}\mathsf{Cov}_{e^\alpha N}^{\pi_{\mathsf{D}}}(\overline{\pi} \| \pi') - \varepsilon_{\mathsf{stat}},
$$

where we denote $\varepsilon_{\mathsf{stat}} = \frac{8\log(2|\Pi'|/\delta)}{n}$. Then, note that for any $\pi \in \Pi$, there exists $\pi' \in \Pi'$ such that $|\log \pi(y \mid x) - \log \pi'(y \mid x)| \leq \alpha$ for $\forall x, y$, we know

$$
\left\{t \in [n] : \frac{\overline{\pi}(y^t \mid x^t)}{\pi'(y^t \mid x^t)} \geq e^\alpha N\right\} \subseteq \left\{t \in [n] : \frac{\overline{\pi}(y^t \mid x^t)}{\pi(y^t \mid x^t)} \geq N\right\}
$$

and hence $\widehat{\mathsf{Cov}}_{e^\alpha N}(\overline{\pi} \| \pi') \leq \widehat{\mathsf{Cov}}_N(\overline{\pi} \| \pi)$. Similarly, $\mathsf{Cov}_{e^\alpha N}^{\pi_{\mathsf{D}}}(\overline{\pi} \| \pi') \geq \mathsf{Cov}_{e^{2\alpha}N}^{\pi_{\mathsf{D}}}(\overline{\pi} \| \pi)$. Hence, under the above event, it holds that

$$
\begin{aligned}
\widehat{\mathsf{Cov}}_N(\overline{\pi} \| \pi) &\geq \widehat{\mathsf{Cov}}_{e^\alpha N}(\overline{\pi} \| \pi') \geq \frac{1}{2}\mathsf{Cov}_{e^\alpha N}^{\pi_{\mathsf{D}}}(\overline{\pi} \| \pi') - \varepsilon_{\mathsf{stat}} \\
&\geq \frac{1}{2}\mathsf{Cov}_{e^{2\alpha}N}^{\pi_{\mathsf{D}}}(\overline{\pi} \| \pi) - \varepsilon_{\mathsf{stat}}.
\end{aligned}
$$

Since $\pi \in \Pi$ is arbitrary, the proof is hence completed. $\qquad\square$

## J.3 Proof of Theorem G.1 (Coverage for MLE with Convex Classes)

Let $\alpha \geq 0$, $N' \geq 1$, $N \geq 2e^{2\alpha}N'$ be fixed. By definition and concavity of $\theta \mapsto \pi_\theta(y \mid x)$, we know $\theta^\star$ is an optimal solution of the following concave problem

$$\theta^\star \in \arg\max_{\theta \in \Theta} \mathbb{E}_{(x,y) \sim \pi_D}[\log \pi_\theta(y \mid x)].$$

Hence, the optimality of $\theta^\star$ implies that

$$\langle \theta - \theta^\star, -\mathbb{E}_{\pi_D}[\nabla \log \pi_{\theta^\star}(y \mid x)] \rangle \geq 0, \qquad \forall \theta \in \Theta.$$

Consider the function $F(\theta) = \mathbb{E}_{\pi_D}\left[\frac{\pi_\theta(y|x)}{\pi_{\theta^\star}(y|x)}\right] - 1$, which is also concave by Assumption G.1. For any $\theta \in \Theta$,

$$\langle \theta - \theta^\star, -\nabla F(\theta^\star) \rangle = \left\langle \theta - \widehat{\theta}, -\mathbb{E}_{\pi_D}\left[\frac{\nabla \pi_{\theta^\star}(y \mid x)}{\pi_{\theta^\star}(y \mid x)}\right] \right\rangle = \left\langle \theta - \widehat{\theta}, -\mathbb{E}_{\pi_D}[\nabla \log \pi_{\theta^\star}(y \mid x)] \right\rangle \geq 0.$$

Therefore, $F$ attains its maximum over $\Theta$ at $\theta^\star$, i.e., $F(\theta) \leq F(\theta^\star) = 0$ for any $\theta \in \Theta$.

Similarly, it is also clear that $\theta \mapsto \sum_{i=1}^n \log \pi_\theta(y^i \mid x^i)$ is concave, and hence $\widehat{\pi} = \pi_{\widehat{\theta}}$, where $\widehat{\theta} \in \Theta$ satisfies

$$\left\langle \theta - \widehat{\theta}, \sum_{i=1}^n -\nabla \log \pi_{\widehat{\theta}}(y^i \mid x^i) \right\rangle \geq 0, \qquad \forall \theta \in \Theta.$$

In particular, we consider the function

$$\widehat{F}(\theta) := \sum_{i=1}^n \left[\frac{\pi_\theta(y^i \mid x^i)}{\pi_{\widehat{\theta}}(y^i \mid x^i)} - 1\right].$$

Under Assumption G.1, $\widehat{F}$ is concave, and for any $\theta \in \Theta$,

$$\left\langle \theta - \widehat{\theta}, -\nabla \widehat{F}(\widehat{\theta}) \right\rangle = \left\langle \theta - \widehat{\theta}, -\sum_{i=1}^n \frac{\nabla \pi_{\widehat{\theta}}(y^i \mid x^i)}{\pi_{\widehat{\theta}}(y^i \mid x^i)} \right\rangle = \left\langle \theta - \widehat{\theta}, \sum_{i=1}^n -\nabla \log \pi_{\widehat{\theta}}(y^i \mid x^i) \right\rangle \geq 0.$$

Therefore, $\widehat{F}$ attains its maximum over $\Theta$ at $\widehat{\theta}$, and in particular, $\widehat{F}(\theta^\star) \leq \widehat{F}(\widehat{\theta}) = 0$. This implies

$$\sum_{i=1}^n \left[\frac{\pi_{\theta^\star}(y^i \mid x^i)}{\widehat{\pi}(y^i \mid x^i)} - \log \frac{\pi_{\theta^\star}(y^i \mid x^i)}{\widehat{\pi}(y^i \mid x^i)} - 1\right] \leq \sum_{i=1}^n \log \widehat{\pi}(y^i \mid x^i) - \sum_{i=1}^n \log \pi_{\theta^\star}(y^i \mid x^i). \tag{54}$$

In the following, we use that $N \geq 2$. Note that $x - \log x - 1 \geq 0$ for any $x > 0$, and $x \mapsto x - \log x - 1$ is increasing for $x \geq 1$. Therefore, Eq. (54) implies that

$$(N - \log N - 1) \cdot n \cdot \widehat{\mathsf{Cov}}_N(\pi_{\theta^\star} \| \widehat{\pi}) \leq \widehat{L}_n(\widehat{\pi}) - \widehat{L}_n(\pi_{\theta^\star}). \tag{55}$$

Then, by Lemma J.2, we have with probability at least $1 - \delta$, for all $\pi \in \Pi$,

$$\widehat{\mathsf{Cov}}_N(\pi_{\theta^\star} \| \pi) \geq \frac{1}{2} \cdot \mathbb{P}_{\pi_D}\left(\frac{\pi_{\theta^\star}(y \mid x)}{\pi(y \mid x)} \geq e^{2\alpha}N\right) - \frac{\log(\mathcal{N}_\infty(\Pi, \alpha)/\delta)}{n}, \qquad \forall \pi \in \Pi.$$

Further, by Lemma H.3, the following holds with probability at least $1 - \delta$: For any $\theta \in \Theta$,

$$\begin{aligned}
\widehat{L}_n(\pi_\theta) - \widehat{L}_n(\pi_{\theta^\star}) &= \sum_{i=1}^n \log \frac{\pi_\theta(y^i \mid x^i)}{\pi_{\theta^\star}(y^i \mid x^i)} \\
&\leq n \log \mathbb{E}_{\pi_D}\left[\frac{\pi_\theta(y \mid x)}{\pi_{\theta^\star}(y \mid x)}\right] + \inf_{\epsilon \geq 0}\{\log(\mathcal{N}_\infty(\Pi, \epsilon)/\delta) + 2n\epsilon\} \\
&\leq \inf_{\epsilon \geq 0}\{\log(\mathcal{N}_\infty(\Pi, \epsilon)/\delta) + 2n\epsilon\},
\end{aligned}$$

where we use $\mathbb{E}_{\pi_D}\left[\frac{\pi_\theta(y|x)}{\pi_{\theta^\star}(y|x)}\right] = F(\theta) + 1 \leq 1$ for any $\theta \in \Theta$. By union bound, we have shown that with probability at least $1 - 2\delta$,

$$\mathbb{P}_{\pi_D}\left(\frac{\pi_{\theta^\star}(y \mid x)}{\widehat{\pi}(y \mid x)} \geq e^{2\alpha}N\right) \lesssim \frac{\log(\mathcal{N}_\infty(\Pi, \alpha)/\delta)}{n} + \frac{1}{N}\inf_{\epsilon \geq 0}\left\{\frac{\log \mathcal{N}_\infty(\Pi, \epsilon)}{n} + \epsilon\right\}.$$

Note that

$$
\begin{aligned}
\mathsf{Cov}_{e^{2\alpha}NN'}(\widehat{\pi}) = \mathbb{P}_{\pi_{\mathsf{D}}}\left( \frac{\pi_{\mathsf{D}}(y \mid x)}{\widehat{\pi}(y \mid x)} \geq e^{2\alpha}NN' \right) \\
\leq \mathbb{P}_{\pi_{\mathsf{D}}}\left( \frac{\pi_{\theta^\star}(y \mid x)}{\widehat{\pi}(y \mid x)} \geq e^{2\alpha}N \right) + \mathbb{P}_{\pi_{\mathsf{D}}}\left( \frac{\pi_{\mathsf{D}}(y \mid x)}{\pi_{\theta^\star}(y \mid x)} \geq N' \right).
\end{aligned}
$$

Therefore, the proof is completed by rescaling $N \leftarrow Ne^{-2\alpha}/N'$, $\delta \leftarrow \frac{\delta}{2}$ and combining the inequalities above. $\qquad\square$

### J.4 PROOFS FOR SUPPORTING RESULTS

**Proof of Proposition G.1 (a).** Assume that $B \geq \log(5n)$ and $n \geq d \geq 2$. Consider $\mathcal{X} = \perp$, $\mathcal{Y} = [d]$ and let the feature map be given by $\phi(y) = Be_y$ for $y \in \mathcal{Y}$, where $(e_1, \ldots, e_d)$ is the coordinate basis of $\mathbb{R}^d$. We consider $\Theta = \{\theta \in \mathbb{R}^d : \|\theta\|_\infty \leq 1\}$, and we set

$$
\theta^\star = \frac{\log(4n)}{2B} \cdot \left( e_1 - \sum_{j=2}^{d} e_j \right).
$$

Then it holds that

$$
\pi_{\mathsf{D}}(1) = \frac{4n}{d-1+4n}, \qquad \pi_{\mathsf{D}}(y) = \frac{1}{d-1+4n}, \qquad \forall y > 1.
$$

Given the dataset $\mathcal{D} = \{y^1, \cdots, y^n\}$, we consider the random variables $n_y = |\{i \in [n] : y^i = y\}|$. Note that under $\mathcal{D} \sim \pi_{\mathsf{D}}$, it holds that

$$
\mathbb{E}\left[ \sum_{y>1} n_y \right] = \mathbb{E}\left[ \sum_{t=1}^{n} \mathbb{I}\{y^t \neq 1\} \right] \leq \frac{n(d-1)}{d-1+4n} \leq \frac{d-1}{4}.
$$

In particular, with probability at least 0.5, it holds that $\sum_{y>1} n_y \leq \frac{d-1}{2}$, i.e., the set $\mathcal{Y}_0 := \{y \in [d] : n_y = 0\}$ has cardinality at least $\frac{d-1}{2}$.

In the following, we condition on this event analyze the MLE $\widehat{\theta}$. By the definition of MLE,

$$
\widehat{\theta} \in \arg\max_{\theta \in \Theta} -n \log\left( \sum_{y \in [d]} e^{B\theta_y} \right) + B \sum_{y \in [d]} n_y \theta_y.
$$

We denote $p_y := \pi_{\widehat{\theta}}(y) = \frac{e^{B\widehat{\theta}_y}}{\sum_{i \in [d]} e^{B\widehat{\theta}_i}}$. Then, the KKT conditions imply that for each $y \in [d]$, either $p_y = \frac{n_y}{n}$, or $\widehat{\theta}_y = -1$ and $p_y \geq \frac{n_y}{n}$, or $\widehat{\theta}_y = 1$ and $p_y \leq \frac{n_y}{n}$. In particular, for any $y \in \mathcal{Y}_0$, $p_y > 0 = \frac{n_y}{n}$, and hence it must hold that $\widehat{\theta}_y = -1$. Then, because $\sum_{y \in [d]} p_y = 1 = \sum_{y \in [d]} \frac{n_y}{n}$, there must exist $j \in [d]$ such that $p_j < \frac{n_j}{n}$, and by the KKT condition we have $\widehat{\theta}_j = 1$. Therefore, for any $y \in \mathcal{Y}_0$, it holds that $p_y \leq \frac{e^{-B}}{e^{-B}+e^B} \leq \frac{1}{e^{2B}}$, and in particular $\frac{\pi_{\mathsf{D}}(y)}{\pi_{\widehat{\theta}}(y)} \geq \frac{e^{2B}}{4n+d-1} \geq e^B$. This implies that

$$
\mathsf{Cov}_{e^B}(\pi_{\widehat{\theta}}) = \mathbb{P}_{\pi_{\mathsf{D}}}\left( \frac{\pi_{\mathsf{D}}(y)}{\pi_{\widehat{\theta}}(y)} \geq e^B \right) \geq \mathbb{P}_{\pi_{\mathsf{D}}}(\mathcal{Y}_0) \geq \frac{d-1}{2(d-1+4n)} \geq \frac{d-1}{10n}.
$$

This is the desired lower bound. $\qquad\square$

**Proof of Proposition G.1 (b).** Let $\epsilon = c_0\sqrt{\frac{d}{n}}$ and $p = \frac{c_0\epsilon^2}{\log N}$ for a sufficiently small absolute constant $c_0 > 0$, $\mathcal{X} = \{0, 1, \cdots, d\}$, $\mathcal{Y} = \{0, 1\}$, and the distribution $\mu$ be given by $\mu(0) = p$, $\mu(1) = \cdots = \mu(d) = \frac{1-p}{d}$.

Let the data distribution $\pi_{\mathsf{D}}$ be $\pi_{\mathsf{D}}(\cdot \mid i) = \mathrm{Ber}(1/2)$ for $i \in [d]$ and $\pi_{\mathsf{D}}(1 \mid 0) = 1$. For any $\theta \in \Theta := \{+1, -1\}^d$, we define $\pi_\theta$ as

$$
\pi_\theta(\cdot \mid 0) = \mathrm{Ber}\left( \frac{1}{N} \right), \qquad \pi_\theta(\cdot \mid i) = \mathrm{Ber}\left( \frac{1+\epsilon\theta_i}{2} \right), \qquad \forall i \in [d].
$$

Consider the model class $\Pi = \{\pi_{\mathsf{D}}\} \cup \{\pi_\theta : \theta \in \Theta\}$. Note that for any $\theta \in \Theta$, $\mathsf{Cov}_N(\pi_{\mathsf{D}} \,\|\, \pi_\theta) \geq \mu(0) = p$.

Then, we can calculate

$$\widehat{L}_n(\pi_\theta) - \widehat{L}_n(\pi_{\mathsf{D}}) = -C(0,1)\log N + \sum_{i\in[d]}[C(i,1)\log(1+\epsilon\theta_i) + C(i,0)\log(1-\epsilon\theta_i)],$$

where we denote $C(x,y) = |\{t \in [n] : (x^t, y^t) = (x,y)\}|$. We further write $C(x) = C(x,0) + C(x,1)$. Taking maximum over $\theta \in \Theta = \{-1,1\}^d$ gives

$$\max_{\theta\in\Theta} \widehat{L}_n(\pi_\theta) - \widehat{L}_n(\pi_{\mathsf{D}})$$

$$= -C(0)\log N + \frac{1}{2}\sum_{i\in[d]}\left[|C(i,1)-C(i,0)|\log\frac{1+\epsilon}{1-\epsilon} + C(i)\log(1-\epsilon^2)\right]$$

$$\geq -C(0)\log N - n\epsilon^2 + \frac{\epsilon}{2}\sum_{i\in[d]}|C(i,0)-C(i,1)|,$$

In the following, we denote $\Delta_i = C(i,1) - C(i,0)$ and $\Delta := \sum_{i\in[d]}\Delta_i$. Note that for any $i \in [d]$, condition on $C(i)$, $\Delta_i$ is a sum of $C(i)$ i.i.d. random variables drawn from $\mathsf{Unif}(\{-1,1\})$, and hence

$$\mathbb{E}[(\Delta_i)^2 \mid C(i)] = C(i), \qquad \mathbb{E}[|\Delta_i| \mid C(i)] \geq \sqrt{\frac{C(i)}{2}},$$

where we apply Khintchine's inequality. In addition, we note that $C(i) \sim B(n,q)$ is a binomial random variable, where $q = \frac{1-p}{d}$. Hence, $\mathbb{E}[C(i)] = nq$, and to lower bound $\mathbb{E}\sqrt{C(i)}$, we invoke Lemma J.3 (stated and proven in the sequel) to show that $\mathbb{E}\sqrt{C(i)} \geq \sqrt{nq}\left(1 - \frac{1-q}{2nq}\right) \geq \frac{\sqrt{nq}}{2}$ (because $n \geq 2d$ and hence $nq \geq 1$). Therefore,

$$\mathbb{E}[\Delta] = \sum_{i\in[d]}\mathbb{E}[|\Delta_i|] \geq \frac{1}{\sqrt{2}}\sum_{i\in[d]}\mathbb{E}[\sqrt{C(i)}] \geq \frac{d\sqrt{nq}}{2\sqrt{2}},$$

and we can also bound $\mathbb{E}(\Delta)^2 \leq d\sum_{i\in[d]}\mathbb{E}(\Delta_i)^2 = d\sum_{i\in[d]}\mathbb{E}[C(i)] = dn(1-p) = d^2nq$. Then, by Paley-Zygmund inequality, it holds that

$$\mathbb{P}(\Delta > b\,\mathbb{E}[\Delta]) \geq (1-b)^2\frac{(\mathbb{E}[\Delta])^2}{\mathbb{E}[\Delta^2]} \geq \frac{(1-b)^2}{8}, \qquad \forall b \in [0,1].$$

We choose $b = 1 - \sqrt{0.88}$ to be a numeric constant so that $\mathbb{P}(\Delta > b\,\mathbb{E}[\Delta]) \geq 0.11$. By Markov's inequality, it also holds that $\mathbb{P}(C(0) \geq 100np) \leq 0.01$. In the following, we condition on the event $E = \{\Delta > b\,\mathbb{E}[\Delta]\} \cap \{C(0) \leq 100np\}$ (note that $\mathbb{P}(E) \geq 0.1$). Then, we have

$$\max_{\theta\in\Theta} \widehat{L}_n(\pi_\theta) - \widehat{L}_n(\pi_{\mathsf{D}}) \geq -C(0)\log N - n\epsilon^2 + \frac{\epsilon}{2}\Delta > \frac{b\epsilon\sqrt{nd}}{8} - 100np\log N - n\epsilon^2 \geq 0,$$

as long as $c_0 \leq 10^{-4}$. This implies that there exists $\theta \in \Theta$ such that $\widehat{\pi} = \pi_\theta$, and hence $\mathsf{Cov}_N(\widehat{\pi}) \geq p$. This is the desired lower bound. $\qquad\square$

**Lemma J.3.** *For non-negative random variable $Z$, it holds that $\mathbb{E}[\sqrt{Z}] \geq \sqrt{\mathbb{E}[Z]}\left(1 - \frac{\mathrm{Var}[Z]}{2(\mathbb{E}[Z])^2}\right)$.*

**Proof of Lemma J.3.** Note that the inequality $\sqrt{u} \geq \frac{3u-u^2}{2}$ holds for $u \geq 0$. Setting $u = \frac{Z}{\mathbb{E}[Z]}$ and taking expectation completes the proof. $\qquad\square$

## K  PROOFS FOR AUTOREGRESSIVE LINEAR MODELS

### K.1  ORGANIZATION

This section contains proofs for all of the results in Sections 4 to 6 concerning autoregressive linear models (3). We begin with the proof of Theorem 4.2 (MLE for autoregressive linear models).

We then present the proofs for various SGD methods, starting with vanilla SGD (Proposition 5.1; upper and lower bounds), followed by normalized SGD (Theorem 5.1), test-time training (Theorem 6.1), and expert-guided gradient normalization (Theorem G.2). The final subsection provides an additional lower bound, showing that the dependence on the parameter $\sigma_\star^2$ is necessary in high dimension.

Throughout this section, all upper bounds are derived under Assumptions 2.1 and 2.2, i.e., we assume that $\Theta \subseteq \mathbb{B}_2(1)$, $\phi : \mathcal{X} \times \mathcal{V}^\star \to \mathbb{B}_2(R)$, and $\pi_D = \pi_{\theta^\star}$ is realized by some parameter $\theta^\star \in \Theta$.

**Notation and preliminaries.** For any $f : \mathcal{X} \times \mathcal{V}^\star \to \mathbb{R}$ and dataset $\mathcal{D} = \{(x^i, y^i_{1:H})\}_{i \in [n]}$, we write

$$\widehat{\mathbb{E}}_\mathcal{D}[f] := \frac{1}{n} \sum_{i=1}^n f(x^i, y^i_{1:H}),$$

For notational simplicity, we denote

$$\bar{\phi}_\theta(x, y_{1:h-1}) = \mathbb{E}_{y_h \sim \pi_\theta(\cdot | x, y_{1:h-1})}[\phi(x, y_{1:h})],$$

and

$$\phi^\star(x, y_{1:h}) := \phi(x, y_{1:h}) - \bar{\phi}_{\theta^\star}(x, y_{1:h-1}),$$
$$\mathrm{Var}_{\pi_D}(x, y_{1:h-1}) := \mathbb{E}_{y_h \sim \pi_\theta(\cdot | x, y_{1:h-1})} \|\phi^\star(x, y_{1:h})\|^2.$$

Then, by definition,

$$\nabla \log \pi_\theta(y_{1:H} \mid x) = \sum_{h=1}^H \big(\phi(x, y_{1:h}) - \bar{\phi}_\theta(x, y_{1:h-1})\big)$$
$$= \sum_{h=1}^H \phi^\star(x, y_{1:h}) + \sum_{h=1}^H \big(\bar{\phi}_{\theta^\star}(x, y_{1:h-1}) - \bar{\phi}_\theta(x, y_{1:h-1})\big), \qquad (56)$$

and it holds that $\sigma_\star^2 = \mathbb{E}_{\pi_D}\Big[\sum_{h=1}^H \mathrm{Var}_{\pi_D}(x, y_{1:h-1})\Big]$.

In addition, we write

$$\epsilon_\theta(x, y_{1:h-1}) = D_{\mathsf{KL}}(\pi_D(\cdot \mid x, y_{1:h-1}) \,\|\, \pi_\theta(\cdot \mid x, y_{1:h-1})). \qquad (57)$$

For any $\theta \in \Theta$, the key quantity of interest is $D_{\mathsf{seq},N}(\pi_D \,\|\, \pi_\theta)$, defined via

$$D_{\mathsf{seq},N}(\pi_D \,\|\, \pi_\theta) = \mathbb{E}_{\pi_D} \min\left\{\log N, \sum_{h=1}^H D_{\mathsf{KL}}(\pi_D(\cdot \mid x, y_{1:h-1}) \,\|\, \pi_\theta(\cdot \mid x, y_{1:h-1}))\right\}$$
$$= \mathbb{E}_{\pi_D} \min\left\{\log N, \sum_{h=1}^H \epsilon_\theta(x, y_{1:h-1})\right\}.$$

By Proposition F.10, it holds that $\mathsf{Cov}_N(\pi_\theta) \leq \frac{2}{\log N - 1} D_{\mathsf{seq},N}(\pi_D \,\|\, \pi_\theta)$.

Further, by concavity, we have

$$\epsilon_\theta(x, y_{1:h-1}) \leq \langle \bar{\phi}_\theta(x, y_{1:h-1}) - \bar{\phi}_{\theta^\star}(x, y_{1:h-1}), \theta - \theta^\star \rangle. \qquad (58)$$

By Lemma H.4, it holds that

$$\|\bar{\phi}_{\theta^\star}(x, y_{1:h-1}) - \bar{\phi}_\theta(x, y_{1:h-1})\| \leq 4\sqrt{\mathrm{Var}_{\pi_D}(x, y_{1:h-1}) \cdot \epsilon_\theta(x, y_{1:h-1})} + 8B\epsilon_\theta(x, y_{1:h-1}). \qquad (59)$$

## K.2 PROOF OF THEOREM 4.2 (COVERAGE FOR MLE FOR AUTOREGRESSIVE LINEAR MODELS)

We prove the following slightly stronger result. Theorem 4.2 follows immediately by combining Theorem K.1 and Proposition F.10.

**Theorem K.1.** *Suppose that [Assumption 2.2](#) holds. Then the MLE $\widehat{\pi}$ achieves*

$$\mathbb{E}_{\mathcal{D}}[D_{\mathsf{seq},N}(\pi_{\mathsf{D}} \,\|\, \widehat{\pi})] \lesssim \sqrt{\frac{\sigma_\star^2 \log N}{n}} + \frac{B^2 \log N}{n},$$

*for any parameter $N \geq 2$, where the divergence $D_{\mathsf{seq},N}(\cdot \,\|\, \cdot)$ is defined in [Proposition F.10](#).*

We begin with two central technical lemmas, which are proven in the sequel. The first lemma is a consequence of the fact that the MLE $\widehat{\pi} = \pi_{\widehat{\theta}}$ maximizes the empirical likelihood, i.e.,

$$\widehat{\theta} = \arg\max_{\theta \in \Theta} \widehat{\mathbb{E}}_{\mathcal{D}}[\log \pi_\theta(y_{1:H} \mid x)], \tag{60}$$

where we recall that for any dataset $\mathcal{D} = \{(x^i, y^i_{1:H})\}_{i \in [n]}$, we write $\widehat{\mathbb{E}}_{\mathcal{D}}[f] := \frac{1}{n} \sum_{i=1}^n f(x^i, y^i_{1:H})$ for any $f : \mathcal{X} \times \mathcal{V}^\star \to \mathbb{R}$. [Lemma K.1](#) shows that in expectation, a sum of per-step conditional KL divergences between $\pi_{\mathsf{D}}$ and $\widehat{\pi}$ is bounded (this does not imply a bound on sequence-level KL divergence, since $\widehat{\theta}$ is dependent on the data $\mathcal{D}$).

**Lemma K.1.** *Recall that we denote $\epsilon_\theta(x, y_{1:h-1}) = D_{\mathsf{KL}}(\pi_{\mathsf{D}}(\cdot \mid x, y_{1:h-1}) \,\|\, \pi_\theta(\cdot \mid x, y_{1:h-1}))$. Further, define*

$$E_1 := \widehat{\mathbb{E}}_{\mathcal{D}}\left[\sum_{h=1}^H \epsilon_{\widehat{\theta}}(x, y_{1:h-1})\right] \tag{61}$$

*Then it holds that $\mathbb{E}[E_1] \leq \frac{2\sigma_\star}{\sqrt{n}}$.*

Define $A := \log N$. The next lemma is a uniform convergence-like argument which shows that the quantity $E_1$ above—when truncated at a certain level $A$—concentrates around its expectation up to a multiplicative factor. This argument is inspired by the *fractional covering* method introduced in [Chen et al. (2024a)](#); [Chen & Rakhlin (2025)](#).

**Lemma K.2.** *Fix any $\Delta \in (0, \frac{1}{200B}]$, $\delta \in (0, 1)$, and let $J = \exp\left(\frac{1}{\Delta^2} + 2\right) \log(1/\delta)$. Let $\Theta' := \{\theta_1, \cdots, \theta_J\}$, where $\theta_1, \cdots, \theta_J \sim \mathcal{N}(0, \Delta^2 I)$ are sampled i.i.d. Then the following holds with probability at least $1 - \delta$ over the randomness of $\Theta'$ and $\mathcal{D}$:*

*(1) For any $j \in [J]$, it holds that*

$$\mathbb{E}_{\pi_{\mathsf{D}}} \min\left\{A, \sum_{h=1}^H \epsilon_{\theta_j}(x, y_{1:h-1})\right\} \leq 2\widehat{\mathbb{E}}_{\mathcal{D}} \min\left\{A, \sum_{h=1}^H \epsilon_{\theta_j}(x, y_{1:h-1})\right\} + \frac{8A \log(4J/\delta)}{n}.$$

*(2) There exists $j \in [J]$ such that*

$$\mathbb{E}_{\pi_{\mathsf{D}}} \min\left\{A, \sum_{h=1}^H \epsilon_{\widehat{\theta}}(x, y_{1:h-1})\right\} \leq 2\,\mathbb{E}_{\pi_{\mathsf{D}}} \min\left\{A, \sum_{h=1}^H \epsilon_{\theta_j}(x, y_{1:h-1})\right\} + C\Delta^2 \sigma_\star^2, \tag{62}$$

*and*

$$\begin{aligned}
\widehat{\mathbb{E}}_{\mathcal{D}} \min\left\{A, \sum_{h=1}^H \epsilon_{\theta_j}(x, y_{1:h-1})\right\} &\leq 2\widehat{\mathbb{E}}_{\mathcal{D}} \min\left\{A, \sum_{h=1}^H \epsilon_{\widehat{\theta}}(x, y_{1:h-1})\right\} \\
&\quad + C\Delta^2 \widehat{\mathbb{E}}_{\mathcal{D}}\left[\sum_{h=1}^H \mathrm{Var}_{\pi_{\mathsf{D}}}(x, y_{1:h-1})\right],
\end{aligned} \tag{63}$$

*where $C = 1000$ is a numeric constant.*

Above, the distribution of $\pi_\theta$ under $\theta \sim \mathcal{N}(0, \Delta^2 I)$ can be viewed as a fractional cover for $\Pi$ in the sense of [Chen et al. (2024a)](#). In particular, working with the fractional cover offers the following technical advantages:

- The fractional cover $\mathcal{N}(0, \Delta^2 I)$ incurs error $\sigma_\star^2 \Delta^2$ (see [Lemma K.3](#)) that depends only on the variance at the ground-truth parameter $\theta^\star$. This contrasts with classical coverings, which enforce a uniform bound for all $\theta \in \Theta$.

- For $\Theta = \mathbb{B}_2^d(1)$, the $L_\infty$ covering number of $\Pi$ (cf. Definition 4.1) scales with the dimension $d$. A standard approach to deriving dimension-independent bounds is to apply symmetrization techniques and use a data-dependent $L_2$ covering to show uniform convergence. In contrast, our fractional-covering approach avoids the (technically subtle) symmetrization step because the cover $\{\theta_1, \ldots, \theta_J\} \sim \mathcal{N}(0, \Delta^2 I)$ is drawn independently of the dataset $\mathcal{D}$.

**Completing the proof.** Equipped with the lemmas above, we complete the proof as follows. First, we condition on the success event $\mathcal{E}$ of Lemma K.2, and let $j \in [J]$ be an index such that (62) and (63) hold. Then, we can upper bound (recall that $A = \log N$ and $D_{\mathsf{seq},N}(\cdot \| \cdot)$ is defined in Proposition F.10)

$$
\begin{aligned}
D_{\mathsf{seq},N}\big(\pi_\mathsf{D} \| \pi_{\widehat{\theta}}\big) &= \mathbb{E}_{\pi_\mathsf{D}} \min\bigg\{ A, \sum_{h=1}^H \epsilon_{\widehat{\theta}}(x, y_{1:h-1}) \bigg\} \\
&\leq 2\,\mathbb{E}_{\pi_\mathsf{D}} \min\bigg\{ A, \sum_{h=1}^H \epsilon_{\theta_j}(x, y_{1:h-1}) \bigg\} + C\Delta^2 \sigma_\star^2 \\
&\leq 4\widehat{\mathbb{E}}_\mathcal{D} \min\bigg\{ A, \sum_{h=1}^H \epsilon_{\theta_j}(x, y_{1:h-1}) \bigg\} + \frac{16A \log(4J/\delta)}{n} + C\Delta^2 \sigma_\star^2 \\
&\leq 8\widehat{\mathbb{E}}_\mathcal{D} \min\bigg\{ A, \sum_{h=1}^H \epsilon_{\widehat{\theta}}(x, y_{1:h-1}) \bigg\} \\
&\quad + 4C\Delta^2 \widehat{\mathbb{E}}_\mathcal{D}\bigg[ \sum_{h=1}^H \mathrm{Var}_{\pi_\mathsf{D}}(x, y_{1:h-1}) \bigg] + \frac{16A \log(4J/\delta)}{n} + C\Delta^2 \sigma_\star^2.
\end{aligned}
$$

where the first inequality uses (62), the second inequality uses Lemma K.2 (1), and the third inequality uses (63). Therefore, we denote $\sigma^2(\mathcal{D}) := \widehat{\mathbb{E}}_\mathcal{D}\big[ \sum_{h=1}^H \mathrm{Var}_{\pi_\mathsf{D}}(x, y_{1:h-1}) \big]$, and we have shown that for any $\delta \in (0,1)$, any $\Delta \in (0, \frac{1}{200B}]$, it holds that

$$
\mathbb{P}_{\mathcal{D} \sim \pi_\mathsf{D}}\bigg( D_{\mathsf{seq},N}\big(\pi_\mathsf{D} \| \pi_{\widehat{\theta}}\big) \geq C_1\bigg( E_1 + \Delta^2 \sigma^2(\mathcal{D}) + \Delta^2 \sigma_\star^2 + \frac{A}{n}\bigg( \frac{1}{\Delta^2} + \log(1/\delta) \bigg) \bigg) \bigg) \leq \delta,
$$

where $C_1 > 0$ is an absolute constant.

Since $\delta \in (0,1)$ is arbitrary, integrating the tail inequality above yields the following bound on the expected value:

$$
\begin{aligned}
\mathbb{E}\big[ D_{\mathsf{seq},N}\big(\pi_\mathsf{D} \| \pi_{\widehat{\theta}}\big) \big] &\leq C_1\bigg( \mathbb{E}[E_1] + \Delta^2\, \mathbb{E}[\sigma^2(\mathcal{D})] + \Delta^2 \sigma_\star^2 + \frac{A}{n}\bigg( \frac{1}{\Delta^2} + 1 \bigg) \bigg) \\
&\leq 2C_1\bigg( \sqrt{\frac{\sigma_\star^2}{n}} + \Delta^2 \sigma_\star^2 + \frac{A}{n\Delta^2} \bigg), \qquad \forall 0 < \Delta \leq \frac{1}{200B}.
\end{aligned}
$$

Choosing $\Delta = \min\bigg\{ \frac{1}{200B}, \big( \frac{A}{\sigma_\star^2 n} \big)^{1/4} \bigg\}$ completes the proof. The coverage upper bound follows immediately from Proposition F.10. $\qquad\square$

### K.2.1 PROOFS FOR SUPPORTING LEMMAS

**Proof of Lemma K.1.** Recall that $\widehat{\pi} = \pi_{\widehat{\theta}}$, where $\widehat{\theta} = \arg\max_{\theta \in \Theta} \widehat{\mathbb{E}}_\mathcal{D}[\log \pi_\theta(y_{1:H} \mid x)]$. Then by concavity of the log-likelihood, we have that

$$
\Big\langle \widehat{\mathbb{E}}_\mathcal{D}\big[ \nabla \log \pi_{\widehat{\theta}}(y_{1:H} \mid x) \big], \theta - \widehat{\theta} \Big\rangle \leq 0, \qquad \forall \theta \in \Theta.
$$

Using the expression (56) and $\theta^\star \in \Theta$, we know

$$
\widehat{\mathbb{E}}_\mathcal{D}\bigg[ \Big\langle \sum_{h=1}^H \big( \phi(x, y_{1:h}) - \bar{\phi}_{\widehat{\theta}}(x, y_{1:h-1}) \big), \theta^\star - \widehat{\theta} \Big\rangle \bigg] \leq 0.
$$

Therefore, combining the inequality above with Eq. (58), we have

$$\widehat{\mathbb{E}}_{\mathcal{D}}\left[\sum_{h=1}^{H}\epsilon_{\widehat{\theta}}(x,y_{1:h-1})\right] = \widehat{\mathbb{E}}_{\mathcal{D}}\left[\sum_{h=1}^{H}D_{\mathsf{KL}}(\pi_{\mathsf{D}}(\cdot\mid x,y_{1:h-1})\,\|\,\widehat{\pi}(\cdot\mid x,y_{1:h-1}))\right]$$

$$\leq \widehat{\mathbb{E}}_{\mathcal{D}}\left[\sum_{h=1}^{H}\left\langle\bar{\phi}_{\theta^{\star}}(x,y_{1:h-1})-\bar{\phi}_{\widehat{\theta}}(x,y_{1:h-1}),\theta^{\star}-\widehat{\theta}\right\rangle\right]$$

$$\leq \widehat{\mathbb{E}}_{\mathcal{D}}\left[\sum_{h=1}^{H}\left\langle\bar{\phi}_{\theta^{\star}}(x,y_{1:h-1})-\phi(x,y_{1:h}),\theta^{\star}-\widehat{\theta}\right\rangle\right]$$

$$\leq 2\left\|\widehat{\mathbb{E}}_{\mathcal{D}}\left[\sum_{h=1}^{H}\phi^{\star}(x,y_{1:h})\right]\right\| =: E_1',$$

where we recall that $\phi^{\star}(x,y_{1:h}) := \phi(x,y_{1:h}) - \bar{\phi}_{\theta^{\star}}(x,y_{1:h-1})$. By definition, it holds that $\mathbb{E}_{\pi_{\mathsf{D}}}[\phi^{\star}(x,y_{1:h})\mid x,y_{1:h-1}] = 0$, and hence

$$\mathbb{E}(E_1')^2 = \mathbb{E}\left\|\widehat{\mathbb{E}}_{\mathcal{D}}\left[\sum_{h=1}^{H}\phi^{\star}(x,y_{1:h})\right]\right\|^2$$

$$= \frac{1}{n}\mathbb{E}_{\pi_{\mathsf{D}}}\left\|\sum_{h=1}^{H}\phi^{\star}(x,y_{1:h})\right\|^2 = \frac{1}{n}\mathbb{E}_{\pi_{\mathsf{D}}}\left[\sum_{h=1}^{H}\|\phi^{\star}(x,y_{1:h})\|^2\right] = \frac{\sigma_{\star}^2}{n}.$$

This gives the desired upper bound. $\qquad\square$

**Proof of Lemma K.2.** By Freedman's inequality (Lemma H.2) and the union bound, it follows that (1) holds with probability at least $1 - \frac{\delta}{2}$. In the remainder of the proof, we prove (2).

Define the following weight function $\alpha = \alpha_{\widehat{\theta}} : \mathcal{X} \times \mathcal{V}^{\star} \to [0,1]$:[9]

$$\alpha_{\widehat{\theta}}(x,y_{1:h-1}) = \begin{cases} 1, & \sum_{j\leq h-1}\epsilon_{\widehat{\theta}}(x,y_{1:j}) \leq A, \\ 0, & \sum_{j<h-1}\epsilon_{\widehat{\theta}}(x,y_{1:j}) \geq A, \\ \frac{A-\sum_{j<h-1}\epsilon_{\widehat{\theta}}(x,y_{1:j})}{\epsilon_{\widehat{\theta}}(x,y_{1:h-1})}, & \text{otherwise.} \end{cases}$$

We also define $\mathsf{F}(a,b) = |a-b| - \frac{1}{2}a$. The properties of $\mathsf{F}(\cdot,\cdot)$ and the weight function $\alpha$ are summarized in Lemma K.4 (stated and proven in the sequel).

Then, by Lemma K.4, it holds that for any $\theta \in \Theta$,

$$\mathbb{E}_{\pi_{\mathsf{D}}}\min\left\{A,\sum_{h=1}^{H}\epsilon_{\widehat{\theta}}(x,y_{1:h-1})\right\} \leq 2\,\mathbb{E}_{\pi_{\mathsf{D}}}\min\left\{A,\sum_{h=1}^{H}\epsilon_{\theta}(x,y_{1:h-1})\right\}$$

$$+ 2\,\mathbb{E}_{\pi_{\mathsf{D}}}\left[\sum_{h=1}^{H}\alpha(x,y_{1:h-1})\mathsf{F}\big(\epsilon_{\widehat{\theta}}(x,y_{1:h-1}),\epsilon_{\theta}(x,y_{1:h-1})\big)\right],$$

and

$$\widehat{\mathbb{E}}_{\mathcal{D}}\min\left\{A,\sum_{h=1}^{H}\epsilon_{\theta}(x,y_{1:h-1})\right\} \leq 2\widehat{\mathbb{E}}_{\mathcal{D}}\min\left\{A,\sum_{h=1}^{H}\epsilon_{\widehat{\theta}}(x,y_{1:h-1})\right\}$$

$$+ \widehat{\mathbb{E}}_{\mathcal{D}}\left[\sum_{h=1}^{H}\alpha(x,y_{1:h-1})\mathsf{F}\big(\epsilon_{\widehat{\theta}}(x,y_{1:h-1}),\epsilon_{\theta}(x,y_{1:h-1})\big)\right],$$

Therefore, it remains to control the error $\sum_{h=1}^{H}\alpha(x,y_{1:h-1})\mathsf{F}\big(\epsilon_{\widehat{\theta}}(x,y_{1:h-1}),\epsilon_{\theta}(x,y_{1:h-1})\big)$ under both $\mathbb{E}_{\pi_{\mathsf{D}}}[\cdot]$ and $\widehat{\mathbb{E}}_{\mathcal{D}}[\cdot]$. We next state the following lemma (proven in the sequel), which leverages

---

[9]Inspired by the analysis here, we also adopt this weight function in the SGD update (35) with the *truncated* stochastic gradient estimator.

the structure of Gaussian distribution. This result can be viewed as a fractional covering number bound (Chen et al., 2024a) and hence generalizes the argument of Chen & Rakhlin (2025, Proposition C.4).

**Lemma K.3.** *For any $K \geq 1$, $\Delta \in (0, \frac{1}{100KB}]$, $\theta \in \mathbb{B}_2(1)$, distributions $\rho_1, \cdots, \rho_K$ over $\mathcal{Z} := \mathcal{X} \times \mathcal{V}^\star$, and weight function $\alpha : \mathcal{Z} \to [0, 1]$, it holds that*

$$- \log \mathbb{P}_{\theta' \sim \mathcal{N}(0,\Delta^2)} \big( \forall i \in [K], \mathbb{E}_{z \sim \rho_i} \alpha(z) \mathsf{F}(\epsilon_\theta(z), \epsilon_{\theta'}(z)) \leq 70K^2 \Delta^2 \, \mathbb{E}_{z \sim \rho_i} \mathrm{Var}_{\pi_\mathsf{D}}(z) \big) \leq \frac{1}{\Delta^2} + 2,$$

*where we recall that $\mathsf{F}(a, b) = |a - b| - \frac{1}{2}a$.*

In the following, we apply Lemma K.3 with $K = 2$, parameter $\theta = \widehat{\theta}$, weight function $\alpha$, and the distributions $\rho_1, \rho_2$ defined as follows:

- Let $\rho_1$ be the distribution of $x' = (x, y_{1:h-1})$ under $x \sim \mu$, $y_{1:H} \sim \pi_\mathsf{D}(\cdot \mid x)$ and $h \sim \mathsf{Unif}([H])$.
- Let $\rho_2$ be the distribution of $x' = (x^t, y_{1:h-1}^t)$ under $t \sim \mathsf{Unif}([n])$ and $h \sim \mathsf{Unif}([H])$.

By definition, it holds that

$$\mathbb{E}_{z \sim \rho_1} \alpha(z) \mathsf{F}(\epsilon_\theta(z), \epsilon_{\theta'}(z)) = \frac{1}{H} \mathbb{E}_{\pi_\mathsf{D}} \left[ \sum_{h=1}^{H} \alpha(x, y_{1:h-1}) \mathsf{F}\big(\epsilon_{\widehat{\theta}}(x, y_{1:h-1}), \epsilon_\theta(x, y_{1:h-1})\big) \right],$$

$$\mathbb{E}_{z \sim \rho_1} \mathrm{Var}_{\pi_\mathsf{D}}(z) = \frac{1}{H} \mathbb{E}_{\pi_\mathsf{D}} \left[ \sum_{h=1}^{H} \mathrm{Var}_{\pi_\mathsf{D}}(x, y_{1:h-1}) \right] = \frac{\sigma_\star^2}{H},$$

$$\mathbb{E}_{z \sim \rho_2} \alpha(z) \mathsf{F}(\epsilon_\theta(z), \epsilon_{\theta'}(z)) = \frac{1}{H} \widehat{\mathbb{E}}_\mathcal{D} \left[ \sum_{h=1}^{H} \alpha(x, y_{1:h-1}) \mathsf{F}\big(\epsilon_{\widehat{\theta}}(x, y_{1:h-1}), \epsilon_\theta(x, y_{1:h-1})\big) \right],$$

$$\mathbb{E}_{z \sim \rho_2} \mathrm{Var}_{\pi_\mathsf{D}}(z) = \frac{1}{H} \widehat{\mathbb{E}}_\mathcal{D} \left[ \sum_{h=1}^{H} \mathrm{Var}_{\pi_\mathsf{D}}(x, y_{1:h-1}) \right].$$

Now, consider the following set for any $\theta \in \Theta$:

$$\Theta_\theta^+ := \big\{ \forall i \in \{1, 2\}, \mathbb{E}_{z \sim \rho_i} \alpha(z) \mathsf{F}(\epsilon_\theta(z), \epsilon_{\theta'}(z)) \leq 300\Delta^2 \, \mathbb{E}_{z \sim \rho_i} \mathrm{Var}_{\pi_\mathsf{D}}(z) \big\}.$$

By Lemma K.3, it holds that

$$q(\theta) := \mathbb{P}_{\theta' \sim \mathcal{N}(0,\Delta^2 I)}(\theta' \in \Theta_\theta^+) \geq \exp\left( -\frac{1}{\Delta^2} - 2 \right), \qquad \forall \theta \in \Theta, \forall \Delta \in (0, \frac{1}{200B}].$$

Therefore, we have

$$\mathbb{P}\left( \forall j \in [J], \theta_j \notin \Theta_{\widehat{\theta}}^+ \mid \widehat{\theta} \right) = \mathbb{P}_{\theta_1, \cdots, \theta_J \sim \mathcal{N}(0,\Delta^2 I)}\left( \forall j \in [J], \theta_j \notin \Theta_{\widehat{\theta}}^+ \right)$$

$$\leq (1 - q(\widehat{\theta}))^J \leq \exp\left( -Jq(\widehat{\theta}) \right) \leq \frac{\delta}{2},$$

and hence $\mathbb{P}\left( \exists j \in [J], \theta_j \in \Theta_{\widehat{\theta}}^+ \right) \geq 1 - \frac{\delta}{2}$. The proof of Lemma K.2 (2) is thus completed, as Eq. (62) and Eq. (63) hold for any $j \in [J]$ such that $\theta_j \in \Theta_{\widehat{\theta}}^+$. $\qquad \square$

**Proof of Lemma K.3.** We first fix any $h \in [H]$ and $z = (x, y_{1:h-1}) \in \mathcal{X} \times \mathcal{V}^{h-1}$ and analyze the behavior of $\log \pi_{\theta'}(y_h \mid z)$ under $\theta' \sim \mathcal{N}(\theta, \Delta^2 I)$.

By definition, we have $\pi_{\theta'}(y_h \mid z) \propto_{y_h} \pi_\theta(y_h \mid z) \cdot \exp(\langle \theta' - \theta, \phi(z, y_h) \rangle)$, i.e.,

$$\log \pi_{\theta'}(y_h \mid z) - \log \pi_\theta(y_h \mid z) = \langle \theta' - \theta, \phi(z, y_h) \rangle - \log \mathbb{E}_{y_h \sim \pi_\theta(\cdot|z)} \exp(\langle \theta' - \theta, \phi(z, y_h) \rangle).$$

Therefore,

$$\epsilon_\theta(z) - \epsilon_{\theta'}(z) = D_{\mathsf{KL}}(\pi_\mathsf{D}(y_h = \cdot \mid z) \,\|\, \pi_\theta(y_h = \cdot \mid z)) - D_{\mathsf{KL}}(\pi_\mathsf{D}(y_h = \cdot \mid z) \,\|\, \pi_{\theta'}(y_h = \cdot \mid z))$$

$$= \mathbb{E}_{\pi_\mathsf{D}(\cdot|z)} \langle \theta' - \theta, \phi(z, y_h) \rangle - \log \mathbb{E}_{y_h \sim \pi_\theta(\cdot|z)} \exp(\langle \theta' - \theta, \phi(z, y_h) \rangle)$$

$$= \langle \theta' - \theta, \bar{\phi}_{\theta^\star}(z) - \bar{\phi}_\theta(z) \rangle - \log \mathbb{E}_{y_h \sim \pi_\theta(\cdot|z)} \exp\big(\langle \theta' - \theta, \phi(z, y_h) - \bar{\phi}_\theta(z) \rangle\big),$$

where we recall that $\bar{\phi}_\theta(z) = \mathbb{E}_{y_h \sim \pi_\theta(\cdot|z)}[\phi(z, y_h)]$.

In the following, we denote $\phi_\theta(z, y_h) := \phi(z, y_h) - \bar{\phi}_\theta(z)$, and

$$E_{\theta'}^+(z) := \log \mathbb{E}_{y_h \sim \pi_\theta(\cdot|z)} \exp(\langle \theta' - \theta, \phi_\theta(z, y_h) \rangle),$$
$$E_{\theta'}^-(z) := \langle \theta' - \theta, \bar{\phi}_{\theta^\star}(z) - \bar{\phi}_\theta(z) \rangle.$$

We first bound $E_{\theta'}^+(z)$. By definition, we have $E_{\theta'}^+(z) = D_{\mathsf{KL}}(\pi_\theta(\cdot \mid z) \| \pi_{\theta'}(\cdot \mid z)) \geq 0$. Further, using Jensen's inequality, for any $z \in \mathcal{Z}$, we have

$$\mathbb{E}_{\theta' \sim \mathcal{N}(\theta, \Delta^2 I)}\left[ E_{\theta'}^+(z) \right] \leq \log \mathbb{E}_{\theta' \sim \mathcal{N}(\theta, \Delta^2 I)} \mathbb{E}_{y_h \sim \pi_\theta(\cdot|z)}[\exp(\langle \theta' - \theta, \phi_\theta(z, y_h) \rangle)]$$
$$= \log \mathbb{E}_{y_h \sim \pi_\theta(\cdot|z)} \exp\left( \frac{1}{2}\Delta^2 \|\phi_\theta(z, y_h)\|^2 \right)$$
$$\leq \Delta^2 \, \mathbb{E}_{y_h \sim \pi_\theta(\cdot|z)} \|\phi_\theta(z, y_h)\|^2,$$

where the last inequality follows from $e^t \leq 1 + 2t$ for $t \in [0, 1]$. Further, using Lemma H.4, we have

$$\mathbb{E}_{y_h \sim \pi_\theta(\cdot|z)} \|\phi_\theta(z, y_h)\|^2 = \mathbb{E}_{y_h \sim \pi_\theta(\cdot|z)} \|\phi(z, y_h) - \phi_\theta(z)\|^2$$
$$\leq 3\, \mathbb{E}_{y \sim \pi_D(\cdot|z)} \|\phi(z, y_h) - \phi_{\theta^\star}(z)\|^2 + 16B^2 D_{\mathsf{KL}}(\pi_D(\cdot \mid z) \| \pi_\theta(\cdot \mid z))$$
$$= 3\mathrm{Var}_{\pi_D}(z) + 16B^2 \epsilon_\theta(z).$$

Next, we bound $|E_{\theta'}^-(z)|$. Under $\theta' \sim \mathcal{N}(\theta, \Delta^2 I)$, it is clear that $\langle \theta' - \theta, \bar{\phi}_{\theta^\star}(z) - \bar{\phi}_\theta(z) \rangle \sim \mathcal{N}(0, \Delta^2 \|\bar{\phi}_{\theta^\star}(z) - \bar{\phi}_\theta(z)\|^2)$ for any fixed $z$. Therefore, it holds that

$$\mathbb{E}_{\theta' \sim \mathcal{N}(\theta, \Delta^2 I)} |E_{\theta'}^-(z)| = \sqrt{\frac{2}{\pi}} \Delta \cdot \|\bar{\phi}_{\theta^\star}(z) - \bar{\phi}_\theta(z)\|$$
$$\leq \Delta \cdot \left( 4\sqrt{\mathrm{Var}_{\pi_D}(z) \cdot \epsilon_\theta(z)} + 8B\epsilon_\theta(z) \right)$$
$$\leq \left( \frac{1}{8K} + 8B\Delta \right) \epsilon_\theta(z) + 32K\Delta^2 \mathrm{Var}_{\pi_D}(z),$$

where the second line uses Eq. (59).

Combining the inequalities above and taking expectation of $z \sim \rho_i$, we know that for $i \in [K]$, it holds that

$$\mathbb{E}_{\theta' \sim \mathcal{N}(\theta, \Delta^2 I)}\left[ \mathbb{E}_{z \sim \rho_i}\left[ \alpha(z) E_{\theta'}^+(z) \right] \right] \leq \Delta^2 \, \mathbb{E}_{z \sim \rho_i}\left[ 3\mathrm{Var}_{\pi_D}(z) + 16B^2 \alpha(z)\epsilon_\theta(z) \right],$$
$$\mathbb{E}_{\theta' \sim \mathcal{N}(\theta, \Delta^2 I)}\left[ \mathbb{E}_{z \sim \rho_i}\left[ \alpha(z) |E_{\theta'}^-(z)| \right] \right] \leq \mathbb{E}_{z \sim \rho_i}\left[ 32K\Delta^2 \mathrm{Var}_{\pi_D}(z) + \left( \frac{1}{8K} + 8B\Delta \right)\alpha(z)\epsilon_\theta(z) \right],$$

and hence by Markov's inequality and $\Delta \leq \frac{1}{100KB}$, it holds that $p := \mathbb{P}_{\theta' \sim \mathcal{N}(\theta, \Delta^2 I)}(\theta' \notin \Theta^-) \geq \frac{1}{2}$, where we denote $\Theta^- = \cup_{i \in [K]}\Theta_i^-$, and

$$\Theta_i^- := \left\{ \theta' \in \mathbb{R}^d : \mathbb{E}_{z \sim \rho_i} \alpha(z)|\epsilon_\theta(z) - \epsilon_{\theta'}(z)| \geq \mathbb{E}_{z \sim \rho_i}\left[ (6K + 64K^2)\Delta^2 \mathrm{Var}_{\pi_D}(z) + \frac{1}{2}\alpha(z)\epsilon_\theta(z) \right] \right\}.$$

Note that $D_{\mathsf{KL}}\left( \mathcal{N}(\theta, \Delta^2 I) \| \mathcal{N}(0, \Delta^2 I) \right) = \frac{\|\theta\|^2}{2\Delta^2} \leq \frac{1}{2\Delta^2}$. Hence, by data-processing inequality, we can bound $q := \mathbb{P}_{\theta' \sim \mathcal{N}(0, \Delta^2 I)}(\theta' \notin \Theta^-)$ as

$$\frac{1}{2\Delta^2} \geq D_{\mathsf{KL}}\left( \mathcal{N}(\theta, \Delta^2 I) \| \mathcal{N}(0, \Delta^2 I) \right) \geq D_{\mathsf{KL}}(\mathrm{Ber}(p) \| \mathrm{Ber}(q))$$
$$= p \log \frac{p}{q} + (1-p) \log \frac{1-p}{1-q} \geq \frac{1}{2} \log(1/q) - \log 2.$$

This implies that $-\log q \leq \frac{1}{\Delta^2} + 2$, giving the desired result. $\qquad\square$

**Lemma K.4.** *Suppose that* $a_1, \cdots, a_H, b_1, \cdots, b_H \geq 0, A \geq 0$. *Define* $\mathsf{F}(a, b) = |a - b| - \frac{1}{2}a$. *Let*

$$
\alpha_h = \begin{cases} 1, & \sum_{j \leq h} a_j \leq A, \\ 0, & \sum_{j < h} a_j > A, \\ \frac{A - \sum_{j < h} a_j}{a_h}, & \textit{otherwise.} \end{cases}
$$

*Then clearly* $\alpha_h \in [0, 1] \ \forall h \in [H]$, *and it holds that* $\sum_{h=1}^{H} \alpha_h a_h = \min\left\{A, \sum_{h=1}^{H} a_h\right\}$, *and*

$$
\min\left\{A, \sum_{h=1}^{H} a_h\right\} \leq 2\min\left\{A, \sum_{h=1}^{H} b_h\right\} + 2\sum_{h=1}^{H} \alpha_h \mathsf{F}(a_h, b_h),
$$

*and*

$$
\min\left\{A, \sum_{h=1}^{H} b_h\right\} \leq 2\min\left\{A, \sum_{h=1}^{H} a_h\right\} + \sum_{h=1}^{H} \alpha_h \mathsf{F}(a_h, b_h).
$$

**Proof of Lemma K.4.** Fix the sequence $a_1, \cdots, a_H$. We first prove that

$$
\sum_{h=1}^{H} \alpha_h a_h = \min\left\{A, \sum_{h=1}^{H} a_h\right\}. \tag{64}
$$

To do so, we consider two cases.

**Case 1:** $\sum_{h=1}^{H} a_h \leq A$**.** In this case, $\alpha_h = 1 \forall h \in [H]$, and the equation holds trivially.

**Case 2:** $\sum_{h=1}^{H} a_h > A$**.** In this case, we let $\ell \in [H]$ be the maximal index such that $\alpha_\ell > 0$. Then, by definition, $\sum_{j < \ell} a_j \leq A$ and $\sum_{j \leq \ell} a_j > A$, and $\alpha_\ell = \frac{A - \sum_{j < \ell} a_j}{a_\ell}$. Hence,

$$
\sum_{h=1}^{H} \alpha_h a_h = \sum_{h=1}^{\ell} \alpha_h a_h = \sum_{j < \ell} a_j + \alpha_\ell a_\ell = A.
$$

We also note that from the proof above, we also know that for any sequence $(c_1, \cdots, c_H)$ such that $c_h \geq a_h$ for $h \in [H]$, we have

$$
\min\left\{A, \sum_{h=1}^{H} c_h\right\} \leq \sum_{h=1}^{H} \alpha_h c_h. \tag{65}
$$

Equipped with these results, we prove the inequalities in the lemma statement. We note that

$$
\sum_{h=1}^{H} \alpha_h \mathsf{F}(a_h, b_h) = \sum_{h=1}^{H} \alpha_h |a_h - b_h| - \frac{1}{2}\sum_{h=1}^{H} \alpha_h a_h,
$$

or equivalently,

$$
\sum_{h=1}^{H} \alpha_h |a_h - b_h| = \sum_{h=1}^{H} \alpha_h \mathsf{F}(a_h, b_h) + \frac{1}{2}\min\left\{A, \sum_{h=1}^{H} a_h\right\}.
$$

Therefore,

$$
\begin{aligned}
\min\left\{A, \sum_{h=1}^{H} a_h\right\} = \sum_{h=1}^{H} \alpha_h a_h &\leq \min\left\{A, \sum_{h=1}^{H} b_h\right\} + \sum_{h=1}^{H} \alpha_h |a_h - b_h| \\
&= \min\left\{A, \sum_{h=1}^{H} b_h\right\} + \sum_{h=1}^{H} \alpha_h \mathsf{F}(a_h, b_h) + \frac{1}{2}\min\left\{A, \sum_{h=1}^{H} a_h\right\}.
\end{aligned}
$$

Re-organizing yields the first inequality. Similarly, we have

$$\min\left\{A, \sum_{h=1}^{H} b_h\right\} \leq \min\left\{A, \sum_{h=1}^{H}(a_h + |a_h - b_h|)\right\} \leq \sum_{h=1}^{H} \alpha_h(a_h + |a_h - b_h|)$$

$$= \frac{3}{2}\min\left\{A, \sum_{h=1}^{H} a_h\right\} + \sum_{h=1}^{H} \alpha_h \mathsf{F}(a_h, b_h).$$

The proof is hence completed. □

### K.3   PROOF OF PROPOSITION 5.1 (VANILLA SGD: COVERAGE UPPER BOUND)

We first invoke the following standard lemma.

**Lemma K.5.** *Suppose that the sequence* $(\theta^t, g^t)_{t\geq 1}$ *satisfies* $\theta^{t+1} = \mathrm{Proj}_\Theta(\theta^t + \eta g^t)$ *for* $t \geq 1$. *Then it holds that for any* $\theta^\star \in \Theta$, $T \geq 1$,

$$\sum_{t=1}^{T}\langle -g^t, \theta^t - \theta^\star\rangle \leq \frac{\|\theta^\star - \theta^0\|^2}{2\eta} + \frac{\eta}{2}\sum_{t=1}^{T}\|g^t\|^2. \tag{66}$$

Specializing Lemma K.5 to the SGD update (8) and taking expectation, we have

$$\mathbb{E}\left[\sum_{t=1}^{T}\langle -\nabla \log \pi_{\theta^t}(y^t \mid x^t), \theta^t - \theta^\star\rangle\right] \leq \frac{2}{\eta} + \frac{\eta}{2}\mathbb{E}\left[\sum_{t=1}^{T}\|\nabla \log \pi_{\theta^t}(y^t \mid x^t)\|^2\right]. \tag{67}$$

Note that $(x^t, y^t) \mid \theta^t \sim \pi_\mathsf{D}$, and hence

$$\mathbb{E}[\nabla \log \pi_{\theta^t}(y^t \mid x^t) \mid \theta^t] = \mathbb{E}_{(x,y)\sim\pi_\mathsf{D}}[\nabla \log \pi_{\theta^t}(y \mid x)] = \nabla_\theta D_{\mathsf{KL}}(\pi_\mathsf{D} \| \pi_\theta)|_{\theta=\theta^t}.$$

Further, by convexity, it holds that for any $\theta \in \Theta$,

$$G(\theta) := \mathbb{E}_{\pi_\mathsf{D}}[\langle -\nabla \log \pi_\theta(y \mid x), \theta - \theta^\star\rangle] = \langle \nabla_\theta D_{\mathsf{KL}}(\pi_\mathsf{D} \| \pi_\theta), \theta - \theta^\star\rangle \geq D_{\mathsf{KL}}(\pi_\mathsf{D} \| \pi_\theta).$$

Therefore, we have

$$\mathbb{E}\left[\sum_{t=1}^{T} D_{\mathsf{KL}}(\pi_\mathsf{D} \| \pi_{\theta^t})\right] \leq \mathbb{E}\left[\sum_{t=1}^{T} G(\theta^t)\right] \leq \frac{2}{\eta} + \frac{\eta}{2}\mathbb{E}\left[\sum_{t=1}^{T}\mathbb{E}_{(x,y)\sim\pi_\mathsf{D}}\|\nabla \log \pi_{\theta^t}(y \mid x)\|^2\right].$$

On the other hand, using the fact that $\log \pi_\theta(y \mid x)$ is concave and $(HB^2)$-smooth (i.e., $-HB^2 I \preceq \nabla^2 \log \pi_\theta(y \mid x) \preceq 0$),

$$\|\nabla \log \pi_\theta(y \mid x) - \nabla \log \pi_{\theta^\star}(y \mid x)\|^2 \leq HB^2 \cdot \langle \theta - \theta^\star, \nabla \log \pi_{\theta^\star}(y \mid x) - \nabla \log \pi_\theta(y \mid x)\rangle$$

Taking expectation of $(x, y) \sim \pi_\mathsf{D}$ and using the fact that $\mathbb{E}_{\pi_\mathsf{D}}[\nabla \log \pi_{\theta^\star}(y \mid x)] = 0$, we have

$$\mathbb{E}_{\pi_\mathsf{D}}\|\nabla \log \pi_\theta(y \mid x) - \nabla \log \pi_{\theta^\star}(y \mid x)\|^2 \leq HB^2 \cdot G(\theta), \qquad \forall\theta \in \Theta.$$

Further, note that $\mathbb{E}_{\pi_\mathsf{D}}\|\nabla \log \pi_{\theta^\star}(y \mid x)\|^2 = \sigma_\star^2$, it holds that

$$\mathbb{E}_{\pi_\mathsf{D}}\|\nabla \log \pi_\theta(y \mid x)\|^2 \leq 2\sigma_\star^2 + 2HB^2 \cdot G(\theta), \qquad \forall\theta \in \Theta. \tag{68}$$

Combining the inequalities above, we can conclude that

$$\mathbb{E}\left[\sum_{t=1}^{T} G(\theta^t)\right] \leq \frac{2}{\eta} + \eta HB^2 \mathbb{E}\left[\sum_{t=1}^{T} G(\theta^t)\right] + \eta T\sigma_\star^2.$$

We conclude that as long as $\eta \leq \frac{1}{2HB^2}$, it holds

$$\frac{4}{\eta} + 2\eta T\sigma_\star^2 \geq \mathbb{E}\left[\sum_{t=1}^{T} G(\theta^t)\right] \geq \mathbb{E}\left[\sum_{t=1}^{T} D_{\mathsf{KL}}(\pi_\mathsf{D} \| \pi_{\theta^t})\right].$$

This is the desired upper bound. □

**Proof of Lemma K.5.** A standard result (e.g., Hazan (2016)) is that because the projection operator $\text{Proj}_\Theta$ is an contraction, we have that for all $t \in [T]$, the update satisfies

$$
\begin{aligned}
\|\theta^t - \theta^\star\|^2 &- \|\theta^{t+1} - \theta^\star\|^2 \\
&\geq \|\theta^t - \theta^\star\|^2 - \|\theta^t + \eta g^t - \theta^\star\|^2 \\
&= 2\eta\langle -g^t, \theta^t - \theta^\star\rangle - \eta^2\|g^t\|^2.
\end{aligned}
\tag{69}
$$

Summing this inequality across steps $t = 1, 2, \cdots, T$, telescoping, and taking expectation, we have

$$
\sum_{t=1}^T \langle -g^t, \theta^t - \theta^\star\rangle \leq \frac{\|\theta^\star - \theta^0\|^2 - \|\theta^\star - \theta^{T+1}\|^2}{2\eta} + \frac{\eta}{2}\sum_{t=1}^T \|g^t\|^2.
\tag{70}
$$

This gives the desired upper bound. $\qquad\square$

### K.4 PROOF OF PROPOSITION 5.1 (VANILLA SGD: COVERAGE LOWER BOUND)

In the following, we construct $\mathcal{X} = [\frac{8}{HB}, +\infty) \sqcup \{-, +\}$, $\mathcal{V} = \{-1, 0, 1\}$ and $\Theta = \mathbb{B}_2(1)$ with $d = 2$. We fix parameters $B \geq \bar{B} \geq 1$.

**Construction of $\phi$.** We first construct a map $v : \mathcal{X} \times \mathcal{V} \to \mathbb{R}^2$ as follows. For any $\eta \geq \frac{8}{HB}$, we define $\alpha_\eta = \frac{\eta HB}{2(\eta HB - 1)} \leq \frac{5}{8}$ and let

$$
v(\eta, 0) = [1; 0], \qquad v(\eta, 1) = [\alpha_\eta; \sqrt{1 - \alpha_\eta^2}], \qquad v(\eta, -1) = [\alpha_\eta; -\sqrt{1 - \alpha_\eta^2}].
$$

We further define

$$
v(+, a) = \frac{1}{B}[\bar{B}a; 0], \qquad v(-, a) = \frac{1}{B}[0; \bar{B}a] \qquad \forall a \in \mathcal{V} = \{-1, 0, 1\}.
$$

For $x \in \mathcal{X}, y_{1:h} \in \mathcal{V}^h$, we define $\phi(x, y_{1:h}) = Bv(x, y_h)$.[10]

Under this construction of $\phi$, we then prove the lower bound by considering two cases based on the value of $\eta$.

**Lemma K.6.** *Suppose that $\eta \geq \frac{8}{HB}$, $\log N \leq \frac{HB}{8}$, and $B \geq c_B \log(TH)$ for a large constant $c_B > 1$. Then, with the distribution $\mu$ being supported on $x = \eta$ and $\theta^\star = [1; 0]$, the following holds.*

*(1) The variance of such an instance is bounded: $\sigma_\star \leq 1$.*

*(2) There exists $\theta^0 \in \Theta$ such that with probability at least $0.5$, the SGD sequence $(\theta^t)$ satisfies $\text{Cov}_N(\pi_{\theta^t}) \geq 1 - \frac{1}{2T}$ for all $t \in [T]$.*

**Lemma K.7.** *Suppose that $\eta \leq \frac{8}{HB}$, $\log N \leq \frac{HB}{8}$, and $B \geq \bar{B} \geq c_B \log(TH)$ for a large constant $c_B > 1$. Then, there exists distribution $\mu$ and $\theta^\star \in \Theta$ such that the following holds.*

*(1) The variance of such an instance is bounded: $\sigma_\star \leq 1$.*

*(2) There exists $\theta^0 \in \Theta$ such that with probability at least $0.5$, the SGD sequence $(\theta^t)$ satisfies*

$$
\text{Cov}_N(\pi_{\theta^t}) \geq c \min\left\{1, \frac{HB}{T \cdot \bar{B}^2 \log N}\right\}, \qquad \forall t \in [T].
$$

The proof of Proposition 5.1 (lower bound) is then completed by combining Lemma K.6 and Lemma K.7. $\qquad\square$

**Proof of Lemma K.6.** Fix the parameter $\eta \geq \frac{8}{HB}$. We denote $\bar{\eta} := \eta \cdot HB$ and $\alpha = \alpha_\eta = \frac{\bar{\eta}}{2(\bar{\eta} - 1)} \leq \frac{5}{8}$. Denote

$$
v_0 = [1; 0], \qquad v_1 = [\alpha; \sqrt{1 - \alpha^2}], \qquad v_{-1} = [\alpha; -\sqrt{1 - \alpha^2}].
$$

---

[10]In other words, for any $\theta \in \Theta$, $y_{1:H} \sim \pi_\theta(\cdot \mid x)$ are sampled i.i.d. with $y \sim P_\theta(\cdot \mid x)$, where $P_\theta$ is defined as $P_\theta(a \mid x) = \frac{\exp(B\langle v(x,a),\theta\rangle)}{\sum_{a' \in \mathcal{V}} \exp(B\langle v(x,a'),\theta\rangle)}$.

Under our construction, we have

$$\pi_\theta(y_h \mid \eta, y_{1:h-1}) = \frac{\exp(B\langle\theta, v_{y_h}\rangle)}{\sum_{a\in\mathcal{V}} \exp(B\langle\theta, v_a\rangle)} =: P_\theta(y_h).$$

We study the SGD update starting from $\theta^0 = v_1$. By definition, $\phi(\eta, y_{1:h}) = Bv(\eta, y_h)$, and hence

$$\nabla \log \pi_\theta(y_{1:H} \mid \eta) = \sum_{h=1}^{H}\left( Bv(\eta, y_h) - \mathop{\mathbb{E}}_{a\sim P_\theta}[Bv(\eta, a)] \right) = B\sum_{h=1}^{H}\left( v_{y_h} - \mathop{\mathbb{E}}_{a\sim P_\theta}[v_a] \right).$$

In the following, we denote

$$\widehat{F}(y_{1:H}) := \frac{1}{H}\sum_{h=1}^{H} v_{y_h}, \qquad F(\theta) := \mathop{\mathbb{E}}_{a\sim\pi_\theta}[v_a] = \frac{\sum_{a\in\mathcal{V}} a\exp(B\langle\theta, v_a\rangle)}{\sum_{a\in\mathcal{V}}\exp(B\langle\theta, v_a\rangle)}.$$

Then, the SGD update can be written as

$$u^t = \theta^t + \bar{\eta}\left(\widehat{F}(y_{1:H}^t) - F(\theta^t)\right), \qquad \theta^{t+1} = \mathrm{Proj}_\Theta(u^t).$$

We make the following claims.

**Claim 1.** For $a \in \{-1, 0, 1\}$ and $\|\theta - v_a\| \leq \frac{1}{16}$, it holds that $1 - P_\theta(a) \leq 2e^{-B/4} =: \epsilon_1$ and hence $\|F(\theta) - v_a\| \leq 2\epsilon_1$.

**Claim 2.** Suppose that $\epsilon_1 \leq \min\left\{\frac{1}{4TH}, \frac{1}{5HB^2}\right\}$. Then it holds that $\sigma_\star \leq 1$. Further, with probability at least 0.5, it holds that $\widehat{F}(y_{1:H}^t) = e_0$ for all $t \in [T]$.

In the following, we condition on this event.

**Claim 3.** By definition, for $a \in \{-1, 1\}$, we have $\|v_a + \bar{\eta}(v_0 - v_a)\| = \bar{\eta} - 1$ and $v_{-1} + v_1 = \frac{\bar{\eta}}{\bar{\eta}-1}v_0$.

**Claim 4.** Let $\epsilon = 16\epsilon_1$. Suppose that $\epsilon \leq \frac{1}{16}$. Then for $a \in \{-1, 1\}$, if $\|\theta^t - v_a\| \leq \epsilon$, then it holds that $\|\theta^{t+1} - v_{-a}\| \leq \epsilon$.

**Claim 5.** Suppose that $\epsilon_1 \leq \frac{1}{2TH}$ and $\log N \leq \frac{HB}{8}$. Then $\mathrm{Cov}_N(\pi_\mathsf{D} \| \pi_\theta) \geq 1 - \frac{1}{2T}$ for $\theta \in \Theta$ such that $\min\{\|\theta - v_1\|, \|\theta - v_{-1}\|\} \leq \frac{1}{16}$.

Combining the above claims, we know that there is a constant $C$ such that as long as $B \geq c_B \log(TH)$, it holds that $\sigma_\star \leq 1$. Further, under the success event of claim 2, it holds that for $a \in \{-1, 1\}$, $\|\theta^t - v_a\| \leq \frac{1}{16}$ for all $t \in [T]$ such that $2 \mid t - a$. Therefore, by Claim 5, this gives $\mathrm{Cov}_N(\pi_{\theta^t}) \geq \frac{1}{2}$ as long as $\log N \leq \frac{HB}{8}$. $\qquad\square$

**Proof for Claims 1-5.** To prove Claim 1, we note that $\langle\theta, v_a\rangle \geq 1 - \|\theta - v_a\| \geq \frac{15}{16}$ and for $i \neq a$, $\langle\theta, v_i\rangle \leq \langle v_a, v_i\rangle + \|\theta - v_a\| \leq \alpha + \frac{1}{16} \leq \frac{11}{16}$. Therefore,

$$1 - P_\theta(a) \leq \frac{\sum_{i\neq a} e^{B\langle\theta, v_i\rangle}}{e^{B\langle\theta, v_a\rangle}} \leq \frac{2}{e^{B/4}} = \epsilon_1.$$

This completes the proof of Claim 1.

Next, we prove Claim 2. Recall that $\theta^\star = [1; 0] = v_0$. By Claim 1, we know $1 - P_{\theta^\star}(0) \leq \epsilon_1$, and hence $\mathrm{Var}_{a\sim P_{\theta^\star}}[v_a] \leq 5\epsilon_1$. This implies $\sigma_\star^2 = HB^2\mathrm{Var}_{a\sim P_{\theta^\star}}[v_a] \leq 5HB^2\epsilon_1 \leq 1$.

We also know $\mathbb{P}_{\pi_\mathsf{D}}(y_h = 0 \ \forall h \in [H]) = P_{\theta^\star}(0)^H \geq (1 - \epsilon_1)^H \geq 1 - H\epsilon_1$. Therefore, taking the union bound, we know $\mathbb{P}(y_h^t = 0 \ \forall h \in [H], t \in [T]) \geq 1 - TH\epsilon_1 \geq \frac{1}{2}$. This completes the proof of Claim 2.

Furthermore, for any $\theta$ such that $\min\{\|\theta - v_1\|, \|\theta - v_{-1}\|\} \leq \frac{1}{16}$, as long as $\log N \leq H(\log(1 - \epsilon_1) - \log(\epsilon_1))$, we have

$$\mathrm{Cov}_N(\pi_\mathsf{D} \| \pi_\theta) \geq \left(1 - \frac{1}{2n}\right)\mathbb{I}\{H\log\pi_\mathsf{D}(0) - H\log\pi_\theta(0) \geq \log N\} \geq 1 - \frac{1}{2n}.$$

In particular, this is ensured when $\log N \leq \frac{HB}{8}$. This completes the proof of Claim 5.

Claim 3 follows immediately from the definition of $\alpha$, $v_0$, $v_1$ and $v_{-1}$.

Finally, we prove Claim 4. Recall that $u^t := \theta^t + \bar{\eta}\Big(\widehat{F}(y_{1:H}^t) - F(\theta^t)\Big)$. Then it holds that

$$\|u^t - (\bar{\eta}-1)v_{-a}\| = \|u^t - \bar{\eta}v_0 + (\bar{\eta}-1)v_a\| \leq \|\theta^t - v_a\| + \bar{\eta}\|\widehat{F}(y_{1:H}^t) - v_0\| + \bar{\eta}\|F(\theta^t) - v_a\|$$
$$\leq \epsilon + 2\bar{\eta}\epsilon_1 =: \epsilon'.$$

In particular, it holds that $|\|u^t\| - (\bar{\eta}-1)| \leq \epsilon'$ and hence $\|u^t\| \geq \bar{\eta} - 1 - \epsilon' = (1-2\epsilon_1)\bar{\eta} - 1 - \epsilon \geq \frac{\bar{\eta}}{2} \geq 1$. Therefore, $\theta^{t+1} = \text{Proj}_\Theta(u^t) = \frac{u^t}{\|u^t\|}$, and we can bound

$$\|\theta^{t+1} - v_{-a}\| = \left\| \frac{u^t - (\bar{\eta}-1)v_{-a}}{\|u^t\|} + v_{-a}\left(\frac{\bar{\eta}-1}{\|u^t\|} - 1\right) \right\|$$
$$\leq \frac{\|u^t - (\bar{\eta}-1)v_{-a}\|}{\|u^t\|} + \frac{|\bar{\eta}-1 - \|u^t\||}{\|u^t\|}$$
$$\leq \frac{2\epsilon'}{\|u^t\|} \leq \frac{4\epsilon'}{\bar{\eta}} = \frac{4}{\bar{\eta}}\epsilon + 8\epsilon_1 \leq \epsilon.$$

$\square$

**Proof of Lemma K.7.** We again denote $\bar{\eta} = HB\eta \leq 8$. We choose $\theta^\star = [\frac{1}{2}; \frac{1}{2}]$, and let the distribution $\mu$ be supported on $\{-, +\}$:

$$\mu(+) = 1 - \mu(-) = \min\left\{1, \frac{BH}{512en\bar{B}^2 \log N}\right\}.$$

Note that for $x \in \{-, +\}$, $\pi_0(1 \mid x) = \frac{e^{\bar{B}/2}}{e^{-\bar{B}/2}+1+e^{\bar{B}/2}}$, and hence $1 - \pi_0(y_1 = 1 \mid x) \leq 2e^{-\bar{B}/2}$. Therefore, similar to Case 1, we have the following claims.

**Claim 1.** Suppose that $\bar{B} \geq c_B \log(TH)$ for a large constant $c_B > 0$. Then it holds that $\sigma_\star \leq 1$, and with probability at least 0.5, it holds that $\sum_{t=1}^T \mathbb{I}\{x^t = +\} \leq 4T\mu(1)$, and $y_h^t = 1$ for all $h \in [H], t \in [T]$.

In the following, we condition on this event. We choose $r \leq \frac{1}{2}$ such that $e^{r\bar{B}} = \frac{H}{4\log N}$, and we let $\theta^0 = [r - \frac{1}{\bar{B}}; \frac{1}{4}]$.

**Claim 2.** For any $\theta \in \Theta \subset \mathbb{R}^2$, it holds that $1 - P_\theta(1 \mid +) \leq \frac{2}{e^{\theta[1]\bar{B}}}$ (where $w[1]$ denotes the first coordinate of a vector $w \in \mathbb{R}^2$). Hence, when $x^t = +$, using $y_h^t \equiv 1$, we have $\nabla \log \pi_\theta(y^t \mid x^t)[2] = 0$ and

$$0 \leq \nabla \log \pi_\theta(y^t \mid x^t)[1] = H\bar{B}(1 - \mathbb{E}_{a \sim P_\theta(\cdot|+)}[a]) \leq 2H\bar{B}(1 - P_\theta(1 \mid +)) \leq \frac{4H\bar{B}}{e^{\theta[1]\bar{B}}}.$$

Similarly, when $x^t = -$, we have

$$\nabla \log \pi_\theta(y^t \mid x^t)[1] = 0, \qquad 0 \leq \nabla \log \pi_\theta(y^t \mid x^t)[2] \leq \frac{4H\bar{B}}{e^{\theta[2]\bar{B}}}.$$

Then, combining the inequalities above with Claim 1, we can inductively show that for any $t \in [T]$,

$$\theta^t[1] - \theta^0[1] \leq \sum_{t=1}^T \mathbb{I}\{x^t = +\} \cdot \frac{4\eta H\bar{B}}{e^{\theta^0[1]\bar{B}}} \leq T \cdot \mu(+)\frac{16e\bar{\eta}\bar{B}}{Be^{r\bar{B}}} \leq \mu(+) \cdot \frac{512e\bar{B}n\log N}{BH} \leq \frac{1}{\bar{B}}.$$

Therefore, we have $\theta^t[1] \leq r$ for $t \leq T$. It remains to prove the following claim.

**Claim 3.** Suppose that $e^{\theta[1]\bar{B}} \leq \frac{H}{4\log N}$. Then it holds that $\text{Cov}_N(\pi_\theta) \geq \frac{\mu(+)}{2}$.

To prove Claim 3, we note that similar to Claim 2, $\mathbb{P}_{\pi_\mathsf{D}}(y_h = 1 \ \forall h \in [H] \mid x = +) \geq \frac{1}{2}$. Further, $\log \pi_\mathsf{D}(y_1 = 1 \mid +) \geq \log(1 - 2e^{-\overline{B}/2}) \geq -3e^{-\overline{B}/2}$ and $\log \pi_\theta(y_1 = 1 \mid +) \leq -\frac{1}{3e^{\theta[1]\overline{B}}}$. Hence, for $y^\star \in \mathcal{V}^H$ being $y_h^\star = 1$ for $h \in [H]$, it holds that

$$\log \pi_\mathsf{D}(y^\star \mid +) - \log \pi_\theta(y^\star \mid +) \geq H \cdot \left( \frac{1}{3e^{\theta[1]\overline{B}}} - 3e^{-\overline{B}/2} \right) \geq \log N.$$

The immediately yields

$$\mathsf{Cov}_N(\pi_\theta) \geq \mu(+) \cdot \mathbb{P}_{\pi_\mathsf{D}}(y = y^\star \mid x = +) \geq \frac{\mu(+)}{2}.$$

$\square$

## K.5 Proof of Theorem 5.1 (Coverage for Normalized SGD)

We denote $M := \log N$. We analyze the normalized SGD iterates assuming $\lambda \geq 8BM$ and $\frac{\lambda \eta}{M} \leq \frac{1}{16}$.

Denote

$$\widetilde{g}(\theta; \mathcal{D}) := \frac{\widehat{g}(\theta; \mathcal{D})}{\lambda + \|\widehat{g}(\theta; \mathcal{D})\|}.$$

Then the normalized SGD update can be rewritten as $\theta^{t+1} = \mathrm{Proj}_\Theta(\theta + \eta \widetilde{g}(\theta^t; \mathcal{D}^t))$. Specializing Lemma K.5 to the normalized SGD update and using $\Theta \subseteq \mathbb{B}_2(1)$ yields

$$\sum_{t=1}^T \langle -\widetilde{g}(\theta^t; \mathcal{D}^t), \theta^t - \theta^\star \rangle \leq \frac{2}{\eta} + \eta \sum_{t=1}^T \|\widetilde{g}(\theta^t; \mathcal{D}^t)\|^2.$$

Taking an expectation on both sides and noting that $\mathcal{D}^t \sim \pi_\mathsf{D}$ is generated independently, we have

$$\mathbb{E}\left[ \sum_{t=1}^T \mathbb{E}_{\mathcal{D} \sim \pi_\mathsf{D}} \langle -\widetilde{g}(\theta^t; \mathcal{D}), \theta^t - \theta^\star \rangle \right] \leq \frac{2}{\eta} + \eta \, \mathbb{E}\left[ \sum_{t=1}^T \mathbb{E}_{\mathcal{D} \sim \pi_\mathsf{D}} \|\widetilde{g}(\theta^t; \mathcal{D})\|^2 \right]. \tag{71}$$

In what follows, we prove a number of upper and lower bounds for the expressions involving $\widetilde{g}(\theta; \mathcal{D})$ above, then combine them with Eq. (71) to complete the proof.

**Intermediate bounds.** Recall that we write $\epsilon_\theta(x, y_{1:h-1}) = D_\mathsf{KL}(\pi_\mathsf{D}(\cdot \mid x, y_{1:h-1}) \| \pi_\theta(\cdot \mid x, y_{1:h-1}))$. Also recall that we adopt the notation that for any function $f$ and dataset $\mathcal{D}$, we write $\widehat{\mathbb{E}}_\mathcal{D}[f] := \frac{1}{|\mathcal{D}|} \sum_{(x, y_{1:H}) \in \mathcal{D}} f(x, y_{1:H})$.

Denote (recall that $D_{\mathsf{seq}, N}(\cdot \| \cdot)$ is defined in Proposition F.10)

$$\epsilon_\theta(\mathcal{D}) := \widehat{\mathbb{E}}_\mathcal{D}\left[ \sum_{h=1}^H \epsilon_\theta(x, y_{1:h-1}) \right], \qquad \Delta_\theta := \mathbb{E}_{\pi_\mathsf{D}} \min\{M, \epsilon_\theta(\mathcal{D})\}.$$

Using the key structural result in Proposition F.10 (recall $M := \log N$), we can bound the coverage in terms of the expected sum of stopped KL divergences as follows:

$$\begin{aligned} \mathsf{Cov}_N(\pi_\theta) &\leq \frac{2}{M-1} D_{\mathsf{seq}, N}(\pi_\mathsf{D} \| \pi_\theta) = \frac{2}{M-1} \mathbb{E}_{\pi_\mathsf{D}} \min\left\{ M, \sum_{h=1}^H \epsilon_\theta(x, y_{1:h-1}) \right\} \\ &\leq \frac{2}{M-1} \mathbb{E}_{\mathcal{D} \sim \pi_\mathsf{D}} \min\left\{ M, \widehat{\mathbb{E}}_\mathcal{D}\left[ \sum_{h=1}^H \epsilon_\theta(x, y_{1:h-1}) \right] \right\} = \frac{2}{M-1} \Delta_\theta. \end{aligned} \tag{72}$$

Therefore, it remains to derive upper bounds on $\Delta_\theta$ for $\theta \in \{\theta^1, \cdots, \theta^T\}$.

**Lemma K.8.** *Suppose that $\lambda \geq 8BM$. It holds that for any $\theta \in \Theta$,*

$$\mathbb{E}_{\pi_\mathsf{D}} \|\widetilde{g}(\theta; \mathcal{D})\|^2 \leq \frac{2\Delta_\theta}{M} + \frac{4M\sigma_\star^2}{\lambda^2} + \frac{\sigma_\star}{\lambda\sqrt{K}}.$$

**Lemma K.9.** *Suppose that $\lambda \geq 8BM$. Denote $\Lambda_\theta := \langle -\widetilde{g}(\theta; \mathcal{D}), \theta - \theta^\star \rangle$. Then it holds that for any $\theta \in \Theta$,*

$$\Delta_\theta \leq 8\lambda\Lambda_\theta + \frac{240B}{K} + 8\left(\frac{M\sigma_\star}{\lambda}\right)^2.$$

**Putting everything together.** Under the notation of Lemma K.8 and Lemma K.9, Eq. (71) can be rewritten as

$$\mathbb{E}\left[\sum_{t=1}^T \Lambda_{\theta^t}\right] \leq \frac{2}{\eta} + \frac{\eta}{2}\mathbb{E}\left[\sum_{t=1}^T \mathbb{E}_{\mathcal{D}\sim\pi_{\mathsf{D}}}\|\widetilde{g}(\theta^t; \mathcal{D})\|^2\right]. \tag{73}$$

Applying Lemma K.8 and Lemma K.9, we have

$$\frac{1}{T}\mathbb{E}\left[\sum_{t=1}^T \Delta_{\theta^t}\right] - \frac{240B}{K} - 8\left(\frac{M\sigma_\star}{\lambda}\right)^2 \leq \frac{8\lambda}{T}\mathbb{E}\left[\sum_{t=1}^T \Lambda_{\theta^t}\right]$$

$$\leq \frac{16\lambda}{T\eta} + \frac{4\eta\lambda}{T}\mathbb{E}\left[\sum_{t=1}^T \mathbb{E}_{\mathcal{D}\sim\pi_{\mathsf{D}}}\|\widetilde{g}(\theta^t; \mathcal{D})\|^2\right]$$

$$\leq \frac{16\lambda}{T\eta} + \frac{8\eta\lambda}{MT}\mathbb{E}\left[\sum_{t=1}^T \Delta_{\theta^t}\right] + \frac{16\eta M\sigma_\star^2}{\lambda} + \frac{4\eta\sigma_\star}{\sqrt{K}},$$

where the first inequality uses Lemma K.9, the second inequality follows from Eq. (73), and the last inequality uses Lemma K.8. Therefore, as long as $\eta\lambda \leq \frac{M}{16}$, it holds that

$$\frac{1}{T}\mathbb{E}\left[\sum_{t=1}^T \Delta_{\theta^t}\right] \lesssim \frac{B}{K} + \left(\frac{M\sigma_\star}{\lambda}\right)^2 + \frac{\lambda}{T\eta} + \frac{\eta M\sigma_\star^2}{\lambda} + \frac{\eta\sigma_\star}{\sqrt{K}}.$$

**Simplifying the upper bound.** In the following, we require $\eta \leq \frac{1}{128B}$ and choose $\lambda = \frac{M}{16\eta}$. Then, it holds that

$$\frac{1}{T}\mathbb{E}\left[\sum_{t=1}^T \Delta_{\theta^t}\right] \lesssim \frac{B}{K} + (\eta\sigma_\star)^2 + \frac{M}{T\eta^2} + \frac{\eta\sigma_\star}{\sqrt{K}} \lesssim \frac{B}{K} + (\eta\sigma_\star)^2 + \frac{M}{T\eta^2},$$

where we use AM-GM inequality and $B \geq 1$. Finally, we may choose $\eta = \min\left\{\frac{1}{128B}, \left(\frac{M}{\sigma_\star^2 T}\right)^{1/4}\right\}$. Recall that $M = \log N$, and hence our choice of $\eta$ gives

$$\frac{1}{T}\mathbb{E}\left[\sum_{t=1}^T D_{\mathsf{seq},N}(\pi_{\mathsf{D}}\,\|\,\pi_{\theta^t})\right] \leq \frac{1}{T}\mathbb{E}\left[\sum_{t=1}^T \Delta_{\theta^t}\right] \lesssim \sqrt{\frac{\sigma_\star^2 \log N}{T}} + \frac{B^2\log N}{T} + \frac{B}{K},$$

which implies (by Eq. (72))

$$\frac{1}{T}\mathbb{E}\left[\sum_{t=1}^T \mathsf{Cov}_N(\pi_{\theta^t})\right] \leq \frac{1}{T}\mathbb{E}\left[\sum_{t=1}^T \frac{2}{\log N - 1}D_{\mathsf{seq},N}(\pi_{\mathsf{D}}\,\|\,\pi_{\theta^t})\right] \lesssim \sqrt{\frac{\sigma_\star^2}{T\log N}} + \frac{B^2}{T} + \frac{B}{K\log N}.$$

This is the desired upper bound. $\qquad\square$

**Remark K.1** (Comparison to standard convex optimization analyses). *On a technical level, we find the proof of Theorem 5.1 to be interesting because it does not pass through KL divergence as an intermediate quantity. More broadly, we do not know how to derive the result as an application of standard analysis techniques in optimization (e.g., via a gradient dominance or PL-type condition), but it would be interesting to see if there is a connection.*[11]

---

[11]We further note that the inherent variance $\sigma_\star^2$ corresponds to the gradient variance at the true parameter $\theta^\star$, and hence is tighter than typical analyses that depend on global notions of variance.

**Proof of Lemma K.8.** Note that $\|\widetilde{g}(\theta;\mathcal{D})\| \leq \min\left\{1, \frac{\|\widehat{g}(\theta;\mathcal{D})\|}{\lambda}\right\}$. Recall that

$$\widehat{g}(\theta;\mathcal{D}) = \widehat{\mathbb{E}}_{\mathcal{D}}[\nabla \log \pi_{\theta}(y \mid x)], \qquad \nabla \log \pi_{\theta}(y \mid x) = \sum_{h=1}^{H}\big(\phi(x, y_{1:h}) - \bar{\phi}_{\theta}(x, y_{1:h-1})\big),$$

with the notation introduced at the beginning of Appendix K.

We decompose $\widehat{g}(\theta;\mathcal{D})$ by introducing

$$\bar{g}_{\theta}(\mathcal{D}) := \widehat{\mathbb{E}}_{\mathcal{D}}\left[\sum_{h=1}^{H}\big(\bar{\phi}_{\theta}(x, y_{1:h-1}) - \bar{\phi}_{\theta^{\star}}(x, y_{1:h-1})\big)\right], \tag{74}$$

and

$$z(\mathcal{D}) := \widehat{\mathbb{E}}_{\mathcal{D}}\left[\sum_{h=1}^{H}\phi^{\star}(x, y_{1:h})\right] = \widehat{\mathbb{E}}_{\mathcal{D}}\left[\sum_{h=1}^{H}\big(\phi(x, y_{1:h}) - \bar{\phi}_{\theta^{\star}}(x, y_{1:h-1})\big)\right]. \tag{75}$$

Then, by definition, $-\widehat{g}(\theta;\mathcal{D}) = \bar{g}_{\theta}(\mathcal{D}) - z(\mathcal{D})$. In the following, we first analyze $\|\bar{g}_{\theta}(\mathcal{D})\|$ and $\|z(\mathcal{D})\|$ separately under $\mathcal{D} = \{(x^i, y_{1:H}^i)\}_{i \in [K]} \sim \pi_{\mathsf{D}}$ and summarize the corresponding upper bounds on in Lemma K.10 (stated and proven in the sequel).

Now, using $\|\widetilde{g}(\theta;\mathcal{D})\| \leq \min\left\{1, \frac{\|\widehat{g}(\theta;\mathcal{D})\|}{\lambda}\right\}$, we know

$$\|\widetilde{g}(\theta;\mathcal{D})\|^2 \leq \mathbb{I}\{\epsilon_{\theta}(\mathcal{D}) > M\} + \mathbb{I}\{\epsilon_{\theta}(\mathcal{D}) \leq M\} \cdot \frac{\|\widehat{g}(\theta;\mathcal{D})\|}{\lambda}$$

$$\leq \mathbb{I}\{\epsilon_{\theta}(\mathcal{D}) > M\} + \frac{\mathbb{I}\{\epsilon_{\theta}(\mathcal{D}) \leq M\}}{\lambda} \cdot \left(4\sqrt{\sigma^2(\mathcal{D}) \cdot \epsilon_{\theta}(\mathcal{D})} + 8B\epsilon_{\theta}(\mathcal{D})\right) + \frac{1}{\lambda}\|z(\mathcal{D})\|$$

$$\leq \frac{1}{M}\min\{M, \epsilon_{\theta}(\mathcal{D})\} + \frac{4}{\lambda}\sqrt{\sigma^2(\mathcal{D}) \cdot \min\{M, \epsilon_{\theta}(\mathcal{D})\}} + \frac{1}{\lambda}\|z(\mathcal{D})\|,$$

where the second inequality uses $\|\widehat{g}(\theta;\mathcal{D})\| \leq \|\bar{g}_{\theta}(\mathcal{D})\| + \|z(\mathcal{D})\|$ and Lemma K.10 (2), and the last inequality uses $\lambda \geq 8BM$ and $\frac{1}{M}\min\{M, \epsilon_{\theta}(\mathcal{D})\} = 1$ when $\epsilon_{\theta}(\mathcal{D}) > M$. Taking expectation of $\mathcal{D} \sim \pi_{\mathsf{D}}$, we have

$$\mathbb{E}_{\pi_{\mathsf{D}}}\|\widetilde{g}(\theta;\mathcal{D})\|^2$$

$$\leq \frac{1}{M}\mathbb{E}_{\pi_{\mathsf{D}}}\min\{M, \epsilon_{\theta}(\mathcal{D})\} + \frac{4}{\lambda}\mathbb{E}_{\pi_{\mathsf{D}}}\sqrt{\sigma^2(\mathcal{D}) \cdot \min\{M, \epsilon_{\theta}(\mathcal{D})\}} + \frac{\sigma_{\star}}{\lambda\sqrt{K}}$$

$$\leq \frac{1}{M}\mathbb{E}_{\pi_{\mathsf{D}}}\min\{M, \epsilon_{\theta}(\mathcal{D})\} + \frac{4\sigma_{\star}}{\lambda}\sqrt{\mathbb{E}_{\pi_{\mathsf{D}}}\min\{M, \epsilon_{\theta}(\mathcal{D})\}} + \frac{\sigma_{\star}}{\lambda\sqrt{K}}$$

$$= \frac{\Delta_{\theta}}{M} + \frac{4\sigma_{\star}}{\lambda}\sqrt{\Delta_{\theta}} + \frac{\sigma_{\star}}{\lambda\sqrt{K}}.$$

where the second inequality follows from Cauchy-Schwarz inequality, Lemma K.10 (3) and the fact that $\mathbb{E}[\sigma^2(\mathcal{D})] = \sigma_{\star}^2$. By AM-GM inequality, it holds that $\frac{4\sigma_{\star}}{\lambda}\sqrt{\Delta_{\theta}} \leq \frac{\Delta_{\theta}}{M} + \frac{4M\sigma_{\star}^2}{\lambda^2}$, and the desired upper bound follows immediately. $\square$

**Lemma K.10.** *For any $\theta \in \Theta$, the following holds:*

*(1) It holds that*

$$\langle \bar{g}_{\theta}(\mathcal{D}), \theta - \theta^{\star}\rangle \geq \widehat{\mathbb{E}}_{\mathcal{D}}\left[\sum_{h=1}^{H}\epsilon_{\theta}(x, y_{1:h-1})\right] =: \epsilon_{\theta}(\mathcal{D})$$

*(2) Denote $\sigma^2(\mathcal{D}) := \widehat{\mathbb{E}}_{\mathcal{D}}\left[\sum_{h=1}^{H}\mathrm{Var}_{\pi_{\mathsf{D}}}(x, y_{1:h-1})\right]$. Then*

$$\|\bar{g}_{\theta}(\mathcal{D})\| \leq 4\sqrt{\sigma^2(\mathcal{D}) \cdot \epsilon_{\theta}(\mathcal{D})} + 8B\epsilon_{\theta}(\mathcal{D}).$$

*(3) It holds that $\mathbb{E}_{\mathcal{D} \sim \pi_{\mathsf{D}}}\|z(\mathcal{D})\|^2 = \frac{\sigma_{\star}^2}{K}$ and*

$$\mathbb{E}_{\mathcal{D} \sim \pi_{\mathsf{D}}}\left(\langle z(\mathcal{D}), \theta - \theta^{\star}\rangle - \frac{1}{2}\epsilon_{\theta}(\mathcal{D})\right)_{+} \leq \frac{30B}{K} =: \alpha.$$

**Proof of Lemma K.10.** Lemma K.10 (1) follows immediately from Eq. (58):

$$\langle \bar{g}_\theta(\mathcal{D}), \theta - \theta^\star \rangle = \widehat{\mathbb{E}}_{\mathcal{D}} \left[ \sum_{h=1}^H \langle \bar{\phi}_\theta(x, y_{1:h-1}) - \bar{\phi}_{\theta^\star}(x, y_{1:h-1}), \theta - \theta^\star \rangle \right]$$

$$\geq \widehat{\mathbb{E}}_{\mathcal{D}} \left[ \sum_{h=1}^H \epsilon_\theta(x, y_{1:h-1}) \right] =: \epsilon_\theta(\mathcal{D}).$$

Lemma K.10 (2) follows immediately from Eq. (59):

$$\|\bar{g}_\theta(\mathcal{D})\| \leq \widehat{\mathbb{E}}_{\mathcal{D}} \left[ \sum_{h=1}^H \|\bar{\phi}_\theta(x, y_{1:h-1}) - \bar{\phi}_{\theta^\star}(x, y_{1:h-1})\| \right]$$

$$\leq \widehat{\mathbb{E}}_{\mathcal{D}} \left[ \sum_{h=1}^H 4\sqrt{\mathrm{Var}_{\pi_{\mathsf{D}}}(x, y_{1:h-1}) \cdot \epsilon_\theta(x, y_{1:h-1})} + 8B\epsilon_\theta(x, y_{1:h-1}) \right]$$

$$\leq 4\sqrt{\sigma^2(\mathcal{D}) \cdot \epsilon_\theta(\mathcal{D})} + 8\epsilon_\theta(\mathcal{D}).$$

It remains to prove Lemma K.10 (3). Note that $K \cdot z(\mathcal{D}) = K \cdot \widehat{\mathbb{E}}_{\mathcal{D}} \left[ \sum_{h=1}^H \phi^\star(x, y_{1:h}) \right] = \sum_{i=1}^K \sum_{h=1}^H \phi^\star(x^i, y_{1:h}^i)$ is a sum of the martingale difference sequence $\{\phi^\star(x^i, y_{1:h}^i)\}_{i \in [K], h \in [H]}$. Therefore, we can calculate

$$\mathbb{E}\|z(\mathcal{D})\|^2 = \frac{1}{K} \mathbb{E}_{\pi_{\mathsf{D}}} \left[ \sum_{h=1}^H \|\phi^\star(x, y_{1:h})\|^2 \right] = \frac{\sigma_\star^2}{K}.$$

Furthermore, by Freedman's inequality (Lemma H.1), for any fixed vector $v$, parameter $\gamma \in (0, \frac{1}{B})$ and $\delta \in (0, 1)$, it holds that

$$\mathbb{P}\left( \sum_{i=1}^K \sum_{h=1}^H \left( \langle \phi^\star(x^i, y_{1:h}^i), v \rangle - \gamma \mathbb{E}\left[ \langle \phi^\star(x^i, y_{1:h}^i), v \rangle^2 \mid x^i, y_{1:h-1}^i \right] \right) \geq \gamma^{-1} \log(1/\delta) \right) \leq \delta.$$

Note that for $v = \theta - \theta^\star$, by Lemma H.5, we have

$$\mathbb{E}\left[ \langle \phi^\star(x^i, y_{1:h}^i), v \rangle^2 \mid x^i, y_{1:h-1}^i \right]$$
$$= \mathbb{E}_{y_h \sim \pi_{\mathsf{D}}(\cdot \mid x^i, y_{1:h-1}^i)} \langle \phi(x^i, y_{1:h-1}^i, y_h) - \bar{\phi}_{\theta^\star}(x^i, y_{1:h-1}^i, y_h), \theta - \theta^\star \rangle^2$$
$$\leq 15B D_{\mathsf{KL}}\left( \pi_{\mathsf{D}}(\cdot \mid x^i, y_{1:h-1}^i) \| \pi_\theta(\cdot \mid x^i, y_{1:h-1}^i) \right) = 15B\epsilon_\theta(x^i, y_{1:h-1}^i).$$

Therefore, setting $\gamma = \frac{1}{30B}$, we have shown that for any $\delta \in (0, 1)$, it holds that

$$\mathbb{P}_{\pi_{\mathsf{D}}}\left( \langle z(\mathcal{D}), \theta - \theta^\star \rangle \geq \frac{1}{2} \widehat{\mathbb{E}}_{\mathcal{D}} \left[ \sum_{h=1}^H \epsilon_\theta(x, y_{1:h-1}) \right] + \frac{30B \log(1/\delta)}{K} \right) \leq \delta.$$

Recall that we denote $\epsilon_\theta(\mathcal{D}) := \widehat{\mathbb{E}}_{\mathcal{D}} \left[ \sum_{h=1}^H \epsilon_\theta(x, y_{1:h-1}) \right]$. Then, for the random variable $V := \frac{K}{30B}\left( \langle z(\mathcal{D}), \theta - \theta^\star \rangle - \frac{1}{2}\epsilon_\theta(\mathcal{D}) \right)$, the above inequality implies that for any $u > 0$, $\mathbb{P}(V \geq u) \leq e^{-u}$, and hence $\mathbb{P}((V)_+ \geq u) \leq e^{-u}$. Therefore, integrating out the above inequality gives $\mathbb{E}[(V)_+] \leq 1$, or equivalently,

$$\mathbb{E}_{\pi_{\mathsf{D}}}\left( \langle z(\mathcal{D}), \theta - \theta^\star \rangle - \frac{1}{2}\epsilon_\theta(\mathcal{D}) \right)_+ \leq \frac{30B}{K} =: \alpha.$$

$\square$

**Proof of Lemma K.9.** Recall that we can decompose $-\widehat{g}(\theta; \mathcal{D}) = \bar{g}_\theta(\mathcal{D}) - z(\mathcal{D})$, where $\bar{g}_\theta(\mathcal{D})$ and $z(\mathcal{D})$ are defined in Eq. (74) and Eq. (75), respectively. Then, we know

$$
\begin{aligned}
\Lambda_\theta &:= \mathbb{E}_{\pi_\mathrm{D}}\langle -\widetilde{g}(\theta; \mathcal{D}), \theta - \theta^\star\rangle \\
&= \mathbb{E}_{\pi_\mathrm{D}}\left[\frac{\langle \bar{g}_\theta(\mathcal{D}), \theta - \theta^\star\rangle - \langle z(\mathcal{D}), \theta - \theta^\star\rangle}{\lambda + \|\widehat{g}_\theta(\mathcal{D})\|}\right] \\
&\geq \mathbb{E}_{\pi_\mathrm{D}}\left[\frac{\epsilon_\theta(\mathcal{D}) - \langle z(\mathcal{D}), \theta - \theta^\star\rangle}{\lambda + \|\widehat{g}_\theta(\mathcal{D})\|}\right] \\
&\geq \frac{1}{2}\mathbb{E}_{\pi_\mathrm{D}}\left[\frac{\epsilon_\theta(\mathcal{D})}{\lambda + \|z(\mathcal{D})\| + \|\bar{g}_\theta(\mathcal{D})\|}\right] - \frac{1}{\lambda}\mathbb{E}_{\pi_\mathrm{D}}\left[\left(\langle z(\mathcal{D}), \theta - \theta^\star\rangle - \frac{1}{2}\epsilon_\theta(\mathcal{D})\right)_+\right] \\
&\geq \frac{1}{2}\mathbb{E}_{\pi_\mathrm{D}}\left[\frac{\epsilon_\theta(\mathcal{D})}{\lambda + \|z(\mathcal{D})\| + \|\bar{g}_\theta(\mathcal{D})\|}\right] - \frac{\alpha}{\lambda},
\end{aligned}
$$

where the first inequality uses Lemma K.10 (1) and the last inequality uses Lemma K.10 (3) and we recall that $\alpha = \frac{30B}{K}$. Note that by Lemma K.10 (2),

$$
\begin{aligned}
&\lambda + \|z(\mathcal{D})\| + \|\bar{g}_\theta(\mathcal{D})\| \\
&\leq \lambda + \|z(\mathcal{D})\| + 4\sqrt{\sigma^2(\mathcal{D}) \cdot \epsilon_\theta(\mathcal{D})} + 8B\epsilon_\theta(\mathcal{D}) \\
&\leq \frac{\max\{M, \epsilon_\theta(\mathcal{D})\}}{M} \cdot \left[2\lambda + \|z(\mathcal{D})\| + 4M\sqrt{\frac{\sigma^2(\mathcal{D})}{\min\{M, \epsilon_\theta(\mathcal{D})\}}}\right],
\end{aligned}
$$

where we use $\min\{M, x\}\max\{M, x\} = Mx$, and $\lambda \geq 8BM$. Combining these two inequalities, we have

$$
\begin{aligned}
2\Lambda_\theta + \frac{2\alpha}{\lambda} &\geq \mathbb{E}_{\pi_\mathrm{D}}\left[\frac{\epsilon_\theta(\mathcal{D})}{\lambda + \|z(\mathcal{D})\| + \|\bar{g}_\theta(\mathcal{D})\|}\right] \\
&\geq \mathbb{E}_{\pi_\mathrm{D}}\left[\frac{\min\{M, \epsilon_\theta(\mathcal{D})\}}{2\lambda + \|z(\mathcal{D})\| + 4M\sqrt{\sigma^2(\mathcal{D})/\min\{M, \epsilon_\theta(\mathcal{D})\}}}\right] \\
&\geq \frac{(\mathbb{E}_{\pi_\mathrm{D}}\min\{M, \epsilon_\theta(\mathcal{D})\})^2}{\mathbb{E}_{\pi_\mathrm{D}}\left[\min\{M, \epsilon_\theta(\mathcal{D})\}(2\lambda + \|z(\mathcal{D})\|) + 4M\sqrt{\sigma^2(\mathcal{D}) \cdot \min\{M, \epsilon_\theta(\mathcal{D})\}}\right]} \\
&\geq \frac{(\mathbb{E}_{\pi_\mathrm{D}}\min\{M, \epsilon_\theta(\mathcal{D})\})^2}{2\lambda\mathbb{E}_{\pi_\mathrm{D}}\min\{M, \epsilon_\theta(\mathcal{D})\} + M\sqrt{\sigma_\star^2/K} + 4M\sqrt{\sigma_\star^2\mathbb{E}_{\pi_\mathrm{D}}\min\{M, \epsilon_\theta(\mathcal{D})\}}} \\
&= \frac{\Delta_\theta^2}{2\lambda\Delta_\theta + M\sigma_\star\left[\frac{1}{\sqrt{K}} + 4\sqrt{\Delta_\theta}\right]},
\end{aligned}
$$

where the last two inequalities follow from Cauchy-Schwarz inequality. Therefore, there are two cases: (a) $\Delta_\theta \leq \frac{1}{K}$, and the desired upper bound is trivially true. (b) $\Delta_\theta \geq \frac{1}{K}$, and then it holds that

$$
2\lambda\Delta_\theta + M\sigma_\star\left[\frac{1}{\sqrt{K}} + 4\sqrt{\Delta_\theta}\right] \leq 2\lambda\Delta_\theta + 5M\sigma_\star\sqrt{\Delta_\theta} \leq 3\lambda\Delta_\theta + \frac{8(M\sigma_\star)^2}{\lambda},
$$

where we use AM-GM inequality. Hence, it holds that

$$
\Delta_\theta^2 \leq 8(\lambda\Lambda_\theta + \alpha)\max\left\{\Delta_\theta, 8\left(\frac{M\sigma_\star}{\lambda}\right)^2\right\},
$$

and reorganizing yields

$$
\Delta_\theta \leq 8\max\left\{(\lambda\Lambda_\theta + \alpha), \sqrt{(\lambda\Lambda_\theta + \alpha)\left(\frac{M\sigma_\star}{\lambda}\right)^2}\right\}
$$

$$
\leq 8(\lambda\Lambda_\theta + \alpha) + 8\left(\frac{M\sigma_\star}{\lambda}\right)^2.
$$

This is the desired result. □

### K.6 PROOF OF THEOREM 6.1 (TEST-TIME TRAINING)

Recall (from Eq. (12)) that we consider the token-level SGD iterates defined as

$$\theta^{t,h+1} = \mathrm{Proj}_\Theta\big(\theta^{t,h} + \eta\nabla\log\pi_{\theta^{t,h}}(y_h^t \mid x^t, y_{1:h-1}^t)\big), \quad \text{for } h = 0, \cdots, H-1, \qquad (76)$$

and $\theta^{t+1} \equiv \theta^{t+1,0} := \theta^{t,H}$ for $t \in [T]$, where $(x^t, y_{1:H}^t) \sim \pi_{\mathsf{D}}$.

To define the guarantee on $\theta^t$ which we are able to derive, we next define the following *test-time parameter update* $\vartheta^{\mathsf{TTT}}(x, y_{1:h}; \theta)$, for a parameter $\theta$ and prompt $x$. It is defined recursively for $h = 0, 1, \cdots, H-1$:

$$\vartheta^{\mathsf{TTT}}(x, y_{1:h}; \theta) := \mathrm{Proj}_\Theta\big(\vartheta^{\mathsf{TTT}}(x, y_{1:h-1}; \theta) + \eta\nabla\log\pi_{\vartheta^{\mathsf{TTT}}(x,y_{1:h-1};\theta)}(y_h \mid x, y_{1:h-1})\big). \qquad (77)$$

We then define a distribution $\pi_\theta^{\mathsf{TTT}} : \mathcal{X} \to \Delta(\mathcal{Y}^H)$ as

$$\pi_\theta^{\mathsf{TTT}}(\cdot \mid x, y_{1:h-1}) := \pi_{\vartheta^{\mathsf{TTT}}(x,y_{1:h-1};\theta)}(\cdot \mid x, y_{1:h-1}). \qquad (78)$$

The distribution $\pi_\theta^{\mathsf{TTT}}$ can be interpreted as an augmented version of the autoregressive linear model $\pi_\theta$ that performs test-time training during sampling.

**Proof.** While the algorithm in Theorem 6.1 might seem somewhat complicated and mysterious, the proof is actually a based on a fairly simple online-to-batch conversion argument. We use a number of basic inequalities already found in the proof of Proposition 5.1 (cf. Appendix K.3).

We first note that we can specialize Lemma K.5 to the token-level SGD update (12), and taking expectation gives

$$\mathbb{E}\left[\sum_{t=1}^T\sum_{h=1}^H \big\langle -\nabla\log\pi_{\theta^{t,h}}(y_h^t \mid x^t, y_{1:h-1}^t), \theta^{t,h} - \theta^\star\big\rangle\right] \le \frac{2}{\eta} + \frac{\eta}{2}\mathbb{E}\left[\sum_{t=1}^T\sum_{h=1}^H\big\|\nabla\log\pi_{\theta^{t,h}}(y_h^t \mid x^t, y_{1:h-1}^t)\big\|^2\right].$$
$$(79)$$

In the following, we denote

$$\epsilon_{t,h} := \mathbb{E}\big[\big\langle -\nabla\log\pi_{\theta^{t,h}}(y_h^t \mid x^t, y_{1:h-1}^t), \theta^{t,h} - \theta^\star\big\rangle\big].$$

By triangle inequality,

$$\|\nabla\log\pi_\theta(y_h \mid x, y_{1:h-1})\|^2$$
$$\le 2\|\nabla\log\pi_{\theta^\star}(y_h \mid x, y_{1:h-1})\|^2 + 2\|\nabla\log\pi_\theta(y_h \mid x, y_{1:h-1}) - \nabla\log\pi_{\theta^\star}(y_h \mid x, y_{1:h-1})\|^2.$$

Using the fact that $\theta \mapsto \log\pi_\theta(y_h \mid x, y_{1:h-1})$ is concave and $B^2$-smooth, it holds that for any $\theta$,

$$\|\nabla\log\pi_\theta(y_h \mid x, y_{1:h-1}) - \nabla\log\pi_{\theta^\star}(y_h \mid x, y_{1:h-1})\|^2$$
$$\le B^2 \cdot \langle\theta - \theta^\star, \nabla\log\pi_{\theta^\star}(y_h \mid x, y_{1:h-1}) - \nabla\log\pi_\theta(y_h \mid x, y_{1:h-1})\rangle.$$

Combining the two inequalities above gives that for all $t \in [T]$, $h \in [H]$,

$$\big\|\nabla\log\pi_{\theta^{t,h}}(y_h^t \mid x^t, y_{1:h-1}^t)\big\|^2$$
$$\le 2\big\|\nabla\log\pi_{\theta^\star}(y_h^t \mid x^t, y_{1:h-1}^t)\big\|^2$$
$$+ 2B^2\big\langle\nabla\log\pi_{\theta^\star}(y_h^t \mid x^t, y_{1:h-1}^t) - \nabla\log\pi_{\theta^{t,h}}(y_h^t \mid x^t, y_{1:h-1}^t), \theta^{t,h} - \theta^\star\big\rangle$$

Note that the conditional distribution of $y_h^t \mid (x^t, y_{1:h-1}^t, \theta^{t,h})$ is given by $y_h^t \sim \pi_{\mathsf{D}}(\cdot \mid x^t, y_{1:h-1}^t)$. Hence, taking the expectation over the entire learning process, we have

$$\mathbb{E}\big[\big\|\nabla\log\pi_{\theta^{t,h}}(y_h^t \mid x^t, y_{1:h-1}^t)\big\|^2\big] \le 2\,\mathbb{E}_{\pi_{\mathsf{D}}}\big\|\nabla\log\pi_{\theta^\star}(y_h \mid x, y_{1:h-1})\big\|^2$$
$$+ 2B^2\,\mathbb{E}\big[\big\langle -\nabla\log\pi_{\theta^{t,h}}(y_h^t \mid x^t, y_{1:h-1}^t), \theta^{t,h} - \theta^\star\big\rangle\big]$$
$$= 2\,\mathbb{E}_{\pi_{\mathsf{D}}}[\mathrm{Var}_{\pi_{\mathsf{D}}}(x, y_{1:h-1})] + 2B^2\epsilon_{t,h}.$$

Plugging the above inequality to Eq. (79) yields

$$\sum_{t=1}^T\sum_{h=1}^H \epsilon_{t,h} \le \frac{2}{\eta} + \frac{\eta}{2}\mathbb{E}\left[\sum_{t=1}^T\sum_{h=1}^H\big\|\nabla\log\pi_{\theta^{t,h}}(y_h^t \mid x^t, y_{1:h-1}^t)\big\|^2\right]$$
$$\le \frac{2}{\eta} + \eta T\,\mathbb{E}_{\pi_{\mathsf{D}}}\left[\sum_{h=1}^H \mathrm{Var}_{\pi_{\mathsf{D}}}(x, y_{1:h-1})\right] + \eta B^2\sum_{t=1}^T\sum_{h=1}^H \epsilon_{t,h}.$$

Therefore, as long as $\eta \leq \frac{1}{2B^2}$, it holds that

$$\sum_{t=1}^{T}\sum_{h=1}^{H}\epsilon_{t,h} \leq \frac{4}{\eta} + 2\eta T\,\mathbb{E}_{\pi_{\mathsf{D}}}\left[\sum_{h=1}^{H}\mathrm{Var}_{\pi_{\mathsf{D}}}(x, y_{1:h-1})\right] = \frac{4}{\eta} + 2\eta T\sigma_{\star}^2.$$

By Eq. (58), it also holds that

$$\epsilon_{t,h} = \mathbb{E}\big[\big\langle -\nabla \log \pi_{\theta^{t,h}}(y_h^t \mid x^t, y_{1:h-1}^t), \theta^{t,h} - \theta^{\star}\big\rangle\big] \geq \mathbb{E}\,D_{\mathsf{KL}}\big(\pi_{\mathsf{D}}(\cdot \mid x^t, y_{1:h-1}^t)\,\|\,\pi_{\theta^{t,h}}(\cdot \mid x^t, y_{1:h-1}^t)\big).$$

Combining the inequalities above, as long as $\eta \leq \frac{1}{2B^2}$, we have that

$$\mathbb{E}\left[\sum_{t=1}^{T}\sum_{h=1}^{H}D_{\mathsf{KL}}\big(\pi_{\mathsf{D}}(\cdot \mid x^t, y_{1:h-1}^t)\,\|\,\pi_{\theta^{t,h}}(\cdot \mid x^t, y_{1:h-1}^t)\big)\right] \leq \sum_{t=1}^{T}\sum_{h=1}^{H}\epsilon_{t,h} \leq \frac{4}{\eta} + 2\eta T\sigma_{\star}^2. \quad (80)$$

Finally, we note that

$$\theta^{t,h} = \vartheta^{\mathsf{TTT}}(x^t, y_{h-1}^t; \theta^t),$$

and that for all $t$ and $h$, $x^t, y_{h-1}^t \mid \theta^t \sim \pi_{\mathsf{D}}$. Therefore, we have the following key identity:

$$\mathbb{E}\big[D_{\mathsf{KL}}\big(\pi_{\mathsf{D}}(\cdot \mid x^t, y_{1:h-1}^t)\,\|\,\pi_{\theta^{t,h}}(\cdot \mid x^t, y_{1:h-1}^t)\big) \mid \theta^t\big]$$
$$= \mathbb{E}_{(x,y)\sim\pi_{\mathsf{D}}}\big[D_{\mathsf{KL}}\big(\pi_{\mathsf{D}}(\cdot \mid x, y_{1:h-1})\,\|\,\pi_{\vartheta^{\mathsf{TTT}}(x, y_{1:h-1}; \theta^t)}(\cdot \mid x, y_{1:h-1})\big)\big]$$
$$= \mathbb{E}_{(x,y)\sim\pi_{\mathsf{D}}}\big[D_{\mathsf{KL}}\big(\pi_{\mathsf{D}}(\cdot \mid x, y_{1:h-1})\,\|\,\pi_{\theta^t}^{\mathsf{TTT}}(\cdot \mid x, y_{1:h-1})\big)\big].$$

Combined with Eq. (80), this implies that

$$\frac{4}{\eta} + 2\eta T\sigma_{\star}^2 \geq \mathbb{E}\left[\sum_{t=1}^{T}\sum_{h=1}^{H}D_{\mathsf{KL}}\big(\pi_{\mathsf{D}}(\cdot \mid x^t, y_{1:h-1}^t)\,\|\,\pi_{\theta^{t,h}}(\cdot \mid x^t, y_{1:h-1}^t)\big)\right]$$
$$= \mathbb{E}\left[\sum_{t=1}^{T}\mathbb{E}\left[\sum_{h=1}^{H}D_{\mathsf{KL}}\big(\pi_{\mathsf{D}}(\cdot \mid x^t, y_{1:h-1}^t)\,\|\,\pi_{\theta^{t,h}}(\cdot \mid x^t, y_{1:h-1}^t)\big) \,\Big|\, \theta^t\right]\right]$$
$$= \mathbb{E}\left[\sum_{t=1}^{T}\mathbb{E}_{\pi_{\mathsf{D}}}\left[\sum_{h=1}^{H}D_{\mathsf{KL}}\big(\pi_{\mathsf{D}}(\cdot \mid x, y_{1:h-1})\,\|\,\pi_{\theta^t}^{\mathsf{TTT}}(\cdot \mid x, y_{1:h-1})\big)\right]\right]$$
$$= \mathbb{E}\left[\sum_{t=1}^{T}D_{\mathsf{KL}}\big(\pi_{\mathsf{D}}\,\|\,\pi_{\theta^t}^{\mathsf{TTT}}\big)\right],$$

where the last equality uses the chain rule for KL divergence.

In particular, we may choose $\eta = \min\left\{\frac{1}{2B^2}, \left(\frac{1}{\sigma_{\star}^2 T}\right)^{1/2}\right\}$ to derive $\frac{1}{T}\,\mathbb{E}\left[\sum_{t=1}^{T}D_{\mathsf{KL}}\big(\pi_{\mathsf{D}}\,\|\,\pi_{\theta^t}^{\mathsf{TTT}}\big)\right] \lesssim \sqrt{\frac{\sigma_{\star}^2}{T}} + \frac{B^2}{T}.$ $\qquad\square$

### K.7 Proof of Theorem G.2 (Gradient Normalization for Distillation)

Specializing Lemma K.5 to the update (35) and taking expectation gives

$$\mathbb{E}\left[\sum_{t=1}^{T}\big\langle -\mathbb{E}_{(x,y)\sim\pi_{\mathsf{D}}}[\widehat{g}_{\theta^t}(y \mid x)], \theta^t - \theta^{\star}\big\rangle\right] \leq \frac{2}{\eta} + \frac{\eta}{2}\,\mathbb{E}\left[\sum_{t=1}^{T}\mathbb{E}_{(x,y)\sim\pi_{\mathsf{D}}}\|\widehat{g}_{\theta^t}(y \mid x)\|^2\right]. \quad (81)$$

In the following, we analyze $\big\langle -\mathbb{E}_{(x,y)\sim\pi_{\mathsf{D}}}[\widehat{g}_{\theta}(y \mid x)], \theta^t - \theta^{\star}\big\rangle$ and $\mathbb{E}_{(x,y)\sim\pi_{\mathsf{D}}}\|\widehat{g}_{\theta^t}(y \mid x)\|^2$ for any $\theta \in \Theta$, following the proof of Proposition 5.1 (cf. Appendix K.3).

**Relating the gradient to stopped KL divergence.** Recall that the estimator $\widehat{g}$ is defined in Eq. (35):

$$\widehat{g}_{\theta}(y \mid x) = \sum_{h=1}^{H}\alpha_{\theta}(x, y_{1:h-1})\nabla \log \pi_{\theta}(y_h \mid x, y_{1:h-1}),$$

and the weight function $\alpha_\theta$ is defined in Eq. (36).

We first recall an elementary property of the quantity $\alpha_\theta$. By Lemma K.4, we have

$$\sum_{h=1}^{H} \alpha_\theta(x, y_{1:h-1}) \epsilon_\theta(x, y_{1:h-1}) = \min\left\{ A, \sum_{h=1}^{H} \epsilon_\theta(x, y_{1:h-1}) \right\}, \tag{82}$$

and hence

$$\begin{aligned}
\mathbb{E}_{(x,y)\sim\pi_{\mathsf{D}}} \left[ \sum_{h=1}^{H} \alpha_\theta(x, y_{1:h-1}) \epsilon_\theta(x, y_{1:h-1}) \right] &= \mathbb{E}_{(x,y)\sim\pi_{\mathsf{D}}} \min\left\{ A, \sum_{h=1}^{H} \epsilon_\theta(x, y_{1:h-1}) \right\} \\
&= D_{\mathsf{seq},N}(\pi_{\mathsf{D}} \,\|\, \pi_\theta),
\end{aligned} \tag{83}$$

where we recall that $D_{\mathsf{seq},N}(\pi_{\mathsf{D}} \,\|\, \pi_\theta)$ is defined in Proposition F.10 and we denote $A = \log N$. Hence,

$$\begin{aligned}
&\left\langle -\mathbb{E}_{(x,y)\sim\pi_{\mathsf{D}}}[\widehat{g}_\theta(y \mid x)], \theta - \theta^\star \right\rangle \\
&= \mathbb{E}_{(x,y)\sim\pi_{\mathsf{D}}} \left[ \sum_{h=1}^{H} \alpha_\theta(x, y_{1:h-1}) \left\langle \bar{\phi}_\theta(x, y_{1:h-1}) - \bar{\phi}_{\theta^\star}(x, y_{1:h-1}), \theta - \theta^\star \right\rangle \right] \\
&\geq \mathbb{E}_{(x,y)\sim\pi_{\mathsf{D}}} \left[ \sum_{h=1}^{H} \alpha_\theta(x, y_{1:h-1}) \epsilon_\theta(x, y_{1:h-1}) \right] = D_{\mathsf{seq},N}(\pi_{\mathsf{D}} \,\|\, \pi_\theta),
\end{aligned} \tag{84}$$

where the inequality uses Eq. (58).

In addition, the following lemma shows that $\mathbb{E}_{(x,y)\sim\pi_{\mathsf{D}}} \|\widehat{g}_\theta(y \mid x)\|^2$ is well-controlled.

**Lemma K.11** (Gradient error bound). *For any $\theta \in \Theta$, it holds that*

$$\mathbb{E}_{(x,y)\sim\pi_{\mathsf{D}}} \|\widehat{g}_\theta(y \mid x)\|^2 \leq (64A + 2)\sigma_\star^2 + 256AB^2 D_{\mathsf{seq},N}(\pi_{\mathsf{D}} \,\|\, \pi_\theta).$$

**Putting everything together.** Finally, combining the inequalities above, we know that

$$\begin{aligned}
\mathbb{E}\left[ \sum_{t=1}^{T} D_{\mathsf{seq},N}(\pi_{\mathsf{D}} \,\|\, \pi_{\theta^t}) \right] &\leq \mathbb{E}\left[ \sum_{t=1}^{T} \left\langle -\mathbb{E}_{(x,y)\sim\pi_{\mathsf{D}}}[\widehat{g}_{\theta^t}(y \mid x)], \theta^t - \theta^\star \right\rangle \right] \\
&\leq \frac{2}{\eta} + \frac{\eta}{2} \mathbb{E}\left[ \sum_{t=1}^{T} \mathbb{E}_{(x,y)\sim\pi_{\mathsf{D}}} \|\widehat{g}_{\theta^t}(y \mid x)\|^2 \right] \\
&\leq \frac{2}{\eta} + \eta T(32A + 1)\sigma_\star^2 + 128AB^2 \mathbb{E}\left[ \sum_{t=1}^{T} D_{\mathsf{seq},N}(\pi_{\mathsf{D}} \,\|\, \pi_{\theta^t}) \right],
\end{aligned}$$

where the first inequality uses Eq. (84), the second inequality follows from Eq. (81), and the third inequality uses Lemma K.11. Therefore, as long as $\eta \leq \frac{1}{2(32A+1)B^2}$, it holds that

$$\mathbb{E}\left[ \sum_{t=1}^{T} D_{\mathsf{seq},N}(\pi_{\mathsf{D}} \,\|\, \pi_{\theta^t}) \right] \lesssim \frac{1}{\eta} + \eta T A \sigma_\star^2.$$

In particular, we may choose $\eta = \min\left\{ \frac{1}{(64\log N + 2)B^2}, \left( \frac{1}{T\sigma_\star^2 \log N} \right)^{1/2} \right\}$ and derive

$$\mathbb{E}\left[ \frac{1}{T} \sum_{t=1}^{T} D_{\mathsf{seq},N}(\pi_{\mathsf{D}} \,\|\, \pi_{\theta^t}) \right] \lesssim \sqrt{\frac{\sigma_\star^2 \log N}{T}} + \frac{B^2 \log N}{T}.$$

By Proposition F.10, this implies

$$\mathbb{E}\left[ \frac{1}{T} \sum_{t=1}^{T} \mathsf{Cov}_N(\pi_{\theta^t}) \right] \lesssim \sqrt{\frac{\sigma_\star^2}{T \log N}} + \frac{B^2}{T}.$$

$$\square$$

**Proof of Lemma K.11.** Fix any $\theta \in \Theta$. By triangle inequality, it holds that

$$\|\widehat{g}_\theta(y \mid x) - \widehat{g}_{\theta^\star}(y \mid x)\|$$

$$\leq \sum_{h=1}^{H} \alpha_\theta(x, y_{1:h-1}) \|\bar{\phi}_{\theta^\star}(x, y_{1:h}) - \bar{\phi}_\theta(x, y_{1:h-1})\|$$

$$\leq \sum_{h=1}^{H} \alpha_\theta(x, y_{1:h-1}) \left(4\sqrt{\mathrm{Var}_{\pi_{\mathrm{D}}}(x, y_{1:h-1}) \cdot \epsilon_\theta(x, y_{1:h-1})} + 8B\epsilon_\theta(x, y_{1:h-1})\right)$$

$$\leq 4 \left(\sum_{h=1}^{H} \alpha_\theta(x, y_{1:h-1})\mathrm{Var}_{\pi_{\mathrm{D}}}(x, y_{1:h-1})\right)^{1/2} \left(\sum_{h=1}^{H} \alpha_\theta(x, y_{1:h-1})\epsilon_\theta(x, y_{1:h-1})\right)^{1/2}$$

$$+ 8B \sum_{h=1}^{H} \alpha_\theta(x, y_{1:h-1})\epsilon_\theta(x, y_{1:h-1})$$

$$\leq 4\sqrt{A \cdot \sum_{h=1}^{H} \mathrm{Var}_{\pi_{\mathrm{D}}}(x, y_{1:h-1})} + 8B \min\left\{A, \sum_{h=1}^{H} \epsilon_\theta(x, y_{1:h-1})\right\}.$$

where the second inequality follows from Eq. (59), the third inequality follows from Cauchy-Schwarz inequality, and the final lines follow from the property (82) of the weight function $\alpha_\theta \in [0, 1]$. Hence, using $(a + b)^2 \leq 2a^2 + 2b^2$, we have

$$\|\widehat{g}_\theta(y \mid x) - \widehat{g}_{\theta^\star}(y \mid x)\|^2$$

$$\leq 32A \left(\sum_{h=1}^{H} \mathrm{Var}_{\pi_{\mathrm{D}}}(x, y_{1:h-1})\right) + 128AB^2 \min\left\{A, \sum_{h=1}^{H} \epsilon_\theta(x, y_{1:h-1})\right\}.$$

Therefore, taking expectation of $(x, y) \sim \pi_{\mathrm{D}}$ and using $\mathbb{E}_{\pi_{\mathrm{D}}}\|\widehat{g}_{\theta^\star}(y \mid x)\|^2 \leq \sigma_\star^2$ and Eq. (83), it holds that

$$\mathbb{E}_{(x,y)\sim\pi_{\mathrm{D}}}\|\widehat{g}_\theta(y \mid x)\|^2 \leq (64A + 2)\sigma_\star^2 + 256AB^2 D_{\mathsf{seq}, N}(\pi_{\mathrm{D}} \| \pi_\theta).$$

This is the desired upper bound. $\square$

### K.8 NECESSITY OF VARIANCE DEPENDENCE IN HIGH DIMENSION

We generalize Proposition 3.2 to show that in the worst case (where $\sigma_\star^2 \asymp HB^2$), the scaling $\mathsf{Cov}_N(\widehat{\pi}) = \Omega(\frac{H}{n \log N})$ can be unavoidable for autoregressive linear model. This implies that the dependence on $\sigma_\star^2$ is generally necessary to achieve upper bounds that do not explicitly scale with $H$.

**Proposition K.1.** *Let $H, B, N, n \geq 1$, and assume $\log N \leq c \min\{H, B^2\}$ for a sufficiently small constant $c > 0$. There exists an instance of the autoregressive linear model class $\Pi$ with $d = H$, $\phi : \mathcal{X} \times \mathcal{V}^\star \to \mathbb{B}_2(B)$, and $\Theta = \mathbb{B}_2(1)$, such that for any proper algorithm $\mathtt{Alg}$ with output $\widehat{\pi} = \pi_{\widehat{\theta}}$ for $\widehat{\theta} \in \Theta$, there exists $\pi_{\mathrm{D}} \in \Pi$, such that under $\pi_{\mathrm{D}}$, it holds that*

$$\mathbb{E}^{\pi_{\mathrm{D}}, \mathtt{Alg}}[\mathsf{Cov}_N(\pi_{\mathrm{D}} \| \widehat{\pi})] \geq c \cdot \min\left\{1, \frac{H}{n \cdot \log N}\right\}.$$

**Proof of Proposition K.1.** We consider $\mathcal{X} = \{+, -\}$, $\mathcal{V} = \{0, 1\}$, and the distribution $\mu$ be given by $\mu(+) = 1 - \mu(-) = p$, where $p \in [0, 1]$ is a parameter to be chosen later. Let the feature map $\phi$ be given by $\phi(-, y_{1:h}) = 0$, $\phi(+, y_{1:h}) = By_h e_h$, where $(e_1, \cdots, e_H)$ is a fixed orthonormal basis of $\mathbb{R}^H$. Note that with this construction, we have $\pi_\theta(y_h = \cdot \mid -, y_{1:h-1}) = \mathrm{Ber}(1/2)$, and

$$\pi_\theta(y_h = \cdot \mid +, y_{1:h-1}) = \mathrm{Ber}\left(\frac{e^{B\theta_h}}{1 + e^{B\theta_h}}\right) =: \pi_{\theta, h}.$$

Note that for any $h \in [H]$, we can bound

$$C_0 B|\theta_h - \theta'_h| \le D_{\mathsf{H}}(\pi_{\theta,h}, \pi_{\theta',h}) \le C_1 B|\theta_h - \theta'_h|,$$

as long as $\theta_h \in [-\frac{1}{B}, \frac{1}{B}]$.

We fix $\epsilon \in [0, \frac{1}{\max\{\sqrt{H}, B\}}]$ to be determined later, and for any $v \in \{-1, 1\}^H$, we let $\theta_v := \epsilon \sum_{h=1}^H v_h e_h$, and

$$\Theta_0 := \left\{\theta_v : v \in \{-1, 1\}^H\right\} \subset \mathbb{B}_2(1), \qquad \Pi_0 := \{\pi_\theta : \theta \in \Theta_0\}.$$

Then a direct argument (see e.g., (Wainwright, 2019, Section 15.3)) shows that when $pn \le \frac{c_0}{B^2 \epsilon^2}$ for a sufficiently small constant $c_0$, there exists $\theta^\star \in \Theta_0$ such that under $\pi_{\mathsf{D}} = \pi_{\theta^\star}$, it holds that

$$\sum_{h=1}^H \mathbb{P}^{\pi_{\mathsf{D}}, \mathrm{Alg}}\left(|\widehat{\theta}_h - \theta_h^\star| \ge \epsilon\right) \ge cH.$$

Therefore, with probability at least $\frac{c}{2}$, it holds that $\sum_{h=1}^H \mathbb{I}\left\{|\widehat{\theta}_h - \theta_h^\star| \ge \epsilon\right\} \ge \frac{cH}{2}$, and this in turn implies

$$\sum_{h=1}^H D_{\mathsf{H}}^2\left(\pi_{\theta^\star, h}, \pi_{\widehat{\theta}, h}\right) \ge c_1 H B^2 \epsilon^2.$$

Then, by Proposition F.11, we know that under the above event, as long as $\log N \le \frac{c_1 H B^2 \epsilon^2}{2}$, we have $\mathsf{Cov}_N(\widehat{\pi}) \ge \frac{p}{2}$. Choosing $\epsilon = \sqrt{\frac{4 \log N}{c_1 H B^2}}$ and $p = \min\{1, \frac{c_0}{n B^2 \epsilon^2}\}$ gives the desired lower bound. $\qquad\square$

## L    PROOFS FROM SECTION 6

### L.1    PROOF OF THEOREM 6.2 (SIMPLE TOURNAMENT)

Below we state and prove a generalization of Theorem 6.2 which holds when the data distribution $\pi_{\mathsf{D}}$ is not necessarily in the model class $\Pi$.

**Theorem 6.2′** (General version of Theorem 6.2). *Fix $N \ge 1$, and consider the estimator $\widehat{\pi}$ from Eq. (14):*

$$\widehat{\pi} := \arg\min_{\pi \in \Pi} \max_{\pi' \in \Pi} \widehat{\mathsf{Cov}}_N(\pi' \,\|\, \pi). \tag{85}$$

*For any $\delta \in (0, 1)$, parameter $a, c \ge 0$, with probability at least $1 - \delta$, it holds that*

$$\mathsf{Cov}_{N^{1+a+2c}}(\widehat{\pi}) \lesssim \min_{\overline{\pi} \in \Pi} \mathsf{Cov}_{N^a}(\overline{\pi}) + \frac{1}{N^{1-a-2c}} + \frac{\log \mathcal{N}_\infty(\Pi; c \log N) + \log \delta^{-1}}{n}. \tag{86}$$

**Proof of Theorem 6.2′.** Fix $N, N' \ge 1, \alpha > 0$, and let $\overline{\pi} \in \arg\min_{\pi \in \Pi} \mathsf{Cov}_{N'}(\pi_{\mathsf{D}} \,\|\, \pi)$. We study the estimator

$$\widehat{\pi} := \arg\min_{\pi \in \Pi} \max_{\pi' \in \Pi} \widehat{\mathsf{Cov}}_N(\pi' \,\|\, \pi). \tag{87}$$

Recall that we denote $\mathsf{Cov}_N^{\pi_{\mathsf{D}}}(\pi' \,\|\, \pi) = \mathbb{P}_{\pi_{\mathsf{D}}}\left(\frac{\pi'(y|x)}{\pi(y|x)} \ge N\right)$ (cf. Lemma J.2). By Lemma J.2, with probability at least $1 - \frac{\delta}{2}$, it holds that

$$\widehat{\mathsf{Cov}}_N(\overline{\pi} \,\|\, \pi) \ge \frac{1}{2} \mathsf{Cov}_{e^{2\alpha} N}^{\pi_{\mathsf{D}}}(\overline{\pi} \,\|\, \pi) - \varepsilon_{\mathsf{stat}}, \qquad \forall \pi \in \Pi, \tag{88}$$

where $\varepsilon_{\mathsf{stat}} = \frac{8 \log(4 \mathcal{N}_\infty(\Pi, \alpha)/\delta)}{n}$. Next, again by Lemma J.2, with probability at least $1 - \frac{\delta}{2}$, it holds that

$$\widehat{\mathsf{Cov}}_N(\pi \,\|\, \overline{\pi}) \le 2 \mathsf{Cov}_{e^{-2\alpha} N}^{\pi_{\mathsf{D}}}(\pi \,\|\, \overline{\pi}) + \varepsilon_{\mathsf{stat}}, \qquad \forall \pi \in \Pi. \tag{89}$$

In the following, we condition on the success event of Eq. (88) and Eq. (89). Then, we can bound

$$
\begin{aligned}
\frac{1}{2}\mathsf{Cov}^{\pi_{\mathsf{D}}}_{e^{2\alpha}N}(\overline{\pi}\,\|\,\widehat{\pi}) - \varepsilon_{\mathsf{stat}} \leq \widehat{\mathsf{Cov}}_N(\overline{\pi}\,\|\,\widehat{\pi}) &\leq \max_{\pi'\in\Pi}\widehat{\mathsf{Cov}}_N(\pi'\,\|\,\widehat{\pi}) \\
&= \min_{\pi\in\Pi}\max_{\pi'\in\Pi}\widehat{\mathsf{Cov}}_N(\pi'\,\|\,\pi) \leq \max_{\pi'\in\Pi}\widehat{\mathsf{Cov}}_N(\pi'\,\|\,\overline{\pi}) \\
&\leq 2\max_{\pi'\in\Pi}\mathsf{Cov}^{\pi_{\mathsf{D}}}_{e^{-2\alpha}N}(\pi'\,\|\,\overline{\pi}) + \varepsilon_{\mathsf{stat}}.
\end{aligned}
$$

Reorganizing yields

$$
\mathsf{Cov}^{\pi_{\mathsf{D}}}_{e^{2\alpha}N}(\overline{\pi}\,\|\,\widehat{\pi}) \leq 4\max_{\pi\in\Pi}\mathsf{Cov}^{\pi_{\mathsf{D}}}_{e^{-2\alpha}N}(\pi\,\|\,\overline{\pi}) + 4\varepsilon_{\mathsf{stat}}. \tag{90}
$$

Note that for any $N''$ and models $\pi$, $\pi'$, $\pi''$,

$$
\mathsf{Cov}^{\pi_{\mathsf{D}}}_{N'N''}(\pi'\,\|\,\pi) \leq \mathsf{Cov}^{\pi_{\mathsf{D}}}_{N'}(\pi'\,\|\,\pi'') + \mathsf{Cov}^{\pi_{\mathsf{D}}}_{N''}(\pi''\,\|\,\pi). \tag{91}
$$

Hence, for any model $\pi\in\Pi$,

$$
\mathsf{Cov}^{\pi_{\mathsf{D}}}_{e^{2\alpha}NN'}(\pi_{\mathsf{D}}\,\|\,\pi) \leq \mathsf{Cov}^{\pi_{\mathsf{D}}}_{N'}(\pi_{\mathsf{D}}\,\|\,\overline{\pi}) + \mathsf{Cov}^{\pi_{\mathsf{D}}}_{e^{2\alpha}N}(\overline{\pi}\,\|\,\pi), \tag{92}
$$
$$
\mathsf{Cov}^{\pi_{\mathsf{D}}}_{e^{-2\alpha}N}(\pi\,\|\,\overline{\pi}) \leq \mathsf{Cov}^{\pi_{\mathsf{D}}}_{N'}(\pi\,\|\,\pi_{\mathsf{D}}) + \mathsf{Cov}^{\pi_{\mathsf{D}}}_{e^{-2\alpha}N/N'}(\pi_{\mathsf{D}}\,\|\,\overline{\pi}). \tag{93}
$$

Therefore, combining the inequalities above, we see that

$$
\begin{aligned}
\mathsf{Cov}_{e^{2\alpha}NN'}(\widehat{\pi}) = \mathsf{Cov}^{\pi_{\mathsf{D}}}_{e^{2\alpha}NN'}(\pi_{\mathsf{D}}\,\|\,\widehat{\pi}) \\
&\leq \mathsf{Cov}^{\pi_{\mathsf{D}}}_{N'}(\pi_{\mathsf{D}}\,\|\,\overline{\pi}) + \mathsf{Cov}^{\pi_{\mathsf{D}}}_{e^{2\alpha}N}(\overline{\pi}\,\|\,\widehat{\pi}) \\
&\leq \mathsf{Cov}^{\pi_{\mathsf{D}}}_{N'}(\pi_{\mathsf{D}}\,\|\,\overline{\pi}) + 4\max_{\pi\in\Pi}\mathsf{Cov}^{\pi_{\mathsf{D}}}_{e^{-2\alpha}N}(\pi\,\|\,\overline{\pi}) + 4\varepsilon_{\mathsf{stat}} \\
&\leq 5\mathsf{Cov}^{\pi_{\mathsf{D}}}_{N'}(\pi_{\mathsf{D}}\,\|\,\overline{\pi}) + 4\max_{\pi\in\Pi}\mathsf{Cov}^{\pi_{\mathsf{D}}}_{e^{-2\alpha}N/N'}(\pi\,\|\,\pi_{\mathsf{D}}) + 4\varepsilon_{\mathsf{stat}} \\
&\leq 5\mathsf{Cov}^{\pi_{\mathsf{D}}}_{N'}(\pi_{\mathsf{D}}\,\|\,\overline{\pi}) + \frac{e^{2\alpha}N'}{N} + 4\varepsilon_{\mathsf{stat}},
\end{aligned}
$$

where the first inequality uses Eq. (92), the second inequality uses Eq. (90), the third inequality uses Eq. (93), and the last inequality follows from the fact that $\mathsf{Cov}^{\pi_{\mathsf{D}}}_A(\pi\,\|\,\pi_{\mathsf{D}}) = \mathbb{P}_{\pi_{\mathsf{D}}}\left(\frac{\pi(y|x)}{\pi_{\mathsf{D}}(y|x)} \geq A\right) \leq \frac{1}{A}$.

The claimed bound (86) follows by setting $\alpha = c\log N$, and $N' = N^a$. $\qquad\square$

## L.2 Proof of Theorem G.3 (Offset Tournament)

**Divergence.** For distributions $P, Q \in \Delta(\mathcal{Y})$, we define the following divergence for $N \geq 1$:[12]

$$
\mathcal{E}_N(P\,\|\,Q) := \max\left\{\mathbb{E}_{y\sim P}\left(\frac{dQ}{dP} - N\right)_+, \mathbb{E}_{y\sim Q}\left(\frac{dP}{dQ} - N\right)_+\right\} \in [0, 1].
$$

Then, for models $\pi, \pi' : \mathcal{X} \to \Delta(\mathcal{Y})$, we further define

$$
\mathcal{E}_{N,\mu}(\pi\,\|\,\pi') := \mathbb{E}_{x\sim\mu}\,\mathcal{E}_N(\pi(\cdot\mid x)\,\|\,\pi'(\cdot\mid x)).
$$

Under this divergence, it holds that for any event $E$,

$$
\mathbb{P}_{\mu,\pi}(E) \leq N \cdot \mathbb{P}_{\mu,\pi'}(E) + \mathcal{E}_{N,\mu}(\pi\,\|\,\pi'), \tag{94}
$$
$$
\mathbb{P}_{\mu,\pi'}(E) \leq N \cdot \mathbb{P}_{\mu,\pi}(E) + \mathcal{E}_{N,\mu}(\pi\,\|\,\pi'), \tag{95}
$$

where $\mathbb{P}_{\mu,\pi}$ is the probability under $x\sim\mu$ and $y\sim\pi(\cdot\mid x)$. Furthermore, we can bound

$$
\mathsf{Cov}_{2N}(\pi) = \mathbb{P}_{\mu,\pi_{\mathsf{D}}}\left(\frac{\pi_{\mathsf{D}}(y\mid x)}{\pi(y\mid x)} \geq 2N\right) \leq \mathcal{E}_{N,\mu}(\pi_{\mathsf{D}}\,\|\,\pi). \tag{96}
$$

---

[12]This divergence is inspired by Huang et al. (2025b), but our definition differs slightly from the standard $\mathcal{E}_M$-divergence (Polyanskiy, 2010; Block & Polyanskiy, 2023).

**Theorem G.3′** (General version of Theorem G.3). *Fix $N, \gamma \geq 1$ such that $N \geq 8\gamma^2$. Consider the estimator*

$$\widehat{\pi} := \arg\min_{\pi \in \Pi} \max_{\pi' \in \Pi} \left\{ \widehat{\mathrm{Cov}}_N(\pi' \,\|\, \pi) - 2\gamma \cdot \widehat{\mathrm{Cov}}_N^\pi(\pi' \,\|\, \pi) \right\}. \tag{97}$$

*Then with probability $1 - \delta$, it holds that*

$$\mathrm{Cov}_{2N\gamma}(\widehat{\pi}) \lesssim \min_{\pi \in \Pi} \mathcal{E}_\gamma(\pi_\mathsf{D} \,\|\, \pi) + \frac{\log(|\Pi|/\delta)}{n}.$$

Note that $\mathcal{E}_\gamma(\pi_\mathsf{D} \,\|\, \pi) = 0$ when $|\log \pi_\mathsf{D}(y \mid x) - \log \overline{\pi}(y \mid x)| \leq \log \gamma$ for any $x \in \mathcal{X}, y \in \mathcal{Y}$. Therefore, Theorem G.3 is an immediate corollary by setting $\gamma = N^a$.

**Proof of Theorem G.3′.** For $\pi, \pi' \in \Pi$, we define the set

$$\mathcal{C}_N(\pi, \pi') = \left\{ (x, y) \mid \frac{\pi(y \mid x)}{\pi'(y \mid x)} \geq N \right\}.$$

Suppose an i.i.d. dataset $\mathcal{D} = \{(x^i, y^i)\}_{i \in [n]} \sim \pi_\mathsf{D}$ is drawn. We write $\widehat{\mathbb{P}}_n = \frac{1}{n} \sum_{i=1}^n \delta_{(x^i, y^i)}$ and $\mu_n = \frac{1}{n} \sum_{i=1}^n \delta_{x^i}$ to denote the empirical measures (i.e., $\widehat{\mathbb{P}}_n$ is the uniform distribution over $\mathcal{D}$), and let $\mathbb{P}_{\mu_n, \pi}$ be the probability under the distribution $x \sim \mu_n, y \sim \pi(\cdot \mid x)$. Under this notation, we have $\widehat{\mathrm{Cov}}_N(\pi' \,\|\, \pi) = \widehat{\mathbb{P}}_n(\mathcal{C}_N(\pi', \pi))$ and we also recall that

$$\widehat{\mathrm{Cov}}_N^{\overline{\pi}}(\pi' \,\|\, \pi) := \frac{1}{n} \sum_{i=1}^n \mathbb{P}_{y \sim \overline{\pi}(\cdot \mid x^i)} \left( \frac{\pi'(y \mid x^i)}{\pi(y \mid x^i)} \geq N \right) = \mathbb{P}_{\mu_n, \overline{\pi}}(\mathcal{C}_N(\pi', \pi)).$$

Thus, the tournament estimator in Eq. (97) can be expressed as

$$\widehat{\pi} := \arg\min_{\pi \in \Pi} \max_{\pi' \in \Pi} \mathcal{L}(\pi, \pi'), \tag{98}$$

where

$$\mathcal{L}(\pi, \pi') := \widehat{\mathbb{P}}_n(\mathcal{C}_N(\pi', \pi)) - 2\gamma \cdot \mathbb{P}_{\mu_n, \overline{\pi}}(\mathcal{C}_N(\pi', \pi)). \tag{99}$$

As an immediate consequence of Lemma H.2 and the union bound, we have the following lemma.

**Lemma L.1.** *Fix $\delta \in (0, 1)$, and define $\varepsilon_{\mathsf{stat}} = \frac{16 \log(16|\Pi|/\delta)}{n}$. With probability $1 - \delta$, the following bounds hold simultaneously:*

*(1) For all $\pi, \pi' \in \Pi$, it holds that*

$$2\mathbb{P}_{\mu, \pi_\mathsf{D}}(\mathcal{C}_N(\pi', \pi)) + \varepsilon_{\mathsf{stat}} \geq \widehat{\mathbb{P}}_n(\mathcal{C}_N(\pi', \pi)) \geq \frac{1}{2}\mathbb{P}_{\mu, \pi_\mathsf{D}}(\mathcal{C}_N(\pi', \pi)) - \varepsilon_{\mathsf{stat}}, \tag{100}$$

$$2\mathbb{P}_{\mu_n, \pi_\mathsf{D}}(\mathcal{C}_N(\pi', \pi)) + \varepsilon_{\mathsf{stat}} \geq \widehat{\mathbb{P}}_n(\mathcal{C}_N(\pi', \pi)) \geq \frac{1}{2}\mathbb{P}_{\mu_n, \pi_\mathsf{D}}(\mathcal{C}_N(\pi', \pi)) - \varepsilon_{\mathsf{stat}}. \tag{101}$$

*(2) For any $\pi \in \Pi$, it holds that $\mathcal{E}_{\gamma, \mu_n}(\pi_\mathsf{D} \,\|\, \pi) \leq 2\mathcal{E}_{\gamma, \mu}(\pi_\mathsf{D} \,\|\, \pi) + \varepsilon_{\mathsf{stat}}$.*

In the following, we fix $\delta \in (0, 1)$ and condition on the success event of Lemma L.1. Let $\overline{\pi} \in \arg\min_{\pi \in \Pi} \mathcal{E}_{\gamma, \mu}(\pi_\mathsf{D} \,\|\, \pi)$. We denote $\varepsilon_{\mathsf{apx}} = \mathcal{E}_{\gamma, \mu}(\pi_\mathsf{D} \,\|\, \overline{\pi})$ and $\varepsilon'_{\mathsf{apx}} = \mathcal{E}_{\gamma, \mu_n}(\pi_\mathsf{D} \,\|\, \overline{\pi})$. Note that by Lemma L.1, we have $\varepsilon'_{\mathsf{apx}} \leq 2\varepsilon_{\mathsf{apx}} + \varepsilon_{\mathsf{stat}}$.

Then, for any $\pi' \in \Pi$,

$$\mathcal{L}(\overline{\pi}, \pi') \leq 2\mathbb{P}_{\mu_n, \pi_\mathsf{D}}(\mathcal{C}_N(\pi', \overline{\pi})) - 2\gamma \mathbb{P}_{\mu_n, \overline{\pi}}(\mathcal{C}_N(\pi', \overline{\pi})) + \varepsilon_{\mathsf{stat}}$$
$$\leq 2\mathcal{E}_{\gamma, \mu_n}(\pi_\mathsf{D} \,\|\, \overline{\pi}) + \varepsilon_{\mathsf{stat}} = \varepsilon'_{\mathsf{apx}} + \varepsilon_{\mathsf{stat}}.$$

where the first inequality uses Eq. (101), and the second inequality uses Eq. (94).

Therefore, we have

$$\max_{\pi' \in \Pi} \mathcal{L}(\widehat{\pi}, \pi') = \min_{\pi \in \Pi} \max_{\pi' \in \Pi} \mathcal{L}(\pi, \pi') \leq \max_{\pi' \in \Pi} \mathcal{L}(\overline{\pi}, \pi') \leq \varepsilon_{\mathsf{stat}} + \varepsilon'_{\mathsf{apx}}.$$

In particular, we know $\mathcal{L}(\widehat{\pi}, \overline{\pi}) \leq \varepsilon_{\mathsf{stat}} + \varepsilon'_{\mathsf{apx}}$. Then, we can bound

$$
\begin{aligned}
\widehat{\mathbb{P}}_n(\mathcal{C}_N(\overline{\pi}, \widehat{\pi})) - \mathcal{L}(\widehat{\pi}, \overline{\pi}) &= 2\gamma \mathbb{P}_{\mu_n, \widehat{\pi}}(\mathcal{C}_N(\overline{\pi}, \widehat{\pi})) \\
&\leq \frac{2\gamma}{N} \mathbb{P}_{\mu_n, \overline{\pi}}(\mathcal{C}_N(\overline{\pi}, \widehat{\pi})) \\
&\leq \frac{2\gamma}{N} \big[ \gamma \mathbb{P}_{\mu_n, \pi_{\mathsf{D}}}(\mathcal{C}_N(\overline{\pi}, \widehat{\pi})) + \varepsilon'_{\mathsf{apx}} \big] \\
&\leq \frac{2\gamma}{N} \Big[ 2\gamma \Big( \widehat{\mathbb{P}}_n(\mathcal{C}_N(\overline{\pi}, \widehat{\pi})) + \varepsilon_{\mathsf{stat}} \Big) + \varepsilon'_{\mathsf{apx}} \Big],
\end{aligned}
$$

where the first inequality follows from the fact that $\overline{\pi}(y \mid x) \geq N\widehat{\pi}(y \mid x)$ for $(x, y) \in \mathcal{C}_N(\overline{\pi}, \widehat{\pi})$, the second inequality uses Eq. (95): $\mathbb{P}_{\mu_n, \overline{\pi}}(E) - \gamma \mathbb{P}_{\mu_n, \pi_{\mathsf{D}}}(E) \leq \mathcal{E}_{\gamma, \mu_n}(\pi_{\mathsf{D}} \| \overline{\pi}) = \varepsilon'_{\mathsf{apx}}$ for any event $E$, and the third inequality uses Eq. (101). Therefore, using $N \geq 8\gamma^2$, we know $\widehat{\mathbb{P}}_n(\mathcal{C}_N(\overline{\pi}, \widehat{\pi})) \leq 5\varepsilon_{\mathsf{stat}} + 2\varepsilon'_{\mathsf{apx}}$. Then, using Eq. (100), we have

$$
\mathsf{Cov}_N^{\pi_{\mathsf{D}}}(\overline{\pi} \| \widehat{\pi}) = \mathbb{P}_{\mu, \pi_{\mathsf{D}}}(\mathcal{C}_N(\overline{\pi}, \widehat{\pi})) \leq 2\widehat{\mathbb{P}}_n(\mathcal{C}_N(\overline{\pi}, \widehat{\pi})) + 2\varepsilon_{\mathsf{stat}} \leq 12\varepsilon_{\mathsf{stat}} + 4\varepsilon'_{\mathsf{apx}}.
$$

By Eq. (91), it holds that

$$
\mathsf{Cov}_{2N\gamma}(\widehat{\pi}) = \mathsf{Cov}_{2N\gamma}^{\pi_{\mathsf{D}}}(\pi_{\mathsf{D}} \| \widehat{\pi}) \leq \mathsf{Cov}_{2\gamma}^{\pi_{\mathsf{D}}}(\pi_{\mathsf{D}} \| \overline{\pi}) + \mathsf{Cov}_N^{\pi_{\mathsf{D}}}(\overline{\pi} \| \widehat{\pi}),
$$

and we also have $\mathsf{Cov}_{2\gamma}^{\pi_{\mathsf{D}}}(\pi_{\mathsf{D}} \| \overline{\pi}) = \mathsf{Cov}_{2\gamma}(\overline{\pi}) \leq \mathcal{E}_{\gamma, \mu}(\pi_{\mathsf{D}} \| \overline{\pi}) = \varepsilon_{\mathsf{apx}}$ by Eq. (96). Combining the inequalities above, we can conclude that

$$
\mathsf{Cov}_{2N\gamma}(\widehat{\pi}) \leq \mathsf{Cov}_N^{\pi_{\mathsf{D}}}(\overline{\pi} \| \widehat{\pi}) + \varepsilon_{\mathsf{apx}} \leq 12\varepsilon_{\mathsf{stat}} + 4\varepsilon'_{\mathsf{apx}} + \varepsilon_{\mathsf{apx}}.
$$

Finally, using Lemma H.2, we have $\varepsilon'_{\mathsf{apx}} \leq 2\varepsilon_{\mathsf{apx}} + \varepsilon_{\mathsf{stat}}$. This is the desired upper bound. $\qquad\square$

