# OpenReview forum: "The Coverage Principle: How Pre-Training Enables Post-Training"
_ICLR.cc/2026/Conference — ICLR 2026 Oral_

### Official Review · Reviewer_w8sn · 2025-10-30

**Soundness:** 4
**Presentation:** 4
**Contribution:** 3
**Rating:** 8
**Confidence:** 4

**Summary:**

In this paper authors introduce a new quantity called coverage profile which is better aligned with a model's performance on downstream tasks that require Best-of-N sampling and/or RL. They show that autoregressive next-token prediction implicitly optimizes coverage. Coverage has better generalization and convergence than CE. While CE can continuously improve during pre-training, coverage can drop at some point which may reflect the decrease in the BoN downstream performance.
The authors perform experiments with graph reasoning tasks to support their theoretical results. They present a way to estimate coverage profile in practice by defining a Tournament procedure to select checkpoints with the best in-class coverage profile. Moreover, they show that test-time training (TT token-level SGD) improves coverage.

**Strengths:**

- The theoretical justification of the coverage principle that can explain insufficiency of cross-entropy for accurately predicting downstream BoN performance is an important contribution.
- The paper is well-written and solid. Theoretical results are sound.
- Algorithmic improvements like gradient normalization in SGD can lead to better coverage and therefore better BoN performance.
- The work opens a new interesting direction in connection of the base model and post-training performance by utilizing the notion of coverage profile.

**Weaknesses:**

- The only concern that I have is that the authors validated their theory on only one task - graph reasoning.

**Questions:**

- Can the scaling law (eq. 3 and therefore Proposition3.1) be validated empirically (e.g. using the tournament procedure)?
-  Figure 1, why coverage has non-integer index (e.g. $Cov_{5.7}$), if $N$ is the number of sampling attempts (eq. 1) or does $N$ mean something else here?

---

> ### Author Response · Authors · 2025-11-20
>
> Thank you for the review! We address each weakness and question below.
>
> ---
>
> > Q1: Can the scaling law (eq. 3 and therefore Proposition3.1) be validated empirically (e.g. using the tournament procedure)?
>
> Existing empirical studies cited in the paper consistently show that RL post-training performance (as well as Best-of-N performance) correlates strongly with the base model’s Pass@N, which is tightly connected to coverage. In contrast, the scaling law in Eq. 3 (and Proposition 3.1) provides only an upper bound on coverage in terms of cross-entropy and therefore on the Best-of-N budget. This bound can be loose in practice. Indeed, both our theoretical insights and the experiment in Figure 1 indicate that coverage can deteriorate even as cross-entropy continues to improve.
>
> Therefore, we believe it is empirically feasible to demonstrate cases where cross-entropy fails to predict downstream RL or Best-of-N performance, effectively “disproving” Eq. 3 and demonstrating coverage as a more reliable proxy.
>
> ---
>
> > Q2: Figure 1, why coverage has non-integer index (e.g. Cov 5.7 ), if N is the number of sampling attempts (eq. 1) or does N mean something else here?
>
> We apologize for the notational clash. To help clarify, in the following we will refer to Best-of-N as Best-of-$M$, where $M$ denotes the number of sampling attempts.
>
> The parameter $N$ in the coverage profile $\mathrm{Cov}_{N}$ is a positive real number which is closely related to the number of Best-of-$M$ sampling attempts, but is not exactly the same. The precise relationship is given in Propositions D.6 and D.7, which we summarize below:
>
> - Upper bound: Given $M$ Best-of-$M$ sampling attempts, the suboptimality for Best-of-$M$ is upper bounded by $\mathrm{Cov}_{N}$ for $N=M/(2\log(1/\delta))$, where \delta is a failure probability parameter.
>
> - Lower bound: Given $M$ Best-of-$M$ sampling attempts, the suboptimality for Best-of-$M$ is also *lower bounded* by $\mathrm{Cov}_{N}$ for $N=2M$ under a certain reward function.
>
> Empirically, we found that setting $N\approx M/2$ (with $M$ is the number of sampling attempts) is a good rule of thumb. This is exactly the relationship we use when plotting the performance of Best-of-M and $\mathrm{Cov}\_{N}$ in our experiments. More specifically, we take the grid of $N\in\\{ 2,3,4,4.6,5.7 \\}$, and plot the curve for $\mathrm{Cov}\_{N}$ and Pass@$M$ with $M=\lceil 2N \rceil$.
>
> We will be sure to clarify this in the final version.

---

### Official Review · Reviewer_RfJt · 2025-11-01

**Soundness:** 3
**Presentation:** 3
**Contribution:** 2
**Rating:** 6
**Confidence:** 2

**Summary:**

Motivated by the observation that downstream results are not always correlated with cross entropy results, this paper introduces coverage that is better related to downstream performance. The authors show through theoretical results that next-token prediction leads to models with good coverage. From a practitioner's perspective, the authors then demonstrate a decoding intervention to improve coverage, and a way to use coverage for selecting models.

**Strengths:**

- well motivated and tied to prior work, figure 1 makes this clear
- theory is sound and directly relates to BoN and post-trained models
- also contains practical recommendations (section 6)

**Weaknesses:**

- lack of experimental evidence. would be nice to expand on the downstream tasks and post-training methods.
- coverage assumes a target distribution, which may limit its applicability for pre-training
- coverage relies on being able to verify samples, which may be difficult for certain (more open-ended) tasks

**Questions:**

Coverage seems to focus on diversity more than quality - could over indexing on this during pre-training make post-training more difficult?

Would you recommend using this instead of perplexity during pre-training? If so, how would you expect it to behave in a scaling law study?

---

> ### Author Response · Authors · 2025-11-20
>
> Thank you for the review! We address each weakness and question below.
>
> ---
>
> > W2: coverage assumes a target distribution, which may limit its applicability for pre-training
>
> Our analysis considers two distributions: the task-specific distribution $\pi_{\mathsf{T}}$ and the pre-training distribution $\pi_{\mathsf{D}}$. We would appreciate clarification from the reviewer regarding which of these they refer to as the “target distribution.”
>
>
> **Regarding the task-specific distribution $\pi_{\mathsf{T}}$: We would like to emphasize that our work only uses this distribution as an analysis tool; it is not explicitly used within any of the algorithms under consideration.**
>
> In particular, our main theoretical results focus purely on the pre-training setting, where the training data are drawn from a distribution $\pi_{\mathsf{D}}$, and we assess a model $\widehat{\pi}$ in terms of its coverage of $\pi_{\mathsf{D}}$. By a “transivity”-type argument, coverage of $\pi_{\mathsf{D}}$ implies coverage for any downstream task distribution $\pi_{\mathsf{T}}$ that is itself covered by $\pi_{\mathsf{D}}$ (which we view as a minimal assumption).
>
> - Concretely, for any downstream task governed by $\pi_{\mathsf{T}}$, the coverage can be bounded as (informally) $\mathrm{Cov}(\pi_\mathsf{T} \|\| \widehat{\pi}) \lesssim \mathrm{Cov}(\pi_\mathsf{T} \|\| \pi_\mathsf{D}) + \mathrm{Cov}(\pi_\mathsf{D} \|\| \widehat{\pi})$ ---we have omitted the dependence on $N$ here for compactness (see Proposition D.5 for a formal statement).
>
> - Thus, whenever the pre-training distribution $\pi_{\mathsf{D}}$ sufficiently covers the downstream distribution $\pi_{\mathsf{T}}$, our guarantees imply that a next-token-prediction model will also achieve good coverage of $\pi_{\mathsf{T}}$. This shows that coverage is applicable and informative in both pre-training and post-training regime.
>
> **Regarding the pre-training distribution $\pi_{\mathsf{D}}$:** While it is true that coverage itself depends on the the probabilities $\pi_{\mathsf{D}}(y|x)$, which are not available during pre-training, our results in Section 6.2 show that it is still possible to optimize for coverage via pairwise comparison (where the idea is to replace $\pi_{\mathsf{D}}(y|x)$ with a measurable proxy). While these results are preliminary, they indicate that coverage—while not directly measurable—can still be a useful lens for designing algorithms.
>
> ---
>
> > Q1: Coverage seems to focus on diversity more than quality - could over indexing on this during pre-training make post-training more difficult?
>
> We view coverage—specifically, the quantity $\mathrm{Cov}(\pi_\mathsf{D} \|\| \widehat{\pi})$ we analyze throughout the paper—as capturing the extent to which $\widehat{\pi}$ inherits diversity present in the pre-training distribution $\pi_{\mathsf{D}}$. Whether this is desirable for post-training depends on the quality of the pre-training distribution itself. If the pre-training distribution is a diverse mixture of quality task-specific data, then good coverage relative to this distribution should lead to better post-training performance. Conversely, if $\pi_{\mathsf{D}}$ consists largely of low quality data, then indeed coverage may not lead to good downstream performance. However, this is an inherent limitation of pre-training with next-token prediction. Thus, we view the (tacit) assumption that $\pi_{\mathsf{D}}$ covers downstream tasks (discussed on Page 3) as fairly minimal.
>
> Let us also mention in passing that prior work has demonstrated that high Pass@N performance (which is closely tied to coverage, as shown in Proposition D.6/D.7) is highly correlated with success for standard RL post-training methods such as GRPO. (e.g., Yue et al. 2025).
>
> ---
>
> > W3: coverage relies on being able to verify samples, which may be difficult for certain (more open-ended) tasks
>
> We agree that for a downstream task of interest with a reward function $r(x,y)$, computing the coverage profile with respect to a task-specific policy $\pi_{\mathsf{T}}$ may require access to the reward function. However, our main theoretical results concern coverage relative to the pre-training distribution $\pi_{\mathsf{D}}$. As discussed above, this in turn implies coverage for the downstream task-specific distribution $\pi_{\mathsf{T}}$, whenever $\pi_{\mathsf{D}}$ itself covers $\pi_{\mathsf{T}}$ (which we view as a minimal assumption). Hence, we believe that coverage of the pre-training distribution $\pi_{\mathsf{D}}$ as a useful proxy for downstream coverage—one that motivates new algorithms and interventions (such as the checkpoint selection methods in Section 6.2). Please also see our response to W2 for more details.

---

> ### Author Response · Authors · 2025-11-20
> **Response (Part 2)**
>
> > Q2: Would you recommend using this instead of perplexity during pre-training? If so, how would you expect it to behave in a scaling law study?
>
> **We emphasize that coverage is not meant to replace perplexity or cross-entropy, but rather to refine and complement them.** As we highlight in Remark C.1, KL divergence/cross-entropy captures the mean of the log-density ratio $\log \pi_{\mathsf{D}}/\widehat{\pi}$, whereas the coverage profile can be viewed as the cumulative distribution function (CDF) of $\log \pi_{\mathsf{D}}/\widehat{\pi}$. The CDF contains more information than any single statistic such as the mean.
>
> Thus, based on our theoretical findings, we do recommend monitoring coverage during pre-training, when feasible (see also our response to W2). Coverage can reveal phenomena that cross-entropy or perplexity alone might not capture, as illustrated in Figure 1.
>
> **Regarding scaling laws:** We believe that understanding the scaling laws for coverage is an important direction, one that we would be excited to explore in future work. In particular, it would be interesting to see if coverage can reveal why some models are easier to post-train than others at scale, even when they have similar cross-entropy. We note that there is already significant empirical evidence that RL performance correlates strongly with the base model’s Pass@N, which in turn is tightly connected to coverage (e.g., Yue et al. 2025).

---

### Official Review · Reviewer_Au4Y · 2025-11-03

**Soundness:** 4
**Presentation:** 4
**Contribution:** 4
**Rating:** 8
**Confidence:** 3

**Summary:**

This paper proposes the coverage principle: next-token pre-training implicitly drives models toward good coverage over high-quality responses, which better predicts post-training success than cross-entropy alone. The authors provide theoretical justification using autoregressive linear models and some assumptions such as assumption on convexity.
Based on the findings, the paper further presents a test-time training decoding scheme that updates logits while sampling; this improves coverage and can outperform pure MLE-based guarantees.In addition, it also provides a method to select checkpoints with better pass@k than cross-entropy-based selection.
This paper poses new insights on understanding the relationship between pretraining and downstream performance and provide effective ways to utilize these insights in practice.

**Strengths:**

1. The paper provides a solid theoretical justification on the proposed coverage hypothesis.
2. The insights can be translated to useful applications, including better test-time sampling methods and better checkpoint selection methods.
3. This paper can inspire future works to develop algorithms that can be more effectively used for improving coverage using methods in addition to traditional next-token-prediction loss.

**Weaknesses:**

1. As a general limitation of the theoretical justification of deep learning, the proofs are shown with linear autoregressive models and assumptions on convexity, therefore the takeaways may have a risk on directly being translated to complex deep neural networks.
2. The hypothesis does not directly indicate a better way than next-token-prediction for pretraining to improve the coverage during training. The proposed distillation-only training is unrealistic in pretriaining.
3. The experiments are conducted on a relatively toy cases, e.g., a graph reasoning task, while real-world tasks are more complicated.

**Questions:**

1. Would it be possible to demonstrate the checkpoint selection or decoding methods using an LLM and complicated tasks, such as HLE, AIME, and factual QAs like SimpleQA, etc.?
2. Your results often rely on the data distribution \pi_D covering the downstream task. Can you formalize how much misspecification in \pi_D is tolerable while keeping useful coverage guarantees?
3. For the gradient normalization trick, what’s a plug-and-play recipe for transformers and how to implement it?
4. How robust is the proposed TTT sampler to distribution shift, long contexts, and temperature samplig?
5. Results focus on graph reasoning with GPT-2–style models. Can you provide additional evidence on natural-language tasks (e.g., code-gen Pass@N or reasoning benchmarks) to test the coverage–performance link?
6. Can you provide an intuitive example where cross-entropy improves while coverage worsens, and a rule-of-thumb for detecting this in training curves?
7. Any other suggestions for LLM pretraining/mid-training before postraining?

---

> ### Author Response · Authors · 2025-11-20
>
> Thank you for the review! We address each weakness and question below.
>
> ---
>
> > W1: As a general limitation of the theoretical justification of deep learning, the proofs are shown with linear autoregressive models and assumptions on convexity, therefore the takeaways may have a risk on directly being translated to complex deep neural networks.
>
> We would like to clarify that our main result (Theorem 4.1), as well as the checkpoint-selection results (Section 6.2 and Appendix L.3), apply to general model classes $\Pi$ and rely only on the covering number of the class. While our *optimization results* indeed rely on the autoregressive linear model assumption, we view these as an important starting point for future investigation, and would be excited to explore broader/deeper classes of models in future work. In particular, the autoregressive linear model can be viewed as a transformer-based language model with weights in all but the final layer frozen. Understanding this simplified class is a natural starting point toward understanding deeper models.
>
> ---
>
> > W2: The hypothesis does not directly indicate a better way than next-token-prediction for pretraining to improve the coverage during training. The proposed distillation-only training is unrealistic in pretraining.
>
> **We would like to emphasize that our results do suggest techniques for coverage during pretraining, notably the checkpoint selection procedures discussed in Section 6.2. These methods apply to general pre-training and are not restricted to distillation**. In this setup, we assume access to a small collection of candidate models (e.g., checkpoints) and aim to identify the one with the best coverage.  In the revision, we will be sure to highlight this aspect of Section 6.2 as explicitly as possible.
>
> ---
>
> > Q1: Would it be possible to demonstrate the checkpoint selection or decoding methods using an LLM and complicated tasks, such as HLE, AIME, and factual QAs like SimpleQA, etc.?
>
> We agree this is an important direction and would be excited to explore this in future work, computational resources permitting.
>
> ---
>
>
> > Q2: Your results often rely on the data distribution \pi_D covering the downstream task. Can you formalize how much misspecification in \pi_D is tolerable while keeping useful coverage guarantees?
>
> The amount of misspecification required by our results can be quantified through the best-in-class coverage, given by
> $\min_{\pi \in \Pi} \mathrm{Cov}(\pi_{\mathsf{D}} \|\| \pi).$
> As long as this quantity is small—that is, the model class $\Pi$ contains at least one model with good coverage of $\pi_{\mathsf{D}}$—the coverage guarantees continue to hold (see Theorem 6.2 and Theorem F.1). Importantly, this requires only that $\Pi$ contain a model with good coverage, which is a much weaker requirement than that $\Pi$ contains a model with low KL divergence, as discussed in the paper.
>
> ---
>
> > Q3: For the gradient normalization trick, what’s a plug-and-play recipe for transformers and how to implement it?
>
> Our gradient-normalization results are motivated by the mechanics of Adam-style optimizers and can be viewed as providing a new form of theoretical support for their effectiveness in pretraining. In this sense, standard Adam (or AdamW) already serves as a practical plug-and-play recipe. Our analysis provides a new perspective on why this widely adopted intervention can be beneficial. We believe that developing more refined or explicitly coverage-aware optimization methods inspired by these ideas is an exciting direction for future work.
>
> ---
>
> > Q4: How robust is the proposed TTT sampler to distribution shift, long contexts, and temperature sampling?
>
> From a theoretical standpoint, our analysis is directly motivated by the distribution-shift challenges that arise along long trajectories. Specifically, we show that the convergence rate of coverage scales only with the effective trajectory length $\sigma_\star$ (Theorem 4.2), rather than the full trajectory length inherent in the KL (Proposition 3.2).
>
> Regarding temperature sampling, we do not have any specific reason to expect the TTT sampler to behave worse than standard autoregressive generation. A deeper understanding of its interaction with temperature remains an interesting topic for future work.
>
> ---
>
> > Q5: Results focus on graph reasoning with GPT-2–style models. Can you provide additional evidence on natural-language tasks (e.g., code-gen Pass@N or reasoning benchmarks) to test the coverage–performance link?
>
> These are all excellent questions and very promising directions for further exploration. Unfortunately, due to resource constraints, we are unable to provide additional large-scale experiments during the rebuttal period.

---

> ### Author Response · Authors · 2025-11-20
> **Response (Part 2)**
>
> > Q6: Can you provide an intuitive example where cross-entropy improves while coverage worsens, and a rule-of-thumb for detecting this in training curves?
>
> Figure 1 provides precisely such an example: while cross-entropy/KL continues to improve, coverage clearly degrades. This behavior aligns with a core motivating example in our theory: the “missing mass” phenomenon (Proposition 3.2 and Remark 3.1). Here, the data distribution contains a mixture of frequently occurring “easy” prompts and rare “hard” prompts. Minimizing cross-entropy incentivizes the model to overfit on the easy prompts, which may in turn reduce coverage on the hard ones.
>
> Importantly, standard cross-entropy training curves in Figure 1 do not reveal this degradation in coverage, which further underscores the need for developing training-time diagnostics and criteria inspired by our coverage framework.
>
> We will point this out more explicitly in the revision.
>
> ---
>
> > Q7: Any other suggestions for LLM pretraining/mid-training before postraining?
>
> As discussed in our response to W2, we believe that our checkpoint-selection procedures could be particularly helpful during pretraining. Although our interventions are primarily motivated by post-training, they should also be beneficial during mid-training. Furthermore, if mid-training involves distillation from a larger model, then our distillation-based techniques naturally apply, since the probabilities of the teacher model are available.

---

### Meta-Review · Area_Chair_QmkW · 2026-01-02

**Summary:**

The paper introduces the coverage principle, finding that next-token pretraining implicitly optimizes coverage, which better predicts downstream Best-of-N and post-training performance than cross-entropy. It provides sound theoretical justification, proposes coverage-based checkpoint selection, and introduces a test-time training decoding method. Experiments on graph reasoning tasks support the claims.

Across reviews, the work is viewed as well-motivated, clearly written, and theoretically sound, offering novel conceptual insight into the relationship between pretraining objectives and post-training performance, with promising practical implications, such as  for decoding and checkpoint selection. A major concern lies in its limited empirical validation — just on a single (graph reasoning) task. Providing additional evidence on natural-language tasks (e.g., code-gen Pass@N or reasoning benchmarks) can better test the generalization and practical value of coverage–performance link. Overall, it is good to accept.

**Reviewer Concerns:**

There are two concerns: (1) empirical validation is limited to a single (graph reasoning) task; (2) theory relies on simplified models and assumptions. Authors are unable to provide additional experiments for (1), and partially address (2) by clarification.

**Reviewer Scores:**

8

---

### Decision · Program_Chairs · 2026-01-26

Accept (Oral)